# PROMPTING DECISION TRANSFORMERS FOR ZERO-SHOT REACH-AVOID POLICIES

## ABSTRACT

Offline goal-conditioned reinforcement learning methods have shown promise for reach-avoid tasks, where an agent must reach a target state while avoiding undesirable regions of the state space. Existing approaches typically encode avoid-region information into an augmented state space and cost function, which prevents flexible, dynamic specification of novel avoid-region information at evaluation time. They also rely heavily on well-designed reward and cost functions, limiting scalability to complex or poorly structured environments. We introduce RADT, a decision transformer model for offline, reward-free, goal-conditioned, avoid-region-conditioned RL. RADT encodes goals and avoid-regions directly as prompt tokens, allowing any number of avoid-regions of arbitrary size to be specified at evaluation time. Using only suboptimal offline trajectories from a random policy, RADT learns reach-avoid behavior through a novel combination of goal and avoid-region hindsight relabeling. We benchmark RADT against 3 existing offline goal-conditioned RL models across 17 tasks, environments, and experimental settings. RADT generalizes in a zero-shot manner to out-of-distribution avoid-region sizes and counts, outperforming baselines that require retraining. In one such zero-shot setting, RADT achieves 35.7% improvement in normalized cost over the best retrained baseline while maintaining high goal-reaching success. We also apply RADT to cell reprogramming in biology, demonstrating its versatility.

## 1 INTRODUCTION

Many high-risk sequential decision-making problems (Liu et al., 2024; Gronauer, 2022; Abouelazm et al., 2024) are naturally framed as reach-avoid tasks (Hsu* et al., 2021; So et al., 2024) (Feng et al., 2025), in which an agent must reach a designated goal state while avoiding undesirable regions of the state space. These problems arise in diverse domains, including robotics (Gronauer, 2022; Ray et al., 2019; Cao et al., 2024) (e.g., robotic arms reaching for targets while avoiding fragile objects), autonomous navigation (Liu et al., 2024; Abouelazm et al., 2024) (e.g., self-driving vehicles avoiding pedestrians), and biology (Wuputra et al., 2020; Lin et al., 2024b) (e.g., cell reprogramming strategies that aim to reach a therapeutic gene expression state without traversing tumorigenic intermediates). Despite domain-specific differences, these tasks share a common structure: they require balancing goal achievement with dynamic avoidance of specified hazards.

Solving reach-avoid problems is especially important in safety-critical environments where entering undesirable states can have irreversible consequences. These environments often preclude online exploration, making offline learning necessary (Liu et al., 2024). Furthermore, in practical deployments, the specification of goals and avoid-regions may change based on user preferences or environmental context. For instance, a robot assistant may need to adapt to new furniture layouts, or a therapeutic model may need to avoid different toxic intermediate states based on patient-specific risk factors. These settings require flexible and interpretable models that support zero-shot generalization to unseen goal and avoid specifications without retraining.

However, reach-avoid learning remains difficult. Several lines of work attempt to address parts of the reach-avoid problem (Section 3), but fail to meet one or more key criteria needed for flexible and effective reach-avoid learning, including: offline learning from suboptimal data, dynamic test-time conditioning on arbitrary goals and avoid-regions, and reward-free training (Section 2, Figure 1). Most existing approaches rely on augmented state representations and carefully shaped cost functions to encode avoid behavior (Cao et al., 2024; Xu et al., 2022a; Zheng et al., 2024; Lee et al., 2022; Le et al., 2019; Liu et al., 2024). This tight coupling of avoid-region semantics to model internals

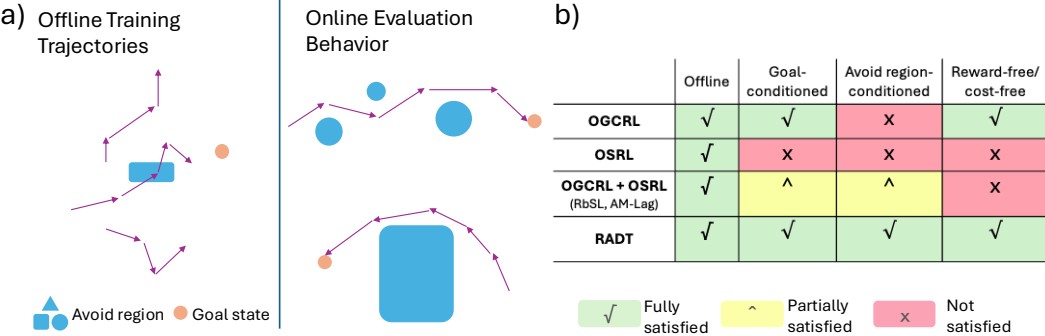

**Figure 1:** (a) An ideal reach-avoid model should learn to avoid arbitrarily specified regions of varying number and size at evaluation time, using only suboptimal, random-policy training data. (b) RADT is the only model that satisfies all criteria for an ideal reach-avoid learner (Section 2).

prevents flexible deployment and limits generalization. Reward-based formulations often struggle to represent multiple behavioral preferences simultaneously (Abouelazm et al., 2024; Freitag et al., 2024; Knox & MacGlashan, 2024), especially when goal-reaching and avoidance conflict. Reward-free methods avoid these issues but lack a mechanism for dynamically conditioning behavior on new avoid constraints (Ghosh et al., 2019; Yang et al., 2022; Janner et al., 2021; Eysenbach et al., 2022). Moreover, many offline approaches rely on expert demonstrations or near-optimal data to learn strong policies (Park et al., 2025; Liu et al., 2024; Cao et al., 2024; Fujimoto et al., 2019; Kumar et al., 2019), which are often not available in safety-critical or high-dimensional tasks (Gangopadhyay et al., 2024; Kumar et al., 2022; Nishimori et al., 2024).

**Present work.** We introduce RADT (Reach-Avoid Decision Transformer), a reward-free, offline RL model for goal-conditioned and avoid-region-conditioned reach-avoid learning (Figure 1). RADT is a decision transformer that represents goals and avoid-regions as prompt tokens. This formulation decouples the reach-avoid specification from the state representation and enables zero-shot generalization to arbitrary numbers and sizes of avoid-regions. RADT learns policies entirely from random-policy trajectories using a novel combination of goal and avoid-region hindsight relabeling, with no need for reward or cost functions. Our main contributions include: ① A prompting framework for reach-avoid learning that encodes goals and avoid-regions as discrete prompt tokens, allowing flexible and interpretable conditioning of behavior at test time. ② A novel avoid-region hindsight relabeling strategy that allows the model to learn successful avoid behavior from suboptimal data. ③ A decision transformer model trained on random-policy data with no reward or cost functions, enabling reach-avoid learning in entirely offline, reward-free settings. ④ Benchmarking across robotics and biological domains evaluates generalization to out-of-distribution avoid-region sizes and counts. ⑤ Strong empirical results showing that RADT generalizes zero-shot to 12 unseen reach-avoid configurations, outperforming 3 existing methods retrained directly on those configurations.

## 2 DESIDERATA/DESIRABLE PROPERTIES OF REACH-AVOID RL MODELS

**Notation.** We first establish the notation used throughout this work. A trajectory $\tau$ of length $T$ is a sequence of alternating states and actions collected by an agent making sequential decisions in an environment:

$$\tau = (\mathbf{s}_1, \mathbf{a}_1, \mathbf{s}_2, \mathbf{a}_2, \ldots, \mathbf{s}_T, \mathbf{a}_T)$$

where $\mathbf{s}_t \in \mathcal{S}$ is the state of the agent at timestep $t$ and $\mathbf{a}_t \in \mathcal{A}$ is the action the agent takes from $\mathbf{s}_t$. $\mathcal{S} \subseteq \mathbb{R}^{d_s}$ represents the state space and $\mathcal{A} \subseteq \mathbb{R}^{d_a}$ represents the action space, which are the spaces of all possible states and actions, respectively. All RL models in this work learn a deterministic policy $\pi(\mathbf{a}_t \mid \cdot)$ that selects the most preferred action $\hat{\mathbf{a}}_t$ for the agent to take at timestep $t$ given contextual inputs, typically including at least the current state $\mathbf{s}_t$ (i.e., $\pi(\mathbf{a}_t \mid \mathbf{s}_t)$). At each timestep of a trajectory, a *state transition* occurs, where the agent takes an action $\mathbf{a}$ from their current state $\mathbf{s}$ and ends up in a new state $\mathbf{s}'$. This is denoted by the transition tuple $(\mathbf{s}, \mathbf{a}, \mathbf{s}')$. If used, the reward function $r(\mathbf{s}, \mathbf{a}, \mathbf{s}')$ and cost function $\texttt{cost}(\mathbf{s}, \mathbf{a}, \mathbf{s}')$ return scalar values for each transition tuple in the trajectory. When applicable, these values are included in the trajectory as follows:

$$\tau = (\mathbf{s}_1, \mathbf{a}_1, r_1, c_1, \mathbf{s}_2, \mathbf{a}_2, \dots)$$

In goal-conditioned settings, a goal state $\mathbf{g} \in \mathcal{S}$ is provided as an additional input, yielding conditional functions such as $\pi(\mathbf{a} \mid \cdot, \mathbf{g})$ or $r(\mathbf{s}, \mathbf{a}, \mathbf{s}', \mathbf{g})$. In avoid-region-conditioned settings with $n_{\text{avoid}}$ avoid-regions $\mathbf{b}_j : j \in \{1, 2, ..., n_{\text{avoid}}\}$, similar conditioning applies. We refer to the center of avoid-region $\mathbf{b}_j$ as its avoid centroid, denoted $\texttt{centroid}(\mathbf{b}_j)$.

**Desiderata/Desirable Properties.** Reach-avoid problems introduce a dual behavioral objective: the agent must reach a desired target state while avoiding explicitly defined regions of the state space. Reach-avoid models that satisfy this behavioral objective under real-world deployment constraints need to achieve the following key properties:

**Property 1 (P1): Pre-collected offline datasets with no online fine-tuning.** The model must learn solely from offline datasets $\mathcal{D}$ containing of pre-collected trajectories $\tau^{(i)}$, with no reliance on online fine-tuning. In safety-critical applications, online exploration may be infeasible, especially when entering avoid-regions could cause irreversible harm or failure.

**Property 2 (P2): Suboptimality-tolerant learning.** The model must learn strong policies from offline datasets that contain only suboptimal trajectories, i.e., those that do not reach the goal or that violate the avoid constraint. Specifically, it should support super-demonstration performance by learning from trajectories $\tau^{(i)}$ that: **(2.1)** fail to reach the target goal state $\mathbf{g}$ at rollout time, and/or **(2.2)** pass through avoid-regions $\mathbf{b}_j$ rather than successfully avoiding them.

**Property 3 (P3): Goal-conditioned generalization.** The model must generalize to any arbitrarily specified goal state $\mathbf{g}$ at evaluation time, without additional training. In real-world scenarios, such as autonomous navigation or therapeutic reprogramming, the target goal is often specified dynamically and cannot be hardcoded at training time.

**Property 4 (P4): Avoid-region-conditioned generalization.** The model must be able to learn a policy that can avoid any dynamically specified avoid-region(s) $\mathbf{b}_j$ of the state space at evaluation time, without additional training/finetuning. This includes supporting changes in: **(4.1)** the number of avoid-regions $n_{\text{avoid}}$, **(4.2)** their locations $\texttt{centroid}(\mathbf{b}_j)$, and **(4.3)** their sizes (i.e., spatial extent of the state space around each $\texttt{centroid}(\mathbf{b}_j)$ to avoid).

**Property 5 (P5): Reward-free learning.** The model must learn reach-avoid behavior without requiring a manually designed reward or cost function. Reward shaping is often brittle and requires expert domain knowledge (Freitag et al., 2024; Knox et al., 2023; Knox & MacGlashan, 2024; Abouelazm et al., 2024), especially when preferences over reaching and avoiding are difficult to encode or conflict. Instead, one should be possible to specify goals and avoid-regions directly as inputs to the model.

While many prior approaches address subsets of these properties, none satisfy all five simultaneously (Figure 1b). We discuss these limitations in detail in Section 3.

## 3 RELATED WORK

We review four key areas in reach-avoid learning: offline goal-conditioned RL (OGCRL), offline safe RL (OSRL), offline goal-conditioned safe RL (OGCSRL), and decision transformer (DT) models. Figure 1b summarizes which properties each class of methods satisfies. Additional discussion appears in Appendix E.

**Offline Goal-Conditioned RL.** OGCRL methods aim to learn policies that generalize to arbitrary goals specified at evaluation time, typically by conditioning on goal states and applying techniques such as hindsight goal relabeling (Andrychowicz et al., 2017). **Reward-based OGCRL** methods (Yang et al., 2023; Kostrikov et al., 2021; Ma et al., 2022) optimize a policy $\pi(\mathbf{a}|\mathbf{s}, \mathbf{g})$ to maximize a goal-conditioned reward function $r(\mathbf{s}, \mathbf{a}, \mathbf{s}', \mathbf{g})$. While this framework supports goal generalization (P3), it does not satisfy P5, as it requires designing reward functions that capture both goal-reaching and avoid desires, which are difficult to construct in practice (Knox et al., 2023; Knox & MacGlashan, 2024; Freitag et al., 2024). **Reward-free OGCRL** methods (Eysenbach et al., 2022; Park et al., 2023; Ghosh et al., 2019; Yang et al., 2022; Lynch et al., 2019; Janner et al., 2021) avoid reward functions by learning from hindsight-relabeled trajectories using supervised learning. These methods satisfy P5 but do not support avoid-region conditioning (P4), as they cannot incorporate constraints beyond the goal.

**Offline Safe RL.** OSRL methods (Zheng et al., 2024; Lee et al., 2022; Xu et al., 2022a; Le et al., 2019; Lin et al., 2024a) are designed for safety-critical tasks, learning policies that satisfy constraints specified via cost functions. A typical approach defines a cost function $\text{cost}(\mathbf{s}, \mathbf{a}, \mathbf{s}')$ and learns a policy that maximizes reward return $\sum_t r_t$ subject to a cost return constraint $\sum_t c_t < k$, often via Lagrangian relaxation (Stooke et al., 2020). These methods can enforce avoid behavior, but they are not generally goal-conditioned (P3), and do not support dynamic conditioning on varying avoid-region configurations (P4). In addition, these methods fail P5 due to their reliance on handcrafted reward and cost functions.

**Offline Goal-Conditioned Safe RL.** This hybrid category combines elements of OGCRL and OSRL and comes closest to satisfying the full set of reach-avoid properties. Representative methods include Recovery-based Supervised Learning (RbSL) and Actionable Models with Lagrangian Constraints (AM-Lag) (Cao et al., 2024; Chebotar et al., 2021; Stooke et al., 2020). These models construct an augmented state space $\mathcal{S}^+ \subseteq \mathbb{R}^{d_s + d_s + n_{\text{avoid}} \cdot d_s}$ containing the agent's state $\mathbf{s} \in \mathbb{R}^{d_s}$, goal state $\mathbf{g} \in \mathbb{R}^{d_s}$, and $n_{\text{avoid}}$ avoid centroids $\text{centroid}(\mathbf{b}_j) \in \mathbb{R}^{d_s}$, then learn policies with goal-conditioned and avoid-conditioned objectives. This design satisfies P3 and P4.2, enabling generalization to arbitrary $\mathbf{g}$ and avoid-region locations. However, these methods do not satisfy P4.1, as the number of avoid-regions $n_{\text{avoid}}$ is fixed in the state space dimension, requiring retraining to accommodate more regions. They also fail to support P4.3, as the spatial extent of avoid-regions is encoded only in the cost function, requiring redefinition and retraining when region size changes. Furthermore, although these models could in principle satisfy P2.2 (learning from trajectories that violate avoid-regions), the training data in Cao et al. (2024) is collected from environments with impassable obstacles, meaning no training trajectories actually pass through avoid-regions (Figure 3b), failing to demonstrate robustness to this type of data suboptimality. See Appendix C.1 for details.

**Decision Transformers.** DT models (Chen et al., 2021; Janner et al., 2021; Zheng et al., 2022; Wu et al., 2023; Wang & Zhou, 2024; Liu et al., 2023; Lin et al., 2023) represent a class of offline RL approaches that frame policy learning as sequence modeling. DTs use causal transformer architectures that take as input a trajectory $\tau$ and autoregressively predict actions, $\pi(\mathbf{a}_t|\tau_{1:t-1}, \mathbf{s}_t)$. Prompting has recently been introduced as a mechanism to extend DTs to goal-conditioned settings (Xu et al., 2022b; Yuan et al., 2024). We build our method off of MGPO (Yuan et al., 2024), which introduces prompt-based conditioning on arbitrary goals, enabling zero-shot generalization across goal states, but not avoid-regions.

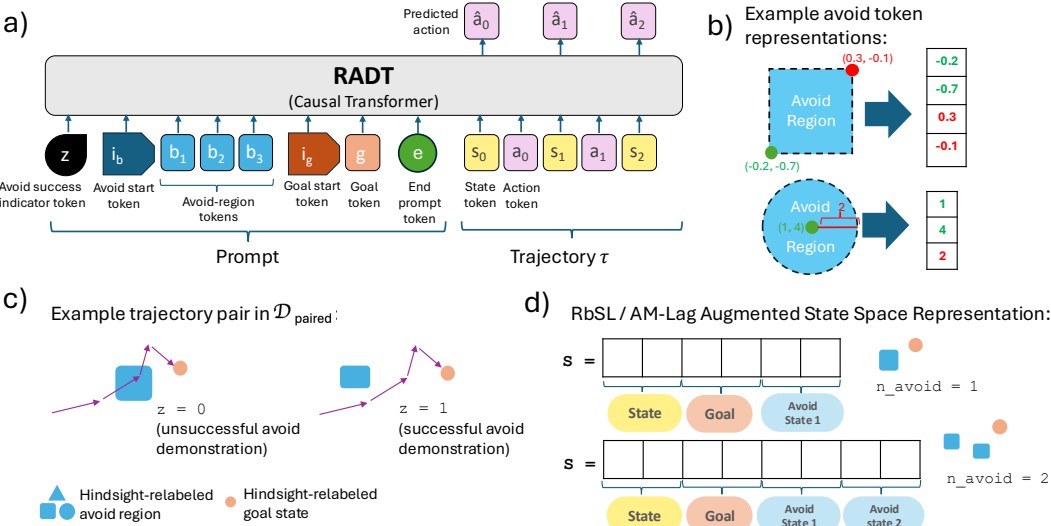

**Figure 2:** (a) RADT receives goal states and avoid-regions as prompt inputs. (b) avoid-regions are represented as avoid-region tokens in the prompt, which can be any vector representation that contains position and size information. The box and spherical representations shown here are examples of valid representations. (c) For each offline trajectory, we generate two versions: one that violates a sampled avoid-region and one that avoids it. Both are labeled with an avoid success token $z$. (d) Prior models encode avoid-regions via augmented state vectors, which grow with the number of avoid-regions, preventing zero-shot generalization to unseen avoid counts.

## 4 REACH-AVOID DECISION TRANSFORMER (RADT)

In this section, we describe the main components of our method, RADT (Figure 2). Similar to MGPO, our method is based on a causal Transformer architecture and utilizes prompts to specify goal states (satisfying P3). However, unlike MGPO, RADT additionally allows for the specification of avoid-regions in the prompt (satisfying P4) and does not require reward-driven online prompt optimization (satisfying P1, P2, P5).

### Prompt Tokens and Avoid-Region Representation

RADT takes in a prompt $p$ to be presented to the Transformer model before a trajectory $\tau$. RADT's autoregressive prediction of the next action during rollout is thus additionally conditioned on this prompt: $\pi(\mathbf{a}_t|p, \tau_{0:t-1}, \mathbf{s}_t)$. The prompt is structured as follows (Figure 2a):

$$p = (z, i_b, \mathbf{b}_1, \mathbf{b}_2, ..., \mathbf{b}_{n_{\text{avoid}}}, i_g, \mathbf{g}, e)$$

Note that there are *six different types* of tokens (units of input into a sequence model) present in the prompt:

① **The avoid success indicator (SI) token,** $z : z \in \{0, 1\}$, indicates whether the trajectory $\tau$ following the prompt successfully circumvents the avoid state: $z = 1$ if all states in $\tau$ exist outside of all avoid-boxes $\mathbf{b}_j : j \in \{1, 2, ..., n_{\text{avoid}}\}$ and $z = 0$ otherwise. This allows us to train the model on both trajectories that demonstrate successful and unsuccessful avoid behaviors; this is important for the model to explicitly learn what *not* to do (see Section 5, Results 4). During evaluation time, we will *always* condition on $z = 1$ to achieve optimal avoid behavior.

② **The avoid start (AS) token**, $i_b$, explicitly indicates to the model that the upcoming tokens represent undesirable avoid-regions. This is to clearly distinguish the avoid-region tokens from the SI token.

③ **The avoid-region tokens**, $\mathbf{b}_j \in \mathcal{S} \subseteq \mathbb{R}^{d_a} : j \in \{1, 2, ..., n_{\text{avoid}}\}$, represents the $n_{\text{avoid}}$ avoid-regions in the state space we would like the agent to circumvent. Our approach is not constrained to any particular vector representation of avoid-regions, as long as the representation contains information about both the position (the centroid) and the spatial extent of the avoid-region. Detailed overview of the representations we use can be found in Appendix A.2. Since prompts can consist of any arbitrary number of avoid-region tokens $\mathbf{b}_j$ and the avoid-region tokens can represent regions of any arbitrary size at evaluation time, this satisfies P4.1, P4.2, and P4.3.

④ **The goal start (GS) token**, $i_g$, explicitly indicates to the model that the next token to be provided represents a desirable goal token. This is to clearly distinguish the goal token in the prompt from the avoid-region tokens.

⑤ **The goal token**, $\mathbf{g} \in \mathcal{S} \subseteq \mathbb{R}^{d_s}$, represents the goal state we would like the agent to achieve. This can be set to any state in the state space at evaluation time (satisfying P3).

⑥ **The prompt end token**, $e$, explicitly marks the end of the prompt. This indicates to the model that the next token marks the beginning of the main input sequence, $\tau$.

The initial representations of tokens of different types have different dimensionalities, as described above, but they are all projected into a shared latent embedding space to acquire common dimensionality before being passed through the rest of the transformer architecture. See Appendix B.1 for details regarding how these different token types are embedded. Since we use prompts to specify the goal and avoid-region information, we do not need to work with an augmented state space $\mathcal{S}^+$, unlike RbSL and AM-Lag (see Section 3), providing us with greater zero-shot flexibility.

### Avoid-Region Relabeling and Training Data

We specifically consider the scenario in which the training dataset $\mathcal{D}$ contains training trajectories $\tau$ that are generated from a purely random policy (satisfying P1 and P2). For each training trajectory $\tau \in \mathcal{D}$, we relabel the last state $\mathbf{s}_T$ as the goal state $\mathbf{g}$ in hindsight. Additionally, we can also relabel *avoid-regions* in hindsight, a novel strategy that gets rid of the need to use a cost function to learn desirable avoid behavior. The intuition for hindsight avoid-region relabeling (HAR) is similar to goal relabeling: it does not matter whether the policy that collected the training trajectory was actually intending to circumvent the hindsight-relabeled avoid-region; we can still treat it as if the region was "meant" to be avoided, as the trajectory demonstrates how to *not* pass through that region.

The details to our HAR approach are described in Appendix A.3. The resulting output of HAR is a dataset $\mathcal{D}_{\text{paired}}$, which contains a *pair* of trajectories $(\tau_{\text{orig}}, \tau_{\text{copy}})$ for each $\tau$ in the original dataset $\mathcal{D}$, where one of $(\tau_{\text{orig}}, \tau_{\text{copy}})$ contains relabeled avoid-regions are successfully avoided (and is labeled with $z = 1$) and the other contains avoid-regions that are violated (and labeled with $z = 0$). The intent is to isolate the concept of avoid success vs. failure from differences in the trajectories themselves, allowing the model to more clearly learn what the avoid success token $z$ represents conceptually (Figure 2c; Section 5, Results 4). Using the prompt format described in the section above, we present these trajectories in $\mathcal{D}_{\text{paired}}$ to a causal transformer and train the model using a causal language modeling objective. Detailed model/training specifications and hyperparameters are included in Appendix B).

## 5 EXPERIMENTS

We evaluate RADT in three sets of reach-avoid environments: `FetchReachObstacle`, `MazeObstacle`, and `Cardiogenesis`. The first two are adapted from Gymnasium Robotics tasks (de Lazcano et al., 2024), but with added avoid-regions. In these environments, we compare RADT against two OGCSRL baselines (RbSL (Cao et al., 2024) and AM-Lag (Cao et al., 2024; Chebotar et al., 2021)) as well as Weighted Goal-Conditioned Supervised Learning (WGCSL) (Yang et al., 2022), a strong OGCRL method that does not explicitly account for avoid-regions. To evaluate domain generality, we also test RADT in a biological setting: cell reprogramming. Additionally, we perform a series of ablation experiments to validate the design decisions in our approach and a series of stress-testing experiments to evaluate how RADT reacts under more extreme out-of-distribution (OOD) avoid specifications, varying mixtures of random vs. expert data, and alternative representations for avoid-regions. Additional task details are provided in Appendix C.

**Results 1: Fetch Reach Environment and Generalization to Varying Avoid-Region Sizes**

**Environment.** The `FetchReachObstacle` environment requires a robotic arm to reach a goal location while avoiding a randomly positioned box, treated as the avoid-region under our formulation (Figure 3a). The positions of the end-effector, goal, and avoid-box are randomly sampled each episode. Unlike prior work such as Cao et al. (2024), the robot can pass through the avoid-box, allowing us to isolate avoid-region reasoning without hard state space constraints (Figure 3b).

**Data.** We collect $2 * 10^6$ timesteps of random-policy trajectories. Goal relabeling and HAR is then applied, using a contour-based sampling strategy to generate diverse box placements (Appendix A.3.2).

**Evaluation Metrics.** While RADT is trained without a cost function, we define one for evaluation: $\text{cost}(s_t, a_t, s_{t+1}) = 1$ if $s_{t+1}$ is inside an avoid-box and $0$ otherwise. We evaluate models using the mean normalized cost return (MNC), computed as the average per-step cost: $\text{MNC}(\tau) = \frac{1}{|\tau|} \sum_{(s,a,s') \in \tau} \text{cost}(s, a, s')$. We also report the goal-reaching success rate (SR), defined as the proportion of episodes that reach the goal. Evaluation is performed over 60 episodes.

**Setup.** We train all models using environments with a fixed avoid-box width (or, in the case of RADT, a max hindsight-relabeled box width) of 0.16. RbSL, AM-Lag, and WGCSL are trained using the above cost function. We then evaluate all models in environments in-distribution (ID) with the same 0.16 avoid-box width. To assess generalization, we evaluate RADT zero-shot on avoid-boxes with out-of-distribution (OOD) widths ranging from 0.16 to 0.24 (1.5× larger). In contrast, baseline models do not support zero-shot generalization to new avoid-box sizes and must be retrained on separate offline datasets generated for each new size (Sections 3 and 4, and Appendix C.1). This creates a disadvantageous setting for RADT, which is evaluated without retraining, while baselines are retrained for each test condition.

**Results.** Results are shown in Table 1. For the ID case (box width 0.16), RADT performs comparably to AM-Lag in MNC and better than RbSL and WGCSL. For SR, RADT outperforms AM-Lag and matches or exceeds the other baselines (Figure 3c). This confirms that both RADT and AM-Lag are competitive for reach-avoid learning, with RADT slightly favoring goal-reaching and AM-Lag slightly favoring cost minimization. In the OOD setting, where avoid-box sizes exceed those seen during training, RADT continues to perform strongly despite being evaluated zero-shot. Across all OOD widths, RADT matches or exceeds the MNC of retrained AM-Lag models and substantially outperforms retrained RbSL and WGCSL models. It also maintains comparable SR to the retrained baselines (Figure 3c). These results show that RADT generalizes to unseen avoid-region sizes, even outperforming methods that are *retrained* for each evaluation setting.

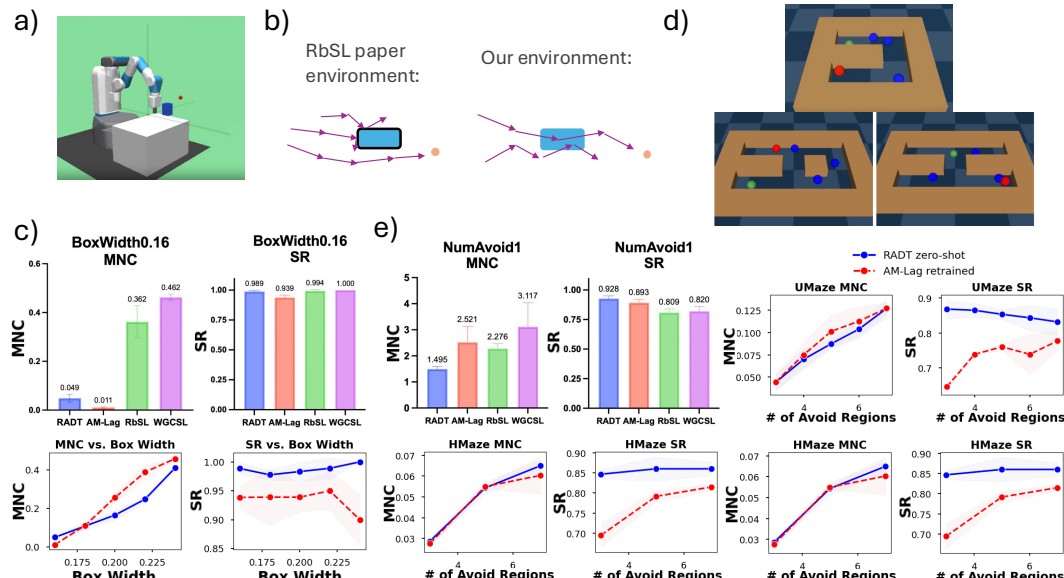

**Figure 3:** (a) Visualization of the `FetchReachObstacle` environment. The red point is the goal; the blue box is the avoid-region. (b) Unlike prior setups, the robot arm can pass through avoid-boxes, allowing training data to include violations so that we can test the models' abilities to learn from suboptimal data. (c) RADT and AM-Lag achieve state-of-the-art MNC and SR on in-distribution box sizes and generalizes zero-shot to OOD avoid-box sizes, matching or surpassing the best baseline (AM-Lag), which needs to be *retrained* on every new avoid-box size (i.e., not zero-shot). (d) Visualization of the `UMazeObstacle`, `AMazeObstacle`, and `HMazeObstacle` environments, with red goal, blue avoid-regions, and green agent. (e) RADT outperforms all baselines on MNC and SR in the in-distribution single-avoid setting for `UMazeObstacle` and generalizes zero-shot to OOD numbers of avoid-regions on all maze tasks, matching the best `retrained` baseline (AM-Lag) in MNC and surpassing it in SR. Error bars show ±1 standard deviation.

### Results 2: Maze Environments and Generalization to Varying Numbers of Avoid-Regions

**Environment.** The `MazeObstacle` environments require a point agent to navigate through a maze to reach a randomly sampled goal location while avoiding randomly placed circular obstacles. As in `FetchReachObstacle`, these obstacles are soft constraints that the agent may pass through, and we refer to them as "avoid-regions." Unlike in `FetchReachObstacle`, the maze itself imposes hard constraints, introducing impassable regions of the state space. This tests RADT's ability to reason over both user-specified avoid-regions and inherent environmental constraints. We evaluate RADT on three environments of varying difficulty: `UMazeObstacle`, `AMazeObstacle` (2 dead ends), and `HMazeObstacle` (4 dead ends) (Figure 3d). The Data/Evaluation pipelines are similar to those of `FetchReachObstacle`, with details in Appendix C.1.

**Setup.** We begin by training all models using data generated in environments with a single avoid-region, and evaluate in matching single-avoid-region settings (ID). To evaluate generalization, we then train and evaluate all models in environments with three avoid-regions, followed by testing in environments with 4-7 avoid-regions to assess zero-shot OOD performance. For each new number of avoid-regions, baseline models (RbSL, AM-Lag, WGCSL) are retrained with an appropriately expanded state space (Figure 2d; see Sections 3 and 4). In contrast, RADT is evaluated zero-shot without any retraining. This design intentionally favors the baselines, as they are tailored to each test setting, whereas RADT is held fixed across all configurations. Only the best-performing baseline in `UMazeObstacle` (AM-Lag) is evaluated on `AMazeObstacle` and `HMazeObstacle`.

**Results.** Results are shown in Table 2. Across all settings of `UMazeObstacle` (both ID and OOD), RADT achieves the lowest MNC, consistently outperforming all baselines. It also outperforms AM-Lag and RbSL on SR, and performs comparably or better than WGCSL. Importantly, for all evaluations with more than three avoid-regions, RADT is deployed zero-shot, while the baselines are retrained. Despite this disadvantage, RADT matches or exceeds their performance, demonstrating its ability to generalize to OOD numbers of avoid-regions without retraining.

In `AMazeObstacle` and `HMazeObstacle`, RADT outperforms AM-Lag in goal-reaching SR across all numbers for avoid-regions. For MNC, RADT outperforms or performs comparably to

**Table 1:** Results for `FetchReachObstacle` with varying box sizes (avg. over 3 seeds).

| Box Width | RADT# | | AM-Lag | | RbSL | | WGCSL | |
|---|---|---|---|---|---|---|---|---|
| | MNC | SR | MNC | SR | MNC | SR | MNC | SR |
| 0.16 | $0.049_{\pm0.016}$ | $0.989_{\pm0.01}$ | $0.011_{\pm0.003}$ | $0.95_{\pm0.029}$ | $0.362_{\pm0.066}$ | $0.994_{\pm0.01}$ | $0.462_{\pm0.0148}$ | $1.0_{\pm0.0}$ |
| 0.18 | $0.107_{\pm0.018}$ | $0.978_{\pm0.009}$ | $0.11_{\pm0.01}$ | $0.95_{\pm0.033}$ | $0.484_{\pm0.016}$ | $1.0_{\pm0}$ | $0.513_{\pm0.048}$ | $0.994_{\pm0.01}$ |
| 0.20 | $0.164_{\pm0.027}$ | $0.983_{\pm0.017}$ | $0.255_{\pm0.023}$ | $0.955_{\pm0.025}$ | $0.571_{\pm0.013}$ | $1.0_{\pm0}$ | $0.588_{\pm0.066}$ | $1.0_{\pm0.0}$ |
| 0.22 | $0.247_{\pm0.018}$ | $0.989_{\pm0.019}$ | $0.387_{\pm0.039}$ | $0.944_{\pm0.02}$ | $0.64_{\pm0.053}$ | $0.99_{\pm0.01}$ | $0.699_{\pm0.019}$ | $0.994_{\pm0.01}$ |
| 0.24 | $0.409_{\pm0.012}$ | $1.0_{\pm0.0}$ | $0.457_{\pm0.008}$ | $0.964_{\pm0.017}$ | $0.701_{\pm0.019}$ | $1.0_{\pm0}$ | $0.729_{\pm0.038}$ | $0.989_{\pm0.01}$ |

# = Model capable of zero-shot generalization. Results highlighted in blue are zero-shot results.

**Table 2:** Results for `MazeObstacle` with varying number of avoid states (avg. over 3 seeds).

| | UMazeObstacle | | | | | | | |
|---|---|---|---|---|---|---|---|---|
| # Avoid | RADT# | | AM-Lag | | RbSL | | WGCSL | |
| | MNC (1e-2) | SR | MNC (1e-2) | SR | MNC (1e-2) | SR | MNC (1e-2) | SR |
| 1 | $1.495_{\pm0.096}$ | $.928_{\pm0.022}$ | $2.521_{\pm0.613}$ | $0.893_{\pm0.029}$ | $2.276_{\pm0.193}$ | $0.809_{\pm0.031}$ | $3.117_{\pm0.922}$ | $0.82_{\pm0.041}$ |
| 3 | $4.455_{\pm0.895}$ | $0.868_{\pm0.028}$ | $4.47_{\pm0.94}$ | $0.645_{\pm0.01}$ | $5.857_{\pm0.754}$ | $0.175_{\pm0.054}$ | $6.92_{\pm0.404}$ | $0.842_{\pm0.033}$ |
| 4 | $7.006_{\pm1.156}$ | $0.865_{\pm0.02}$ | $7.511_{\pm1.342}$ | $0.738_{\pm0.006}$ | $7.648_{\pm0.357}$ | $0.768_{\pm0.043}$ | $9.62_{\pm1.701}$ | $0.852_{\pm0.043}$ |
| 5 | $8.75_{\pm0.531}$ | $0.853_{\pm0.015}$ | $10.14_{\pm1.9}$ | $0.76_{\pm0.01}$ | $9.622_{\pm0.531}$ | $0.053_{\pm0.012}$ | $10.28_{\pm0.503}$ | $0.807_{\pm0.033}$ |
| 6 | $10.38_{\pm1.015}$ | $0.843_{\pm0.038}$ | $11.278_{\pm1.629}$ | $0.738_{\pm0.058}$ | $11.35_{\pm1.604}$ | $0.755_{\pm0.065}$ | $12.364_{\pm1.414}$ | $0.9_{\pm0.017}$ |
| 7 | $12.7_{\pm1.0}$ | $0.832_{\pm0.038}$ | $12.72_{\pm0.702}$ | $0.777_{\pm0.028}$ | $24.17_{\pm3.65}$ | $0.002_{\pm0.003}$ | $14.48_{\pm1.439}$ | $0.825_{\pm0.018}$ |

| | AMazeObstacle | | | | HMazeObstacle | | | |
|---|---|---|---|---|---|---|---|---|
| # Avoid | RADT# | | AM-Lag | | RADT# | | AM-Lag | |
| | MNC (1e-2) | SR | MNC (1e-2) | SR | MNC (1e-2) | SR | MNC (1e-2) | SR |
| 3 | $3.1_{\pm0.361}$ | $0.926_{\pm0.006}$ | $4.467_{\pm0.252}$ | $0.805_{\pm0.02}$ | $2.867_{\pm0.153}$ | $0.847_{\pm0.012}$ | $2.767_{\pm0.208}$ | $0.695_{\pm0.03}$ |
| 5 | $4.933_{\pm0.252}$ | $0.908_{\pm0.024}$ | $5.473_{\pm1.0}$ | $0.808_{\pm0.056}$ | $5.433_{\pm0.306}$ | $0.86_{\pm0.03}$ | $5.5_{\pm0.1}$ | $0.792_{\pm0.014}$ |
| 7 | $7.0_{\pm0.127}$ | $0.922_{\pm0.013}$ | $6.933_{\pm1.415}$ | $0.86_{\pm0.022}$ | $6.5_{\pm0.265}$ | $0.86_{\pm0.017}$ | $6.03_{\pm0.907}$ | $0.815_{\pm0.009}$ |

AM-Lag, with RADT's lead decreasing as the number of avoid-regions becomes more OOD for RADT but remains ID for AM-Lag (5 and 7).

**Results 3: Applications in Biology: Zero-Shot Avoidance in Stochastic Cell Reprogramming**

We apply RADT to a biomedical problem: cell reprogramming. The goal is to transition a cell from one gene expression state to another using sequences of genetic perturbations. This technique underpins regenerative medicine (Wan & Ding, 2023; Vasan et al., 2021), stem cell therapy (Takahashi & Yamanaka, 2006; Guan et al., 2022), and anti-aging strategies (Paine et al., 2024; Pereira et al., 2024). However, intermediate gene expression states encountered during reprogramming may carry risks, e.g. tumorigenesis (Wuputra et al., 2020; Lin et al., 2024b). Safe reprogramming fits the reach-avoid formulation: reach a target state while avoiding undesirable intermediate states (Figure 4a).

**Environment.** We introduce `Cardiogenesis`, an environment based on a well-established Boolean network model of gene expression dynamics during mouse cardiogenesis (Herrmann et al., 2012; Singh, 2024). The model is described in detail in Appendix C.2. The key takeaway is that, unlike the Gymnasium Robotics environments, this domain features: (1) a fully discrete and combinatorial state-action space, (2) high-dimensional interdependencies between state variables due to Boolean logic, and (3) stochastic transitions. See Appendix C.2 for full environment specifications.

**Data.** We generate $6 * 10^4$ random-policy timesteps of training data. Our HAR approach (Section 4) is directly applicable to this discrete setting with minimal modification (Appendix C.2).

**Setup.** We conduct a case study where the reprogramming goal is to reach the first heart field (FHF) state, a critical attractor in cardiac development (Herrmann et al., 2012; Singh, 2024). We train RADT on the offline data, then evaluate its ability to reach the FHF state starting from a distinct attractor state (Appendix C.2). As first pass, we run 200 evaluation episodes with no avoid-region specified. From these, we identify the most frequently visited intermediate state as measured by Visitation Rate$(s) = \{$# of trajectories that visit $s$ at some point$\}/\{$# of total trajectories$\}$. We then run a second set of 200 episodes, this time providing the most visited intermediate state as an *avoid-region token* in the prompt. We evaluate whether RADT is able to discover alternative reprogramming paths that avoid this state and compare both visitation rates and trajectory lengths. This setup is illustrated in Figure 4b. See Appendix C.2 for details.

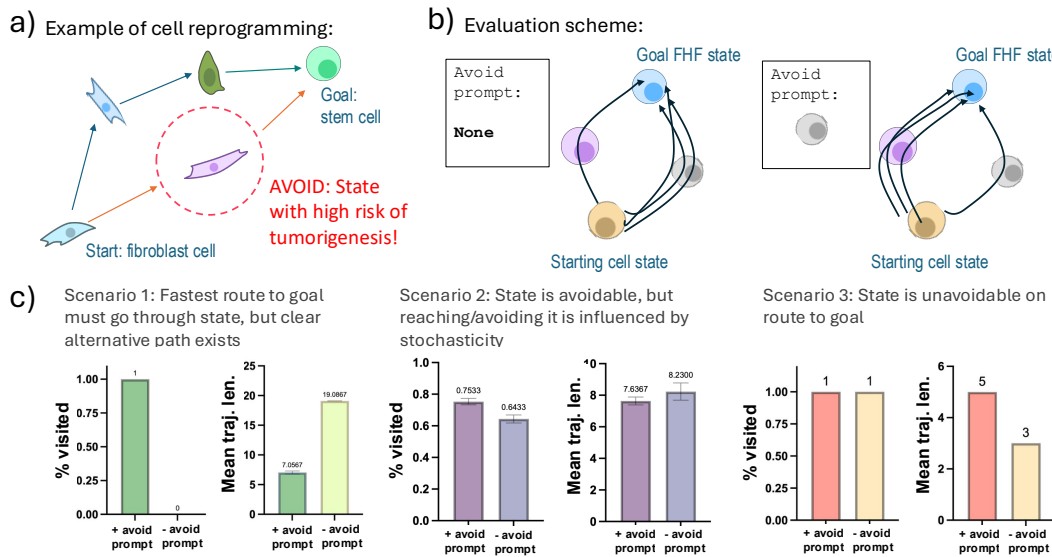

**Figure 4:** (a) Cell reprogramming involves sequential perturbations to reach a target gene expression state while avoiding undesirable intermediate states. (b) Evaluation pipeline: RADT is first run without an avoid-region token. The most frequently visited intermediate state is then added as an avoid-region token, and RADT is re-evaluated. (c) RADT reduces visitation rate to specified avoid states and minimizes time spent in unavoidable avoid states. Error bars show ±1 standard deviation. Some illustrations adapted from NIAID NIH BIOART (Appendix I).

**Results.** All evaluation trajectories successfully reach the FHF goal state, so we report only intermediate state visitation and trajectory lengths. Figure 4c summarizes results. We observe three distinct behavioral patterns, depending on the initial state. **Scenario 1.** The most visited intermediate (state A) is encountered in every trajectory when no avoid-region token is used. When A is included as an avoid-region token, RADT avoids it entirely and instead follows alternative, longer trajectories that always bypass A. **Scenario 2.** The most visited intermediate (state B) is frequently visited when no avoid-region token is used, but it is not visited by all trajectories. This suggests that B is not essential to reach the goal, but has a high chance of being landed on due to stochastic dynamics. When provided as an avoid-region token, RADT reduces visitation rate to state B in this noisy setting. **Scenario 3.** The most visited state (state C) is present in all trajectories regardless of prompt. When C is added as an avoid-region token, RADT cannot avoid passing through it, indicating that it is structurally unavoidable from the given initial state. However, RADT reduces the number of steps spent in C, shortening the portion of the trajectory that includes it. This suggests that even when avoidance is infeasible, the model learns to minimize time spent in unsafe states. These results highlight that RADT supports reach-avoid planning in discrete, structured, and stochastic domains, and also exhibits flexible avoidance strategies, including temporal minimization of contact with undesirable states when full avoidance is not possible. Full results and trajectory visualizations are in Appendix C.2.

### Results 4: Ablation and Additional Stress-testing Experiments

We conduct ablation studies to highlight the importance of the SI, AS, and GS tokens in the prompt, as well as the usage of attention boosting to the prompt. We also perform additional stress-testing experiments to examine the extremities of RADT's OOD generalization, the empirical performance ceiling induced by the usage of random-policy data, and the RADT's robustness to changing avoid-region token representations. We summarize the results here, and leave the detailed experimental setup, results, and discussion in Appendix G.1,G.2,G.3,G.4,G.5 and Tables 6,7,8,9,10,11,12 in Appendix H.

**Ablation of the SI token.** Ablating the SI token results in drastic performance drops as measure by both MNC and SR, demonstrating the importance of including failure examples and the SI token to teach the model what is *undesirable* (Table 6).

**Ablation of the AS/GS tokens.** Ablating the AS/GS tokens results in worse MNC in all environments and worse SR in some environments. The performance drops are not as drastic as those induced in the SI ablation experiment, suggesting that the AS/GS tokens are less critical than the SI token, but still provide noticeable benefit (Table 7).

**Ablation of attention boosting.** We find that adding a bias `adelta` (a hyperparameter) to the attention logits to all prompt tokens, using the strategy described in Silva et al. (2024), noticeably improves the instruction-following ability of RADT to the prompt (Figure 6).

**Ablation of avoid-region relabeling.** The ablation of avoid-region relabeling significantly impairs RADT's MNC performance across both ID and OOD box sizes, demonstrating the critical nature of avoid-region relabeling (Table 11).

**Extremities of OOD generalization.** To test the extremities RADT's zero-shot generalization, we test the zero-shot performance of RADT on `UMazeObstacle` environments with up to 20 avoid-regions. We find that zero-shot RADT is still able to maintain comparable or superior MNC to the best *retrained* baseline (AM-Lag) at 10 avoid-regions (>300% of the maximum seen during training by RADT), demonstrating strong OOD generalization (Table 8). Notably, zero-shot RADT maintains a superior SR to retrained AM-Lag all the way up to and including 20 avoid-regions. Additionally, we conduct an analysis on the effects of long prompts with large numbers of avoid-region tokens in Appendix G.2 "Effects on attention mechanism" and Fig. 11.

**Effects of introducing expert data.** To explore the performance ceiling induced by using purely random-policy data, we examine the effects of introducing varying amounts of expert-policy trajectories into the training data. We observe that the introduction of increasing amounts of expert-policy data does not significantly impact performance on ID box sizes, but does improve avoid ability (MNC) while *impairing* goal-reaching SR on OOD box sizes (Table 9). This suggests the particular expert trajectories used biases RADT to be more conservative in OOD avoid configurations.

**Effects of changing avoid-region representation.** We test RADT's flexibility with regards to handling various avoid-region token representations. We demonstrate that using an alternative representation for avoid-region tokens to the default box representation does not prevent RADT from learning strong reach-avoid policies, establishing that our approach is agnostic to avoid-region token representation (Table 10). Additionally, we discuss in Appendix G.4 "Complex avoid-region shapes" and Fig. 10 how we have implicitly demonstrated RADT handling complex avoid-region shapes due to the composition of many overlapping simple avoid-regions in maze environments.

**Image-based observation spaces and avoid-region representations.** To explore RADT's potential in handling 1) high-dimensional observation spaces and 2) unstructured, flexible avoid-region token representations that can capture abstract avoidance desires without knowledge of a pre-defined vectorization, we conduct a preliminary proof-of-concept demonstration of RADT applied to an image-based version of `UMazeObstcale` that we call `UMazeObstacleImage` (Fig. 12). In this environment, we present states, avoid-region tokens, and goal tokens as images encoded using a ResNet-18 model (He et al., 2016; Marcel & Rodriguez, 2010). RADT achieves superior MNC and comparable or superior goal-reaching SR to baselines in this environment, demonstrating its potential in being adapted to both high-dimensional tasks and flexible avoid-region representations that do not require precise knowledge of a strict, vectorized structure to specify avoid-regions (Table 12). Experiment setup is described in Appendix G.5.

## 6 CONCLUSION

We introduce RADT, a prompting-based offline reinforcement learning model for reach-avoid tasks that satisfies all key desiderata for flexible reach-avoid learning. RADT is trained entirely on suboptimal data and generalizes zero-shot to unseen avoid-region configurations, achieving competitive or superior performance to state-of-the-art methods retrained for each evaluation setting. RADT is domain-agnostic, demonstrating strong results in both robotics and cell reprogramming. We attribute RADT 's versatility to its interpretable prompt-based design and fully data-driven learning procedure that does not rely on reward/cost shaping. These properties enable RADT to serve as a general-purpose framework for safe sequential decision-making under dynamically shifting constraints. Limitations of our model are described in Appendix F.

## ETHICS STATEMENT

Foundational methods for learning reach-avoid policies, like RADT, have strong potential in improving the performance of technologies in many application domains, such as general robotics, autonomous vehicles, and bioengineering. While we believe the advancement of these downstream domains can greatly benefit society, we do acknowledge that such technologies can also be used maliciously. RADT and any derivatives should never be used to create autonomous agents with harmful goals (e.g., self-navigating vehicles that maliciously target individuals) or to induce harmful biological states (e.g., programming cells to pathological cell states). RADT is designed with the intent of enabling the development of *safer* sequential decision making agents that can improve the convenience and health of individuals in our society.

## REPRODUCIBILITY STATEMENT

Hyperparameters and other technical specifications (e.g. RL environment specifications) are detailed in Appendices B and C. An anonymized version of the code repository is submitted as part of Supplementary Materials and is linked in Appendix D, with relevant commands to run key experiments in the README file in the repository.

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

APPENDIX

# A  DATA PREPARATION

## A.1  DATASETS

The download link for the preprocessed datasets, as well as the setup and preprocessing code used to generate these datasets, will be released with the code repository for the project. See Appendix D.

## A.2  AVOID REPRESENTATION DETAILS

avoid-regions are represented in our experiments as *box coordinates* (Fig 2b) by default, though any vector representation could work. The box coordinate vector of an avoid-region $\mathbf{b}_j \in \mathbb{R}^{2*d_s}$ represents a "box" in the state space to be avoided. It is defined such that the first $d_s$ entries represent the lower bounds of each of the state space dimensions (the "lower left corner" of the box in a 3D analogy) and the second $d_s$ entries represent the upper bounds of each of the state space dimensions (the "upper right corner"):

$$\mathbf{b}_j = [\underbrace{l_1, l_2, ..., l_{d_s}}_{\text{lower bounds}}, \overbrace{u_1, u_2 ..., u_{d_s}}^{\text{upper bounds}}]$$

The policy should avoid guiding the agent into the region of the state space bounded by this box. To see how we adapt this box representation for non-rectangular avoid-regions in the `MazeObstacle` (circular avoid-regions) and `Cardiogenesis` environments (discrete avoid states), refer to the "Avoid-Region Relabeling" sections of Appendix C.1 and Appendix C.2, respectively.

In addition to the box representation, we also conduct experiments using the following representation for spherical avoid-regions to demonstrate RADT's flexibility in handling different avoid-region token representations:

$$\mathbf{b}_j = [b_1, b_2, ..., b_{d_s}, r]$$

where $(b_1, b_2, ..., b_{d_s})$ represents the location of $\mathtt{centroid}(\mathbf{b}_j)$ in the state space and $r$ represents the radius of the avoid-region (Figure 2b). Results for the experiment are found in Appendix G.4.

## A.3  AVOID RELABELING DETAILS

### A.3.1  AVOID RELABELING TWO-PASS ALGORITHM

The two-pass hindsight avoid-region relabeling method introduced in Section 4 is described in more detail here and visualized in Algorithm 1 (first pass) and Algorithm 2 (second pass).

For each $\tau^{(i)} \in \mathcal{D}$, we carry out HAR in two passes. In the first pass, we randomly sample avoid-boxes $\mathbf{b}_j : j \in \{1, 2, ..., n_{\text{avoid}}\}$ of random sizes in the state space $\mathcal{S}$ and check whether any $s_t \in \tau^{(i)}$ violate any $\mathbf{b}_j : j \in \{1, 2, ..., n_{\text{avoid}}\}$. If there are no violations, then the SI for $\tau^{(i)}$ is set to $z^{(i)} = 1$, otherwise it is set to $z^{(i)} = 0$. In the second pass, we create a copy of the dataset, $\mathcal{D}_{\text{copy}}$, and go through the same process above with the trajectories $\tau_{\text{copy}}{}^{(i)} \in \mathcal{D}_{\text{copy}}$. This time, however, for a trajectory $\tau_{\text{copy}}{}^{(i)}$, we keep re-sampling avoid-boxes until the avoid success token $z_{\text{copy}}{}^{(i)}$ is the *opposite* of $z^{(i)}$ for the corresponding $\tau^{(i)}$ in the original $\mathcal{D}$.

Combining the datasets $\mathcal{D}$ and $\mathcal{D}_{\text{copy}}$ into $\mathcal{D}_{\text{paired}}$, we now have a *pair* of trajectories $(\tau_{\text{orig}}^{(i)}, \tau_{\text{copy}}^{(i)})$ for each $\tau^{(i)}$ in the original dataset, where one of $(\tau_{\text{orig}}^{(i)}, \tau_{\text{copy}}^{(i)})$ demonstrates successful avoid behavior and the other demonstrates unsuccessful avoid behavior.

### A.3.2  MORE SOPHISTICATED AVOID CENTROID SAMPLING STRATEGIES

The sampling of avoid centroids in Line 4 of Algorithm 1 can be done using naive uniform sampling over the state space (the default) or more sophisticated methods.

**Contour-Based Sampling.** One such sophisticated method we use samples avoid centroids that fit into the nooks of the broad contour of a training trajectory $\tau$ (Figure 5a). This is the sampling method

used for the `FetchReachObstacle` environment (a very open environment with no inherent restricted areas of the state space). We can acquire an outline of the general *contour* of a $\tau$ by calculating a concave hull of the data points $\mathbf{s}_t \in \tau$ in the state space $\mathcal{S}$ using the algorithm presented in Park & Oh (2013) as implemented in the `concave_hull` library (Tang, 2022). We denote the set of states in $\tau$ that belong to the convex hull as $\mathcal{S}_{\text{convex}}$. We can then find "nooks" (concave portions) of the trajectory by calculating the convex hull using the algorithm presented in Graham (1972) as implemented the `concave_hull` library, and then find the points that are part of the concave hull but not the convex hull; these points outline the concave nooks of the contour and we denote the set of these states as $\mathcal{S}_{\text{nook}}$. For each nook in trajectory $\tau$, we then find the two convex hull points in $\mathcal{S}_{\text{convex}}$ bordering the set $\mathcal{S}_{\text{nook}}$ and sample a point between them as the avoid centroid. As a result, the trajectory of states *appears* to be "attempting" to avoid the hindsight-relabeled avoid centroid by wrapping around it (Figure 5a). Although the training trajectory was not trying to avoid this state, this does not matter because the resulting hindsight-relabeled trajectory does demonstrate how to circumvent an avoid state by wrapping around it. This sampling method is found to improve performance on the `FetchReachObstacle` environment (Figure 5b).

**Sampling From a Limited Portion of the State Space.** In certain state spaces, there may be regions where sampling an avoid centroid there would not provide helpful information. Thus, instead of sampling from the entire state space, we can just sample from just the portion of the state space $\mathcal{S}_{\text{limited}} \subset \mathcal{S}$ instead. This is done in the `MazeObstacle` and `Cariogenesis` environments, see Appendices C.1 and C.2 for details.

## B  MODEL/TRAINING DETAILS

### B.1  MODEL ARCHITECTURAL DETAILS AND HYPERPARAMETERS

**Representations and Embeddings for Prompt Tokens.** All prompt tokens, regardless of type, are embedded into the same latent space as the action ($\mathbf{a}_t$) and state ($\mathbf{s}_t$) tokens in the main trajectory sequence: $\mathbb{R}^{d_h}$, where $d_h$ is the hidden dimension/embedding dimension (corresponding to hyperparameter `embed_dim` below). **The avoid success indicator token** $z$ is initially presented to the model as a one-hot vector: $\left[\begin{smallmatrix}1\\0\end{smallmatrix}\right]$ if $z = 1$ and $\left[\begin{smallmatrix}0\\1\end{smallmatrix}\right]$ if $z = 0$. This one-hot representation is then embedded into $\mathbb{R}^{d_h}$ via the learnable embedding matrix $E_z \in \mathbb{R}^{2 \times d_h}$. **The avoid start token**, $i_b$, and **the goal start token**, $i_g$, are initially presented to the model as the one-hot vectors $\left[\begin{smallmatrix}1\\0\end{smallmatrix}\right]$ and $\left[\begin{smallmatrix}0\\1\end{smallmatrix}\right]$, respectively. They are then embedded into $\mathbb{R}^{d_h}$ via the learnable embedding matrix $E_i \in \mathbb{R}^{2 \times d_h}$. The **prompt end start token**, e, is embedded into $\mathbb{R}^{d_h}$ as learnable vector $\mathbf{e} \in \mathbb{R}^{d_h}$. **Avoid-region tokens** $\mathbf{b}_j \in \mathbb{R}^{2d_s}$ and the **goal token** $\mathbf{g} \in \mathbb{R}^{d_s}$ are embedded into $\mathbb{R}^{d_h}$ via learnable embedding matrix $E_b \in \mathbb{R}^{2d_s \times d_h}$ and learnable embedding matrix $E_g \in \mathbb{R}^{d_s \times d_h}$, respectively.

---

**Algorithm 1** Hindsight Avoid-Region Relabeling: Initial Pass

---

**Require:** state space $\mathcal{S}$, offline training dataset $\mathcal{D}$, maximum avoid-box width $w_{\max}$, number of avoid states $n_{\text{avoid}}$
1: **for** training trajectory $\tau^{(i)}$ in $\mathcal{D}$ **do**
2:     Initialize avoid-regions list b_list$^{(i)} \leftarrow []$
3:     **for** $j$ in 1, 2, ..., $n_{\text{avoid}}$ **do**
4:         Randomly initialize avoid centroid $\mathbf{x}_j \in \mathcal{S}$
5:         Randomly choose avoid-box width $w \in [0, w_{\max}]$
6:         avoid-box $\mathbf{b}_j \leftarrow$ concatenate($\mathbf{x}_j - \frac{w}{2}, \mathbf{x}_j + \frac{w}{2}$)   ▷ box of width $w$ centered around $\mathbf{x}_j$
7:         Append $\mathbf{b}_j$ to b_list$^{(i)}$
8:     **end for**
9:     **if** All none of the states in $\tau^{(i)}$ are in any of the avoid-boxes in b_list$^{(i)}$ **then**
10:         Avoid success $z^{(i)} \leftarrow 1$
11:     **else**
12:         Avoid success $z^{(i)} \leftarrow 0$
13:     **end if**
14:     Add $z^{(i)}$ and b_list$^{(i)}$ to the corresponding training prompt $p^{(i)}$ for $\tau^{(i)}$
15: **end for**

---

---

**Algorithm 2** Hindsight Avoid-Region Relabeling: Second Pass

---

**Require:** state space $\mathcal{S}$, copy of the original offline training dataset $\mathcal{D}_{\text{copy}}$, maximum avoid-box width $w_{\max}$, number of avoid states $n_{\text{avoid}}$
1: **for** training trajectory $\tau_{\text{copy}}{}^{(i)}$ in $\mathcal{D}_{\text{copy}}$ **do**
2:     $z_{\text{copy}}{}^{(i)} \leftarrow z^{(i)}$
3:     **while** $z_{\text{copy}}{}^{(i)} = z^{(i)}$ **do**
4:         Initialize avoid-regions list b_list$^{(i)} \leftarrow$ []
5:         **for** $j$ in 1, 2, ..., $n_{\text{avoid}}$ **do**
6:             Randomly initialize avoid centroid $\mathbf{x}_j \in \mathcal{S}$
7:             Randomly choose avoid-box width $w \in [0, w_{\max}]$
8:             avoid-box $\mathbf{b}_j \leftarrow$ concatenate($\mathbf{x}_j - \frac{w}{2}, \mathbf{x}_j + \frac{w}{2}$) ▷ box of width $w$ centered around $\mathbf{x}_j$
9:             Append $\mathbf{b}_j$ to b_list$^{(i)}$
10:         **end for**
11:         **if** All none of the states in $\tau^{(i)}$ are in any of the avoid-boxes in b_list$^{(i)}$ **then**
12:             Avoid success $z_{\text{copy}}{}^{(i)} \leftarrow 1$
13:         **else**
14:             Avoid success $z_{\text{copy}}{}^{(i)} \leftarrow 0$
15:         **end if**
16:     **end while**
17:     Add $z_{\text{copy}}{}^{(i)}$ and b_list$^{(i)}$ to the corresponding training prompt $p_{\text{copy}}{}^{(i)}$ for $\tau_{\text{copy}}{}^{(i)}$
18: **end for**

---

**Attention Boosting.** We find that adding a bias `adelta` (a hyperparameter) to the attention logits to all prompt tokens, using the strategy described in Silva et al. (2024), noticeably improves the instruction-following ability of RADT to the prompt—i.e., improves goal-reaching behavior without negatively impacting avoid behavior (Figure 6).

**Model Hyperparameters.** The base causal transformer architecture we use is based on the HuggingFace implementation (Wolf et al., 2020) of the GPT-2 architecture (Radford et al., 2019); all model details that are not explicitly specified here are set to the default values for the HuggingFace implementation of GPT-2. We perform hyperparameter optimization (HPO) with a simple random search algorithm, using the RayTune framework (Liaw et al., 2018). The tunable model hyperparameters and their respective set of possible values in the search space are described in Table 3. Training hyperparameters are described in the next section.

**Table 3:** Model hyperparameters and search space

| Hyperparameter | Description | Search space |
|---|---|---|
| n_head | Number of attention heads | tune.choice([1, 3, 4, 6, 12]) |
| n_layer | Number of self-attention layers | tune.choice([3, 6, 12]) |
| embed_dim | Size of latent embedding space (hidden dimension) | Fixed to be 64 * n_head |
| adelta | Attention boosting bias to the prompt | tune.choice([0, 1, 2, 3]) |

The model hyperparameter configurations used are `{n_head=4, n_layer=4, embed_dim=256, adelta=2}` for the `FetchReachObstacle` environment, `{n_head=6, n_layer=6, embed_dim=384, adelta=1}` for the `MazeObstacle` environment, and `{n_head=6, n_layer=6, embed_dim=384, adelta=1}` for the `Cardiogenesis` environment.

### B.2  TRAINING DETAILS

**Loss Function.** For a batch of size $B$ of trajectories of length $T$, we use the following two-component loss function during training. The first component is the typical loss used in decision transformer models, the action loss $\mathcal{L}_{\text{action}}$. This is the mean squared error (MSE) between the predictions of next actions $\hat{\mathbf{a_t}}$ based on the last-layer embeddings of $(p, \tau_{1:t-1}, \mathbf{s}_t)$ and the actual next actions $\mathbf{a_t}$:

$$\mathcal{L}_{\text{action}} = \frac{1}{B} \sum_{i=1,2,...,B} \frac{1}{T} \sum_{t=1,2,...,T} (\widehat{\mathbf{a}_t^{(i)}} - \mathbf{a}_t^{(i)})^2$$

The second component is used to encourage RADT to learn how to be aware, at each timestep, whether or not the current state $\mathbf{s_t}$ violates any avoid-box $\mathbf{b}_j : j \in \{1, 2, ..., n_{\text{avoid}}\}$ in prompt $p$. We define the indicator $k_t$ to indicate whether or not $\mathbf{s_t}$ violates any avoid-boxes $\mathbf{b}_j$ in the prompt, with $k_t = 1$ indicating that $\mathbf{s_t}$ does not violate any avoid-boxes and $k_t = 0$ indicating $\mathbf{s_t}$ violates at least one avoid-box. During training *only*, in addition to predicting the next action $\mathbf{a_t}$, we also make RADT predict $k_t$ using the last-layer embeddings of $(p, \tau_{1:t-1}, \mathbf{s}_t)$. We define the box awareness loss $\mathcal{L}_{\text{avoid\_awareness}}$ to be the binary cross entropy (BCE) loss between predicted $\hat{k}_t$ and actual $k_t$.

$$\mathcal{L}_{\text{avoid\_awareness}} = \frac{1}{B} \sum_{i=1,2,...,B} \frac{1}{T} \sum_{t=1,2,...,T} k_t^{(i)} \log(\widehat{k_t^{(i)}}) + (1 - k_t^{(i)}) \log(1 - \widehat{k_t^{(i)}}))$$

The combined loss function is then:

$$\mathcal{L} = \mathcal{L}_{\text{action}} + \alpha \mathcal{L}_{\text{avoid\_awareness}}$$

$\alpha$ is a tunable hyperparameter to balance the different components of the loss function. During hyperparameter optimization, we try values of $\alpha$ ranging from 0.1 to 10 (Fig. 9). Very large values of $\alpha$ (e.g., 5 and 10) de-prioritize $\mathcal{L}_{\text{action}}$ too much and significantly impairs learning a policy with strong goal-reaching (SR) and avoidance (MNC) behavior. While very small values of $\alpha$ (e.g., 0.1) results in the fastest convergence to a policy with strong SR and MNC performance, intermediate $\alpha$ values ($\alpha = 1$) allows the model to eventually converge on a policy with the lowest MNC, with equally good SR. Thus, we set it to 1.0 in all our experiments.

**Stopping Criteria.** We let all RADT models train for 50,000 training steps (no visual improvement in SR or MNC beyond that), checkpointing every 500 steps. As we use a similar training scheme as the one used in the original Prompt DT (Xu et al., 2022b) and MGPO (Yuan et al., 2024) papers, where we sample batches of trajectories with replacement, we cannot use "epoch" as a measurement of training progress. At every checkpointing iteration, we run an evaluation on the model in the same way we did in our experiments (described in Section 5). We choose the checkpoint with the best (lowest) MNC whose SR is within 0.05 from the checkpoint with the highest SR (to ensure we have a model that is not achieving seemingly good avoid ability by significantly sacrificing goal-reaching ability, e.g., not moving). We do the same process for all baselines, except we train the baseline for 500 epochs as is done in the RbSL paper (Cao et al., 2024) (no visual improvement in SR or MNC beyond that).

**Training Hyperparameters.** Training hyperparameters are listed with their respective search spaces in Table 4. The scheduler value `lambdalr` corresponds to the PyTorch scheduler `torch.optim.lr_scheduler.LambdaLR` and the scheduler value `cosinewarmrestarts` corresponds to the scheduler `CosineAnnealingWarmupRestarts` from the `cosine_annealing_warmup` library (Katsura & Baldassarre, 2021) library is used to tune training hyperparameters. Any hyperparameters not explicitly listed default to the values used in MGPO (Yuan et al., 2024).

The chosen training hyperparameter configurations are {batch_size=128, learning_rate=1e-4, scheduler='cosinewarmrestartsq, T_0=1000, warmup_steps=500, alpha1=1} for the FetchReachObstacle environment, {batch_size=32, learning_rate=1e-4, scheduler='lamdalr', warmup_steps=1000, alpha1=1} for the MazeObstacle environment, and {batch_size=128, learning_rate=1e-4, scheduler='lamdalr', warmup_steps=1000, alpha1=1} for the Cardiogenesis environment.

**Table 4:** Training hyperparameters and search space

| Hyperparameter | Description | Search space |
|---|---|---|
| batch_size | Batch size | tune.choice([32, 64, 128, 256]) |
| learning_rate | Maximum learning rate | tune.choice([1e-4, 1e-5]) |
| scheduler | Learning rate scheduler | tune.choice(['cosinewarmrestartsq, 'lambdalrq]) |
| warmup_steps | Number of warmup steps | tune.choice([500, 1000]) |
| T_0 | T_0 parameter to CosineAnnealingWarmupRestarts | Fixed to be 1000 |
| T_mult | T_mult parameter to CosineAnnealingWarmupRestarts | Fixed to be 1 |
| weight_decay | $\lambda$ constant used in weight decay | Fixed to be 1e-4 |
| dropout | Dropout ratio during training | Fixed to be 0.1 |
| alpha1 | The $\alpha$ weight balancing the two-component loss function | tune.loguniform([0.1, 10]) |

**Compute Resources.** All training sessions of RADT models were done using a single H100 GPU. On average, RADT takes around 10 hours (i.e., 10 GPU hours) to converge on a strong policy on

Gymnasium Robotics tasks (depending on the task) in terms of pure *training* time. The *total* amount of time depends on how often we perform validation/evaluation (e.g., performing 100 interactive episodes with the environment during each evaluation session can take quite a bit of time). The maximum memory utilization during training hit 40 GB. We acknowledge this is a higher computation overhead compared to baselines, which can converge on a strong policy using 1 GPU within 2 hours of pure training time, with the maximum memory utilization during training hitting 20-30 GB. However, given that all training was done with a single GPU and a strong policy can be achieved within a day, we do not think RADT is prohibitively expensive to train. Additionally, RADT has the additional benefit of being able to generalize zero-shot to novel avoid-region configurations dynamically specified at test time while other methods do not, removing the need for compute-heavy retraining in many scenarios where baselines would have to be retrained.

## C    Experiment Details

### C.1    Gymnasium Robotics Environments

**Choice of Environments.**  While there are more complicated environments that are part of the Fetch and Maze suites in Gymnasium robotics, we do not evaluate on those, since the focus of those environments is testing for proficiency in long-range planning, skills learning, and hierarchical learning, which are not the focus of the current iteration of RADT. Additionally, we are focusing only on the domain where the training data is 100% generated from a random policy (Criterion 1.2), and it is difficult for RL approaches across the board to just learn good goal-reaching performance on these more complex environments under this data regime (Park et al., 2025), let alone good avoid behavior. With low goal-reaching success rates, our evaluation of avoid behavior will not be very meaningful. As an extreme example, a policy that makes the agent stay in place will achieve an MNC of 0, but its SR will also be 0; this would not be considered good avoid behavior despite the low MNC.

**Passable and Impassable Avoid-Regions.** In our custom environments, we make the avoid-regions passable (i.e., the agent can pass through these avoid states). This is in contrast to physical boxes that provide a hard constraint on the state space (i.e., the agent cannot cross the boundaries of the avoid-box) as is used in Cao et al. (2024). Hard constraints on the state space present an issue: no training trajectories generated in this environment truly violate any $\mathbf{b}_j$, because no training trajectories actually pass *through* any $\mathbf{b}_j$ (Figure 3b). Thus, all trajectories are somewhat optimal, in the sense that they never demonstrate "unsuccessful" avoid behavior. This setup thus fails to demonstrate Property (2.2) in Section 2. We argue that this is an issue because it couples successful goal-reaching behavior with successful avoiding behavior. If training trajectories cannot go through avoid-boxes, all training trajectories that have succeeded in reaching the goal must have done so by circumventing avoid-boxes. A training trajectory where the agent is adamant on attempting to go through an avoid-box will get stuck at the box boundary and fail to reach the goal. Therefore, with a hard constraint setup, goal-reaching ability is tightly coupled with avoid ability, while we wish to *isolate* these two aspects and see if a model can learn to acquire both goal-reaching and avoid abilities when it is possible to only acquire one but not the other. While this may not be important in the specific context of these Gymnasium Robotics tasks, in general, there may be application scenarios in which we would like to avoid a region of the state space that the agent is technically *able* to pass through, but we would prefer it not. For example, there may be a road in which a self-driving vehicle *can* pass through or *has* previously passed through, but we would rather it not on some particular day due to construction traffic.

**Environment Specifications.**    For the baseline models (AM-Lag, RbSL, WGCSL), the `FetchReachObstacle` environment state space consists of 19 dimensions. The first 10 dimensions correspond to the state space of the original `FetchReach` environment in Gymnasium Robotics (de Lazcano et al., 2024) and describe information about the location/orientation of the robot arm and the environment in general. The next 9 dimensions describe information about the single obstacle/avoid-region present in the environment, with the same setup as used in Cao et al. (2024). For RADT, the `FetchReachObstacle` environment just has the first 10 dimensions (i.e., the same state space as the original `FetchReach` environment), as RADT does not utilize an avoid-region-augmented state space. For the baseline models, the `MazeObstacle` environment state space consists of $4 + 2 * n_{\text{avoid}}$ dimensions. The first 4 dimensions correspond to the state space of the original `PointMaze` from Gymnasium Robotics and describe the location and velocity of the

agent. Then, for each avoid-region in the environment, there are an additional 2 dimensions added to the state space representing the $xy$-location of the centroid. Thus, the state space dimensionality gets larger the more avoid-regions the environment is specified to have. For RADT, however, the state space of `MazeObstacle` is just the state space of `PointMaze` and contains 4 dimensions. The avoid-regions in the `MazeObstacle` environment are circles of radius 0.2 around the avoid centroids.

Each environment has a parameter `max_episode_steps`, which dictates the maximum number of timesteps in an episode of interaction with that environment. An episode ends either when the `max_episode_steps` number of timesteps is reached or if the agent successfully reaches the goal. `max_episode_steps` is set to be 50 for `FetchReachObstacle` and 300 for `MazeObstacle`, which correspond to the original default values of `max_episode_steps` for `FetchReach` and `PointMaze_UMaze` from Gymnasium Robotics.

**Length-Normalized Cost Return.** We observe in the visualizations of our preliminary experiments that it is possible to "hack" absolute cost return by attempting to reach the goal state in as few timesteps as possible and rushing directly through the avoid-region (Figure 7). Such a policy, which we shall refer to as the "rushed policy," can achieve low absolute cost by minimizing the number of timesteps in the trajectory in total, and thus also minimizing the absolute number of timesteps spent in the avoid-region. However, for reach-avoid applications where safety can be critical, the rushed policy is not preferred to a slower, more cautious policy (which we shall refer to as the "cautious policy") that may take more timesteps to reach the goal but demonstrates a stronger attempt at circumventing the avoid-region. The cautious policy may result in a trajectory that accumulates a similar absolute cost return as the rushed policy trajectory (because it may have skimmed the avoid-region for a similar absolute number of timesteps as the rushed policy spent in the avoid-region), but because the overall cautious policy trajectory makes a better attempt at circumventing the avoid-region, it achieves a much lower length-normalized cost return compared to the rushed policy trajectory. That is, a smaller *proportion* of the timesteps in the cautious policy trajectory violate the avoid-region compared to the rushed policy trajectory, indicating higher quality avoid behavior compared to the rushed policy trajectory. In safety-critical contexts, as long as the agent can reach the desired goal in a reasonable time (i.e., the specified `max_episode_steps`), the *quality* of avoid behavior is more important than the speed of reaching the goal. Figure 7 demonstrates this with two visual examples.

**Avoid-Region Relabeling.** For training data from the `FetchReachObstacle` environment, hindsight-relabeled avoid centroids are sampled according to the contour-based sampling method described in A.3. For training data from the `MazeObstacle` environments, hindsight-relabeled avoid centroids are sampled using the naive uniform strategy, but only from the parts of the state space that are accessible by the agent; i.e., avoid centroids cannot be sampled inside the walls of the mazes. This is to prevent the model from conflating dynamically specified avoid-regions in the prompt with hard constraints on the state space that are inherent to the environment. Since the avoid-regions in `MazeObstacle` environments are circles of radius 0.2, we choose the circumscribing box of width 0.4 around the avoid centroid for our avoid-box representation in RADT, a conservative choice.

**Baselines Cannot Generalize Zero-shot.** Here, we clarify in more concrete examples why zero-shot generalization to avoid-box sizes and numbers is not feasible with AM-Lag and RbSL as they are set up in Cao et al. (2024).

For the `FetchReachObstacle` tasks, assume at training time the avoid-boxes $b$ in the environment have width $w$. The augmented state space only includes information about the center of the avoid-box: `centroid(`$b$`)`. The *size* of the avoid-box is encoded by the cost function. The cost function that AM-Lag and RbSL are trained on outputs a cost of 1 if the agent's state is within the box of width $w$ surrounding `centroid(`$b$`)`, otherwise, it outputs 0. Now, say we increase the avoid-box sizes to $2w$ at evaluation time. The only information the agents get about the avoid-region at evaluation time is in the centroid information `centroid(`$b$`)` present in the state space, which carries no information about the box size. The agents are still trained to stray away from the box of width $w$ surrounding `centroid(`$b$`)`, as this is the cost function they are trained to minimize the value of. Thus, to generalize to this new box size, we must define a new cost function that outputs a cost of 1 if the agent's state is within the box of width $2w$ surrounding `centroid(`$b$`)`, then *retrain* the agents on this new cost function.

For the `MazeObstacle` tasks, assume at training time that there are 3 avoid-regions $b_1, b_2, b_3$ in the environment. The augmented state space for AM-Lag and RbSL at training time

thus has dimensionality $4 + 2 + 2 + 2 = 10$, with 2 additional dimensions for each of `centroid(`$b_1$`)`, `centroid(`$b_1$`)`, `centroid(`$b_1$`)` (see Environment Specifications above). Say, at evaluation time, we increase the number of avoid-regions to 5: $b_1, b_2, b_3, b_4, b_5$. Now the state space dimensionality is $4 + 2 + 2 + 2 + 2 + 2 = 14$. We will have to retrain the agents on this new state space.

## C.2 Cardiogenesis Environment

**Cardiogenesis Boolean Network Details.** The boolean network model comprises 15 genes, each represented by a binary variable indicating expression (1) or non-expression (0), yielding a discrete state space of size $2^{15}$. The model is depicted by Figure 8a and described mathematically in Singh (2024); Herrmann et al. (2012). Actions correspond to genetic perturbations that flip the expression value of a single gene. After perturbation, the Boolean network to a new gene expression state based on its internal logic. Updates are done asynchronously, as is done in Singh (2024): a random gene node is chosen, and its value is set (either changed or remained in place) such that it satisfies all boolean rules . A *stable attractor state* is defined as a state in which all boolean rules in the boolean network are satisfied, such that the boolean network is self-consistent and further asynchronous updates will not change the node values (assuming absence of an external perturbation).

In our `Cardiogenesis` environment simulations, a state-action-state transition sequence is obtained as follows: 1) we start with the boolean network representing the current gene expression state $\mathbf{s_t}$, 2) an action is chosen (i.e., a chosen gene node is perturbed and has its value flipped from `0` to `1` or vice versa) to create a post-perturbation *transient state*, 3) `k` asynchronous updates to the boolean network are performed to the network, and 4) the resulting state after `k` asynchronous updates is defined to be the next state in the trajectory, $\mathbf{s_{t+1}}$. This is depicted in Figure 8b. Since the boolean network updates are asynchronous, there is some stochasticity involved in the transitions due to the random selection of genes to be updated at each asynchronous update step. The value of `k` affects how stochastic our transitions are; the higher the value of `k`, the more likely the boolean network model will hit a *stable attractor state* in the process, and the less noisy the transitions of our `Cardiogenesis` environment will be. We choose `k=10`. Note that this value of `k` does not guarantee that all states $\mathbf{s_t}$ will be stable attractor states; this is intentional, since in reality, a cell can be perturbed while it is unstable and still in the process of reaching a stable attractor state.

**Avoid-Region Relabeling.** At training time, the offline dataset of 60,000 random policy steps is split into trajectories of length 30. Hindsight goal relabeling is done as usual. During hindsight avoid-region relabeling, we sample avoid-regions from the top 20 most represented states (attractors and non-attractors) in the offline dataset. This choice results in approximately half of all training trajectories being successful demonstrations of avoid behavior on the first pass of hindsight avoid-region relabeling. Since we are working with a discrete state space, avoid states that are discrete states rather than regions of a continuous state space. However, given an avoid state vector $[b_1\ b_2\ ...\ b_{15}]$ : $b_i \in \{0, 1\}$, we can still create an avoid-box representation $\mathbf{b} \in \mathbb{R}^{30}$ by adding some small, arbitrary margin $\epsilon$ around each dimension to create a box:

$$\mathbf{b} = [(b_1 - \epsilon)\quad ...\quad (b_{15} - \epsilon)\quad |\quad (b_1 + \epsilon)\quad ...\quad (b_{15} + \epsilon)]$$

We choose $\epsilon = 0.001$ arbitrarily.

**Start State/Goal State Sampling.** At evaluation time, start states are randomly chosen from stable attractor states. The goal state is fixed as the FHF state (`000010010100000`) for our experiment.

**Example Trajectories from Experiments.** Here, we show some example trajectories depicting the three scenarios described in Section 5. Note that, unless otherwise specified, we depict trajectories where repeated states are collapsed, which we call the "collapsed trajectory." For example, the trajectory (`a, a, b, c, d, d, d`) would be collapsed into the collapsed trajectory (`a, b, c, d`). It is important to note that it is possible for an action to lead to staying in the same state due to the dynamics of the Boolean network model. These trajectory examples are also organized in Table 5.

As an example of Scenario 1, when the starting state is `000000000000000`, RADT's policy, when given no avoid prompt, always leads to a collapsed trajectory:

(`000000000000000, 100000000001100, FHF`)

However, upon adding `100000000001100` as an avoid prompt, the resulting policy always leads to a collapsed trajectory:

**Table 5:** Examples trajectories from the `Cardiogenesis` environment

| Initial state | Most visited intermediate | Examples |
|---|---|---|
| 000000000000000 | 100000000001100 | **- avoid prompt:**
(000000000000000, 100000000001100, FHF)
**+ avoid prompt:**
(000000000000000, 000011010100000, 000010010100000, FHF) |
| 010111111010011 | 001111011011111 | **- avoid prompt:**
(010111111010011, 001111011011111, FHF)
(10111111010011, 010111111010011, 001111011011111,
...000111010011111, 001111011011111, 100000000001100, FHF)
(010111111010011, 001111011011111, 000111010011111, 001111011011111, FHF)
**+ avoid prompt:**
(010111111010011, 000000000000011, 010010010100000,
...010111111010011, 000010010100000, FHF)
(010111111010011, 000010010100000, 010111111010011, 000010010000000, FHF) |
| 000010010100000 | 000010010100000 | **- avoid prompt:**
(000010010100000, 000010010100000, 000010010100000,
...000010010100000, 000010010100000, FHF)
**+ avoid prompt:**
(000010010100000, 000010010100000, 000010010100000, FHF) |

```
(000000000000000, 000011010100000, 000010010100000, FHF)
```

which is a longer, alternative path that deterministically circumvents `100000000001100`.

In Scenario 2, the stochastic environment plays a much more prominent role. For example, when the starting state is `010111111010011`, and RADT is given no avoid prompt, the resulting policy leads to a variety of trajectories. A few examples are:

```
(010111111010011, 001111011011111, FHF)
```

```
(10111111010011, 010111111010011, 001111011011111,
000111010011111, 001111011011111, 100000000001100, FHF)
```

```
(010111111010011, 001111011011111, 000111010011111,
001111011011111, FHF)
```

Upon adding `001111011011111` as an avoid prompt, we still get some similar trajectories involving `001111011011111` as an intermediate state, but we get a higher frequency of trajectories that manage to avoid it, such as:

```
(010111111010011, 000000000000011, 010010010100000,
010111111010011, 000010010100000, FHF)
```

```
(010111111010011, 000010010100000, 010111111010011,
000010010000000, FHF)
```

The takeaway here is the diversity of trajectories that are possible given the same policy due to the stochastic nature of the environment.

As an example of Scenario 3, when the starting state is `000010010100000`, all trajectories have collapsed form:

```
(000010010100000, FHF)
```

when there is no avoid prompt, meaning the cell can directly reach the goal state from the start state without passing through a third intermediate state. However, the uncollapsed form of these trajectories has length 5 and is:

```
(000010010100000, 000010010100000, 000010010100000,
000010010100000, 000010010100000, FHF)
```

indicating that RADT makes multiple attempts to get out of `000010010100000` to land on `FHF` and only manages to get out on the 5th attempt, spending a total of 5 timesteps stuck in the initial state `000010010100000`. However, when we add the initial state `000010010100000` as an avoid prompt (which is unavoidable since it is the starting state), the uncollapsed form of all resulting trajectories has length 3 instead and is:

```
(000010010100000, 000010010100000, 000010010100000, FHF)
```

Here, RADT cannot avoid the avoid state `000010010100000` entirely, but it is encouraged and finds a way to spend less time at the `000010010100000` state, getting out of the state more quickly (40% fewer attempts).

### C.3 OTHER NOTES

**Lidar-based Observation Spaces and Goal-Conditioning.** Regarding the discussion about OSRL approaches in Section 3, it is worth noting that the standard "reach avoid" benchmarking task for OSRL algorithms, the `Safe Navigation` environment from Safety Gymnasium (Ray et al., 2019; Gronauer, 2022), may seem like it is testing the goal-conditioning and avoid-region-conditioning abilities of OSRL models, but it does not require true goal-conditioning. This is because it utilizes a relative-perspective, lidar-based observation space $\mathcal{S}_{\text{lidar}}$ that allows the agent to only have to learn to reach *one* state in $\mathcal{S}_{\text{lidar}}$ in order to "generalize" to any target location in physical space: the **0** vector (indicating that the agent is 0 distance away from the goal). Therefore, models do not need to generalize to any arbitrary goal in the observation space $\mathcal{S}_{\text{lidar}}$, and thus OSRL models do not need to be truly goal-conditioned in $\mathcal{S}_{\text{lidar}}$ (most are not). The same logic can be applied to argue that the tasks do not check for true avoid-region-conditioning in the observation space. While relative-perspective observation spaces are advantageous in this regard, they cannot be conceived or used in all environments, e.g., cell reprogramming.

## D CODE

The code for this project is included in the supplementary materials zip folder for this submission. It can also be accessed at the following anonymized repository: `https://anonymous.4open.science/r/reach-avoid-decision-transformer-2441/`. Our repository is built on top of the repositories for MGPO (Yuan et al., 2024) and RbSL (Cao et al., 2024).

## E DETAILED RELATED WORK

Below are some additional discussions regarding the categories of related work described in Section 3 and their limitations with regard to satisfying the ideal properties described in Section 2

**Offline Goal-Conditioned RL.** While reward-based OGCRL can technically satisfy Property (4) by designing a reward function that takes into account both goal information and avoid-region information (e.g. positive reward for approaching the goal, negative reward for approaching an avoid-region), they *effectively* fails to satisfy Property (4) in practice. While it is theoretically possible to express information about both desirable and undesirable states in a multi-component reward function, such functions are practically difficult to design such that both these sometimes conflicting desires are balanced properly (Knox et al., 2023; Knox & MacGlashan, 2024; Freitag et al., 2024). Also, the reason why reward-free OGCRL algorithms strictly do not satisfy Property (4) is because, without a reward, there is no way to specify any additional desires outside of the goal **g**, such as avoid-regions. Thus, reward-free OGCRL is usually a good fit in situations where the path to the goal does not matter or the demonstrations used for training are expert/optimal demonstrations that take an ideal path to the goal.

An argument can be made that reward design for reward-based methods are trivial in simpler environments and goal-reaching tasks where we can just define a sparse reward function for reaching the goal. Regarding this, reward-free GCRL has one other major advantage: sample-efficiency. The referenced work (Blier et al., 2021) formally explores why reward-free, self-supervised methods (e.g., contrastive learning methods, DT-based methods like ours, etc.) are more sample-efficient than TD-based methods (e.g., Q-learning variants) with a sparse reward function. The intuition, as explained in (Blier et al., 2021), is that TD-based methods in sparse reward settings do not get any learning signal until a reward is observed, and when a reward is observed, credit assignment among the various state transitions in the trajectories is difficult to disentangle without large amounts of data. Self-supervised approaches, on the other hand, can learn from every individual state transition (by learning how to reach every successor state in every state transition). That is, they work by implicitly learning state succession dynamics with every state transition data point; therefore, self-supervised approaches can be considered in between model-based (explicitly learning environment dynamics)

and model-free RL, and are therefore more sample-efficient for the same reasons why model-based RL is often more sample-efficient.

**Offline Safe RL.** OSRL approaches circumvent the challenge of having to design multi-component reward functions by using a separate cost function to capture information about avoid behavior and solving a constrained Markov decision process (Altman, 1996), isolating the handling of goal-reaching and avoiding desires. We claim in Section 3 that OSRL algorithms are not truly goal-conditioned in general; however, this may seem surprising since a very common benchmark for OSRL algorithms is a reach-avoid environment called `Safe Navigation` from Safety Gymnasium (Ray et al., 2019). However, we explain in Appendix C.3 why the tasks in this environment are not true multi-goal RL tasks.

*Online* **Goal-Conditioned Safe RL.** While the RbSL paper (Cao et al., 2024) (and the associated algorithms RbSL and AM-Lag) is one of the only pieces of literature we have found that explicitly attempts to create an approach that is *offline*, goal-conditioned, and avoids region-conditioned, we acknowledge that there exist other goal-conditioned, avoid-region-conditioned algorithms designed for reach-avoid tasks that are *online* algorithms (Feng et al., 2025; So et al., 2024; Hsu* et al., 2021). Online algorithms may work in domains like robotics, where we can reasonably create safe testing environments or have good simulators, but we would like to create an approach for domains in which online training is not feasible.

**Decision Transformers.** We build our model off of MGPO, which is truly goal-conditioned and thus satisfies Property (3). During the offline pretraining phase, MGPO is purely data-driven and does not use a reward function, fully relying on hindsight goal relabeling. However, MGPO does use an online prompt optimization phase in addition to offline pretraining (failing to satisfy Properties (1) and (2)), utilizes reward functions during online finetuning (ultimately failing to satisfy Properties (5)), and does not explicitly take into account avoid-region information, unless it is baked into the reward function design (effectively failing to satisfy Properties (4)). Our work builds upon MGPO such that it is more ideal for reach-avoid tasks, aiming to satisfy these remaining properties.

## F    LIMITATIONS

The flexibility and zero-shot capabilities of RADT are balanced out by the fact that it takes more computational resources to train upfront compared to the baselines models, having many more parameters as it is based on a GPT-2 architecture (see "Compute Resources" section under Appendix B.2). However, given that RADT can be trained to achieve a strong policy on 1 GPU in less than day in most cases, and the fact that its zero-shot generalizability reduces the amount of task-specific retraining required, this is not an unreasonable tradeoff.

Additionally, as explored in our stress-testing experiments, there does seem to be an empirical limit to RADT's zero-shot generalization to OOD avoid-region specifications (Section 5 Results 4, Appendix G.2). While the extent of the OOD generalization is impressive, there may be scenarios in which a further extent of generalization is desirable.

We also acknowledge that there may be reach-avoid application domains that do not strictly require all of the Desirable Properties presented in Section 2. For example, in many real-world contexts, we do have a lot of high-quality expert demonstrations to use for training (e.g., for autonomous driving). Therefore, the set of Desirable Properties we propose is more of an idealistic upper bound, such that a reach-avoid learning approach that satisfies all of these properties will very likely be appropriate for *any* reach-avoid problem.

## G    ADDITIONAL EXPERIMENTS

### G.1    ABLATION EXPERIMENTS

All ablation studies are performed in the `FetchReachObstacle-BoxWidth0.16` environment, and the AS/GS tokens ablation study was additionally done in the `UMazeObstacle` environment with 3, 5, and 7 avoid states.

**Ablation of the SI token.** As shown in Table 6 in the Appendix H, RADT with the SI token (+SI) does drastically better in terms of both MNC and SR compared to RADT with the SI token removed

and trained on the full dataset of both successful and unsuccessful avoid examples (-SI, full data). This is as expected, since without the SI token, the model has no signal to differentiate what is considered successful and unsuccessful avoid behavior in the training examples. Additionally, +SI also drastically outperforms RADT with the SI token removed and trained on a dataset *only* consisting of successful avoid examples (-SI, successes only). This clearly demonstrates the importance of including both failure examples and the SI token to teach the model what is *undesirable*.

**Ablation of the AS/GS tokens.** As shown in Table 7 in Appendix H, RADT with the AS and GS tokens (+AS,GS) performs notably better in terms of both MNC and SR compared to RADT without the AS and GS tokens (-AS,GS) in `FetchReachObstacle`. In the `UMazeObstacle` environments, we observe that MNC is worse in the -AS,GS experiments than in +AS,GS setup (approaching MNC values closer to those seen with the WGCSL baseline), but there is no significant difference in SR. The performance drops in terms of MNC are not as drastic compared to those seen with the -SI ablation experiments, as -AS,GS model may potentially be able to learn that only the last token in the prompt is considered a goal state regardless of how many tokens precede it. These results show that the AS and GS tokens are not as critical as the SI token, but do provide noticeable benefits.

**Ablation of attention boosting.** We find that adding a bias `adelta` (a hyperparameter) to the attention logits to all prompt tokens, using the strategy described in (Silva et al., 2024), noticeably improves the instruction-following ability of RADT to the prompt (Figure 6 in Appendix H).

**Ablation of avoid-region relabeling.** As shown in Table 11 in Appendix H, the complete ablation of avoid-region relabeling (i.e., using the avoid-regions that were originally generated by the environment during the collection of training data) significantly impairs RADT's MNC performance across both ID and OOD box sizes, demonstrating the critical nature of avoid-region relabeling in `FetchReachObstacle`. Note that goal relabeling and the avoidance success indication (SI token) are left intact in these experiments; therefore, the goal-reaching SR is not significantly impacted.

### G.2 Limits of Zero-shot Generalization

To test the limits of RADT's zero-shot OOD generalization, we train RADT on `UMazeObstacle` trajectories with maximum 3 avoid-regions and evaluated on `UMazeObstacle` environments with 10, 15, and 20 avoid-regions. We also evaluate the strongest baseline, AM-Lag, on these environments, but like with the main experiments, AM-Lag is retrained on each new environment (i.e., ID, not zero-shot) which gives it an advantage over RADT, and thus it represents the "ideal" performance. The results included the results in Table 8.

At 10 avoid-regions, RADT 's zero-shot OOD performance is still on par with the retrained AM-Lag model in terms of MNC and better than the retrained AM-Lag model in terms of SR. At 15 and 20 avoid-regions, however, we see that RADT 's zero-shot avoid behavior starts to lag behind the retrained AM-Lag's according to MNC, indicating that RADT 's OOD performance does indeed have a limit (MNC suffers when the number of avoid-regions is over 300% of the maximum number of avoid-regions seen during training). However, interestingly, the zero-shot RADT policy results in a SR that is still higher than the retrained AM-Lag policies. This is a strong result because retraining AM-Lag ensures that scenarios with a larger number of avoid-regions become in-distribution settings for AM-Lag, whereas they remain out-of-distribution for zero-shot RADT.

**Effects on attention mechanism.** Additionally, we qualitatively analyze the effects of increasing prompt length (due to increasing the number of avoid-regions) on the quality of the attention mechanism. Specifically, we check to make sure that long prompt lengths do not result in critical attention defects such as consistent attention sinks or uniform/zero attention.

For each episode during evaluation, we calculate the attention weights, averaged across all attention heads, that the model gives to the first 30 tokens of the input sequence (including both the prompt and the early state/action tokens in the trajectory history) when generating the *last* action token of the trajectory. We plot these values for 1, 5, 15, and 20 avoid-regions (Fig. 11). We choose to analyze the attention mechanism at the last action token as this is the most challenging token in terms of long-range dependence on the prompt. We notice no major attention sinks or debilitating attention uniformity across the prompt tokens that result from long prompt lengths, as there is still healthy diversity in attention distributions across different trajectories even when there is a very large, OOD number of avoid-region tokens in the prompt (e.g. 15-20).

However, we do note a few interesting observations. First, RADT pays very little to no attention to the state/action tokens in the early portion of the trajectory history (occurring right after the prompt tokens) when generating the last action in the trajectory. This is likely because the model has learned that it is sufficient to make action predictions with a Markovian-like policy, taking into account only the more recent states while ignoring historical states from earlier parts of the trajectory history. Second, while there are no major attention sinks and uniformity observed, we do notice a drop in attention to individual avoid-region tokens on *average* with very OOD prompt lengths (e.g. 15 and 20 avoid-regions). Intuitively, this is to be expected. It is more challenging for a model to keep track of a high number of different avoid-regions, so it must be selective in distributing its attention only to the most relevant ones at any particular action; it cannot afford to constantly pay attention to every single avoid-region like it can when there are only a couple. Additionally, this would be consistent with our discussion in Appendix G.4 "More complex avoid-region shapes." As avoid-regions increase in the `MazeObstacle` environments, avoid-regions increasingly start to overlap with one another to compose effectively larger avoid-regions, so it would make sense that the significance of individual avoid-regions would decrease given the large amounts of overlap. While we acknowledge the decrease in average attention to individual avoid-region tokens with longer prompts can be a limitation, it is much more likely that the difficulty of the task itself would limit the number of avoid-regions before issues related to attention across long contexts arise. For example, 20 avoid regions in the maze environment already makes the environment way too crowded. Third, RADT always puts a lot of attention weight on the *goal token* at the end of the prompt, regardless of how many avoid-region tokens precede it or how long the prompt is as a whole. This is not surprising, as the goal token is likely the most relevant token at the last action of the trajectory, and there is ever only one goal token. However, it does show that RADT is robust to increasing prompt length, as it is still able to identify the critical nature of the goal token despite the many more potentially "distracting" tokens in longer prompts.

### G.3 EFFECTS OF EXPERT-POLICY DATA

To examine the performance ceiling induced by only using random-policy data, we run an experiment in the `FetchReachObstacle` environment where we train RADT on a mix of random-policy and expert-policy data provided by the RbSL study (Cao et al., 2024) at 4 levels (0%, 10%, 20%, and 30% expert-policy data). The MNC and goal-reaching SR results are included in Table 9.

Looking at the MNC values, we see that introducing expert-policy data seems to have little effect on avoid performance for ID box sizes (box width 0.16) and slightly OOD box sizes (box width 0.18). This suggests that, for ID box sizes, the performance ceiling of RADT 's avoidance behavior when using only hindsight-relabeled, random policy training data is close to what RADT can ideally achieve by learning from expert demonstrations (i.e., using only hindsight-relabeled, random-policy data provides no clear limitation).

However, when we get to larger, more OOD box sizes (box width 0.20, 0.22, and 0.24), we see that introducing expert-policy data increasingly starts to make a difference and improve avoid performance. This suggests that using only hindsight-relabeled, random policy training data can limit performance ceiling on RADT for *larger, OOD* box sizes (i.e., RADT can generalize better in a zero-shot manner to larger, more OOD box sizes if provided with expert data). A possible explanation for this is that the expert-policy demonstrations provided by the RbSL study are more conservative trajectories (i.e., widely avoiding the box with a much larger spatial buffer) than hindsight-relabeled, random-policy trajectories. When provided with these demonstrations involving large, wide, conservative arcs to avoid the box, RADT can better generalize to larger box sizes even when the box sizes themselves are technically OOD.

Interestingly, we also observe that the goal-reaching SR values are *adversely* impacted by introducing expert-policy data across all box sizes, and more prominently with larger, more OOD box sizes (box width 0.2, 0.22, 0.24). In fact, with only random-policy data, RADT maintains a high SR across all box sizes, but when expert-policy data is introduced, SR deteriorates with increasing box size. A possible explanation for this is that the cost of the improved avoid performance for larger box sizes provided by the inclusion of expert-policy data is a deterioration in goal-reaching ability. This is expected as larger box sizes require longer trajectories to successfully circumvent, so given that the allotted time to reach the goal is fixed across all experiments, it is more likely that the agent does not successfully reach the goal in time when it is successfully circumventing a very large box.

### G.4 EFFECTS OF CHANGING AVOID-REGION TOKEN REPRESENTATION

To examine how flexible RADT is with regards to the vector representation used for avoid-region tokens $\mathbf{b}_j \in \mathbb{R}^{d_a} : j \in \{1, 2, ..., n_{\text{avoid}}\}$, we train and evaluate RADT on the `UMazeObstacle` environment using the alternative representation for spherical avoid-regions, as described in Appendix A.2. This representation should allow the model to learn a tighter bound around the avoid-regions, compared to the more conservative circumscribing box representation described in Appendix C.1.

As the results in Table 10 show, changing the avoid-region token representation from the box representation to the spherical representation does not impact RADT's to learn strong reach-avoid policies, with performance that is comparable to the original `UMazeObstacle` experiments. One difference is that using the performance using the spherical representation seems to lead to *slightly* worse MNC and a higher SR; this is consistent with intuition, as the original circumscribing box representation would provide a more conservative boundary of the avoid-regions in `UMazeObstacle` and thus result in more conservative trajectories that have better MNC at the cost of goal-reaching SR.

**More complex avoid-region shapes.** In addition to being flexible to different avoid-region token representations, RADT can be prompted to avoid more complex avoid-regions by representing them as a composition of multiple smaller avoid-regions. This is implicitly demonstrated in the `MazeObstacle` experiments with a high number of avoid-regions (Section 5 Results 2, Section 5 Results 4 "Extremities of OOD generalization", Appendix G.2). In these settings, the high number of randomly generated avoid-regions will result in groups of partially overlapping avoid-regions that compose together to create effectively larger avoid-regions of complex shapes (Fig. 10).

### G.5 IMAGE-BASED OBSERVATION SPACES AND AVOID-REGION REPRESENTATIONS

To explore RADT's potential in handling 1) high-dimensional observation spaces and 2) unstructured, flexible avoid-region token representations that can capture abstract avoidance desires without knowledge of a pre-defined vectorization, we conduct a preliminary proof-of-concept demonstration of RADT applied to an image-based version of `UMazeObstcale` that we call `UMazeObstacleImage`. Here, we replace the observation space for `UMazeObstacle` with an image-based observation space. To obtain state $\mathbf{s}_t$, we take an 224x224 image of the maze with the agent's current position visualized as a green sphere, then pass the image through Torchvision's pre-trained ResNet-18 model (He et al., 2016; Marcel & Rodriguez, 2010). We then take the representation outputted by the fully-connected layer at the end of the ResNet-18 architecture as our state representation, giving us a 1000-dimensional state vector $\mathbf{s}_t \in \mathbb{R}^{1000}$ (Fig. 12). Similarly, we acquire avoid-region tokens and goal tokens via the same process, except we replace the green sphere representing the agent's position in the original image with blue objects representing the avoid-regions and red spheres representing goals, respectively. Therefore, the avoid-region tokens $\mathbf{b}_j \in \mathbb{R}^{1000}$ and the goal token $\mathbf{g} \in \mathbb{R}^{1000}$ have the same dimensionality as the state vectors.

We then use the same hindsight-relabeled training trajectories from our `UMazeObstacle-NumAvoid1` dataset for training, but replace all the state, avoid-region, and goal tokens with these new representations. We benchmark against AM-Lag, RbSL, and WGCSL as baselines using the same evaluation process described for the `UMazeObstacle` environments. As shown by the results in Table 12, RADT achieves superior MNC and comparable or superior goal-reaching SR to all baselines. The SR values here are lower than the SR results from the original `UMazeObstacle-NumAvoid1` task, but this is to be expected given the much more difficult nature of a high-dimensional state space. The SR values are also within the range of typical SR values that SOTA GCRL approaches achieve on image-based, goal-reaching maze tasks (that do not include an avoidance objective), according to the results from OGBench (which do not involve an avoidance component) (Park et al., 2025). While there is room for additional improvement through further optimization of hyperparameters, this preliminary study demonstrates RADT's strong potential in being adapted to both high-dimensional tasks and flexible avoid-region representations that do not require precise knowledge of a strict, vectorized structure to specify avoid-regions.

## H ADDITIONAL TABLES AND FIGURES

**Table 6:** Results for the success indicator token ablation study for `FetchReachObstacle-BoxWidth0.16` (avg. over 3 seeds).

|  | +SI | -SI, full data | -SI, successes only |
|---|---|---|---|
| **MNC** | $0.049_{\pm0.016}$ | $0.629_{\pm0.018}$ | $0.614_{\pm0.027}$ |
| **SR** | $0.989_{\pm0.01}$ | $0.921_{\pm0.016}$ | $0.917_{\pm0.024}$ |

**Table 7:** Results for the goal start/avoid start tokens ablation study for (avg. over 3 seeds).

| Environment | +AS,GS | | -AS,GS | |
|---|---|---|---|---|
|  | MNC | SR | MNC | SR |
| FetchReachObstacle-BoxWidth0.16 | $0.049_{\pm0.016}$ | $0.989_{\pm0.01}$ | $0.098_{\pm0.014}$ | $0.892_{\pm0.055}$ |
| UMazeObstacle-NumAvoid3 | $0.045_{\pm0.009}$ | $0.868_{\pm0.028}$ | $0.060_{\pm0.005}$ | $0.882_{\pm0.033}$ |
| UMazeObstacle-NumAvoid5 | $0.087_{\pm0.005}$ | $0.853_{\pm0.015}$ | $0.102_{\pm0.004}$ | $0.865_{\pm0.043}$ |
| UMazeObstacle-NumAvoid7 | $0.127_{\pm0.01}$ | $0.832_{\pm0.038}$ | $0.133_{\pm0.008}$ | $0.878_{\pm0.003}$ |

**Table 8:** Results for evaluating RADT on extreme avoid-region numbers in `UMazeObstacle` (avg. over 3 seeds).

| # of avoid-regions | RADT[#] | | AM-Lag | |
|---|---|---|---|---|
|  | MNC (1e-2) | SR | MNC (1e-2) | SR |
| 10 | $\mathbf{16.267}_{\pm1.002}$ | $\mathbf{0.923}_{\pm0.029}$ | $\mathbf{16.2}_{\pm0.5}$ | $0.82_{\pm0.036}$ |
| 15 | $23.667_{\pm0.513}$ | $\mathbf{0.92}_{\pm0.005}$ | $\mathbf{21.6}_{\pm0.656}$ | $0.75_{\pm0.015}$ |
| 20 | $29.333_{\pm1.762}$ | $\mathbf{0.912}_{\pm0.01}$ | $\mathbf{24.397}_{\pm1.114}$ | $0.79_{\pm0.026}$ |

[#] = Model capable of zero-shot generalization. Results highlighted in blue are zero-shot results.

**Table 9:** Results for different levels of expert-random data mixture (avg. over 3 seeds).

| Expert % | 0 | | 10 | | 20 | | 30 | |
|---|---|---|---|---|---|---|---|---|
| Box Width | MNC | SR | MNC | SR | MNC | SR | MNC | SR |
| 0.16 | $0.049_{\pm0.016}$ | $0.989_{\pm0.010}$ | $0.053_{\pm0.004}$ | $0.972_{\pm0.025}$ | $0.055_{\pm0.012}$ | $0.967_{\pm0.029}$ | $0.053_{\pm0.017}$ | $0.95_{\pm0.017}$ |
| 0.18 | $0.099_{\pm0.018}$ | $0.978_{\pm0.009}$ | $0.096_{\pm0.003}$ | $0.921_{\pm0.008}$ | $0.105_{\pm0.024}$ | $0.959_{\pm0.009}$ | $0.103_{\pm0.019}$ | $0.95_{\pm0.033}$ |
| 0.20 | $0.171_{\pm0.027}$ | $0.989_{\pm0.017}$ | $0.146_{\pm0.017}$ | $0.922_{\pm0.011}$ | $0.137_{\pm0.021}$ | $0.928_{\pm0.054}$ | $0.142_{\pm0.031}$ | $0.888_{\pm0.025}$ |
| 0.22 | $0.252_{\pm0.018}$ | $0.978_{\pm0.019}$ | $0.203_{\pm0.032}$ | $0.944_{\pm0.042}$ | $0.191_{\pm0.044}$ | $0.894_{\pm0.034}$ | $0.185_{\pm0.138}$ | $0.856_{\pm0.059}$ |
| 0.24 | $0.407_{\pm0.012}$ | $0.983_{\pm0.0}$ | $0.389_{\pm0.047}$ | $0.906_{\pm0.034}$ | $0.301_{\pm0.015}$ | $0.744_{\pm0.092}$ | $0.317_{\pm0.047}$ | $0.794_{\pm0.041}$ |

**Table 10:** Results for `UMazeObstacle` with alternative representation (avg. over 3 seeds).

| # Avoid | RADT[#] | | AM-Lag | | RADT-AltRep[#] | |
|---|---|---|---|---|---|---|
| | MNC (1e-2) | SR | MNC (1e-2) | SR | MNC (1e-2) | SR |
| 3 | $4.455_{\pm0.895}$ | $0.868_{\pm0.028}$ | $4.47_{\pm0.94}$ | $0.645_{\pm0.01}$ | $4.587_{\pm0.805}$ | $0.91_{\pm0.035}$ |
| 5 | $8.75_{\pm0.531}$ | $0.853_{\pm0.015}$ | $10.14_{\pm1.9}$ | $0.76_{\pm0.01}$ | $8.897_{\pm1.127}$ | $0.883_{\pm0.038}$ |
| 7 | $12.7_{\pm1.0}$ | $0.832_{\pm0.038}$ | $12.72_{\pm0.702}$ | $0.777_{\pm0.028}$ | $12.723_{\pm0.804}$ | $0.887_{\pm0.05}$ |

[#] = Model capable of zero-shot generalization. Results highlighted in blue are zero-shot results.

**Table 11:** Results for the avoid-region relabeling ablation study on the `FetchReachObstacle` environment (avg. over 3 seeds).

| Box Width | + avoid-region relabeling | | - avoid-region relabeling | |
|---|---|---|---|---|
| | MNC | SR | MNC | SR |
| 0.16 | $0.049_{\pm0.016}$ | $0.989_{\pm0.010}$ | $0.372_{\pm0.023}$ | $0.983_{\pm0.016}$ |
| 0.18 | $0.107_{\pm0.018}$ | $0.978_{\pm0.009}$ | $0.516_{\pm0.06}$ | $0.978_{\pm0.01}$ |
| 0.2 | $0.164_{\pm0.027}$ | $0.983_{\pm0.017}$ | $0.654_{\pm0.006}$ | $0.972_{\pm0.025}$ |
| 0.22 | $0.247_{\pm0.018}$ | $0.989_{\pm0.019}$ | $0.667_{\pm0.07}$ | $0.961_{\pm0.01}$ |
| 0.24 | $0.409_{\pm0.012}$ | $1.0_{\pm0.0}$ | $0.784_{\pm0.009}$ | $0.983_{\pm0.017}$ |

Results highlighted in blue are zero-shot results.

**Table 12:** Results for the `MazeObstacleImage-NumAvoid1` environment (avg. over 3 seeds).

| | RADT | AM-Lag | RbSL | WGCSL |
|---|---|---|---|---|
| MNC (1e-2) | $\mathbf{1.1}_{\pm0.95}$ | $1.6_{\pm1.23}$ | $1.87_{\pm0.78}$ | $2.55_{\pm1.44}$ |
| SR | $\mathbf{0.617}_{\pm0.029}$ | $0.427_{\pm0.031}$ | $0.557_{\pm0.121}$ | $\mathbf{0.62}_{\pm0.072}$ |

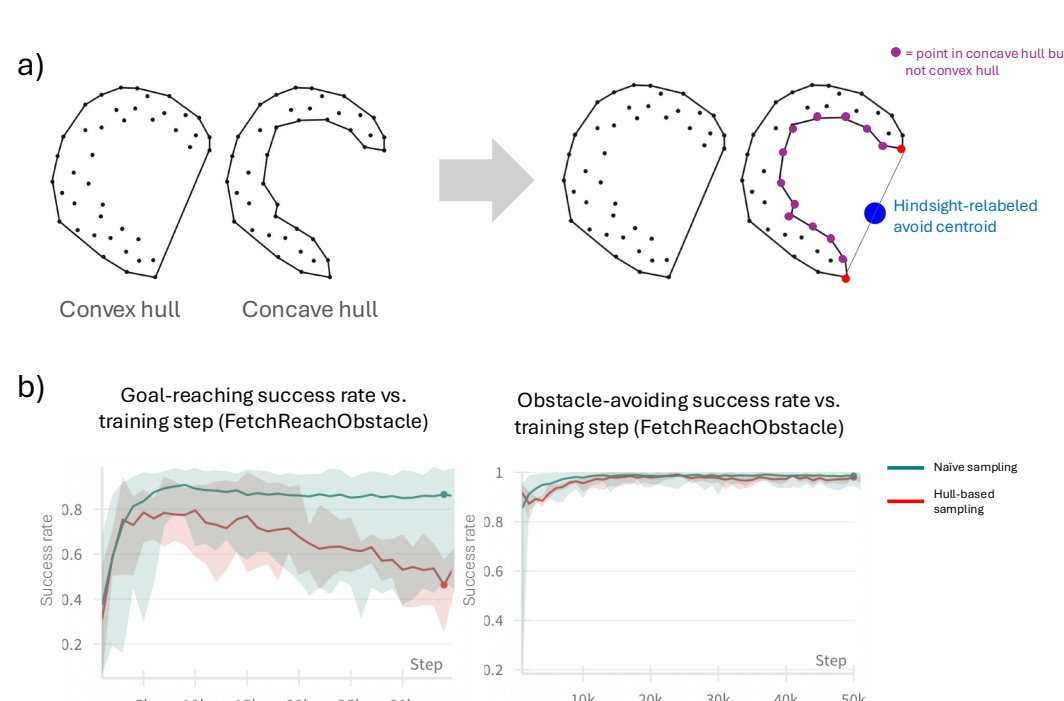

**Figure 5:** a) 2D depiction of the contour-based centroid sampling strategy for avoiding region relabeling. The convex and concave hulls are calculated for the set of data points in the state space for a training trajectory. Points that are part of the concave hull but not the convex hull (depicted here in purple) comprise concave portions of the trajectory that can be viewed as "wrapping around" some unknown avoid centroid. To generate an avoid centroid that fits this intuitive interpretation in hindsight, we locate the two points bordering this concave region (depicted here in red) and sample a point in between them. The resulting trajectory of data points looks like it's "trying" to avoid the blue hindsight-relabeled avoid centroid by wrapping around it. Figure built upon an illustration from Vinh & Dung (2023). b) Using the contour-based strategy for sampling avoids centroids results in a higher maximum SR for the `FetchReachObstacle` environment and notably mitigates/delays overfitting.

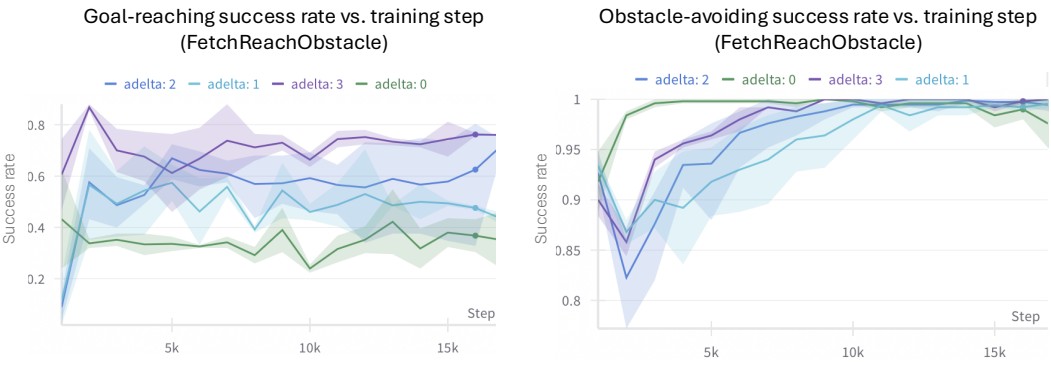

**Figure 6:** The maximum goal-reaching success rate for RADT trained on the FetchReachObstacle task improves with increasing the attention boosting bias `adelta` to the prompt tokens. The maximum obstacle-avoiding success rate, on the other hand, seems to be unaffected by the value of `adelta`.

Rushed policy

Cautious policy

```
Cost return: 2
Length-normalized cost return: 0.67
```

```
Cost return: 3
Length-normalized cost return: 0.3
```

**Figure 7:** A policy that rushes directly through the avoid-region to get to the goal as quickly as possible may earn a low absolute cost return but a high length-normalized cost return. On the other hand, a slower, more cautious policy that takes more timesteps to reach the goal but attempts to circumvent the avoid-region may accumulate a similar absolute cost return, but a lower length-normalized cost return.

a)

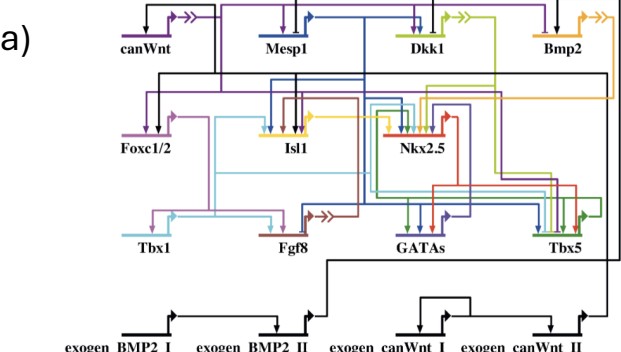

b)

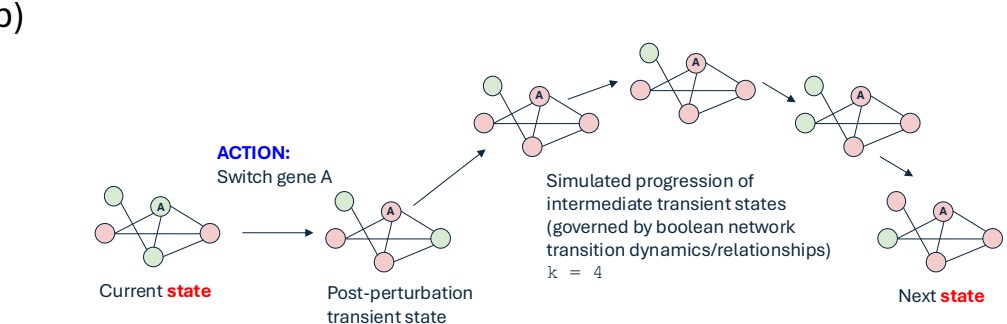

**Figure 8:** a) A diagram representing the 15-gene boolean network model for mouse cardiogenesis. b) A depiction of one state-action-state transition simulated in the `Cardiogenesis` environment, where `k = 4`.

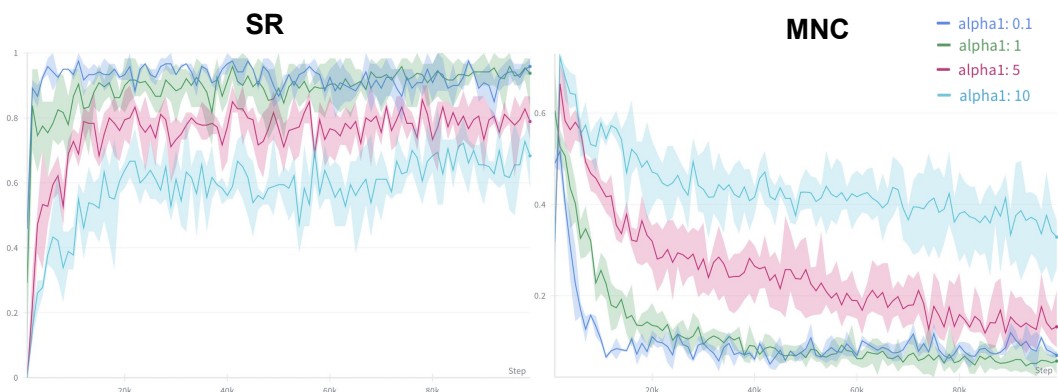

**Figure 9:** The effect of different values for the loss function hyperparameter $\alpha$ (ranging from $\alpha = 0.1$ to $\alpha = 10$). High values of $\alpha$ (e.g. 5, 10) de-prioritze the action loss too much and leads to poor goal-reaching (SR) and avoidance (MNC) behavior in the resulting policy. Small $\alpha$ values (e.g., 0.1, 1) lead to convergence on a policy with strong SR and MNC; however, when $\alpha$ is too small ($\alpha = 0.1$), the lowest MNC achieved is not as low as the lowest MNC achieved when $\alpha = 1$. We chose the value $\alpha = 1$. Training curves shown are averaged across 3 replicate runs per unique value of $\alpha$.

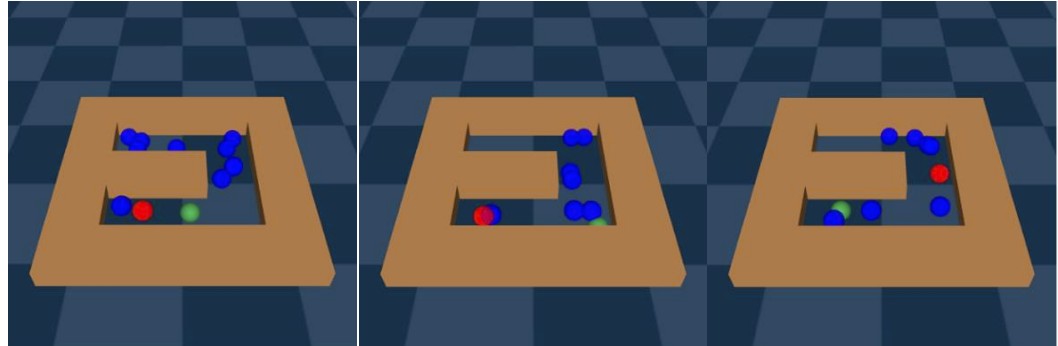

**Figure 10:** When the number of avoid-regions is high (e.g. 7 and 10 avoid-regions depicted in the U-Maze examples here), there is a often quite a lot of overlapping avoid-regions that compose together to form larger avoid-regions of more complex shape that the agent must circumvent.

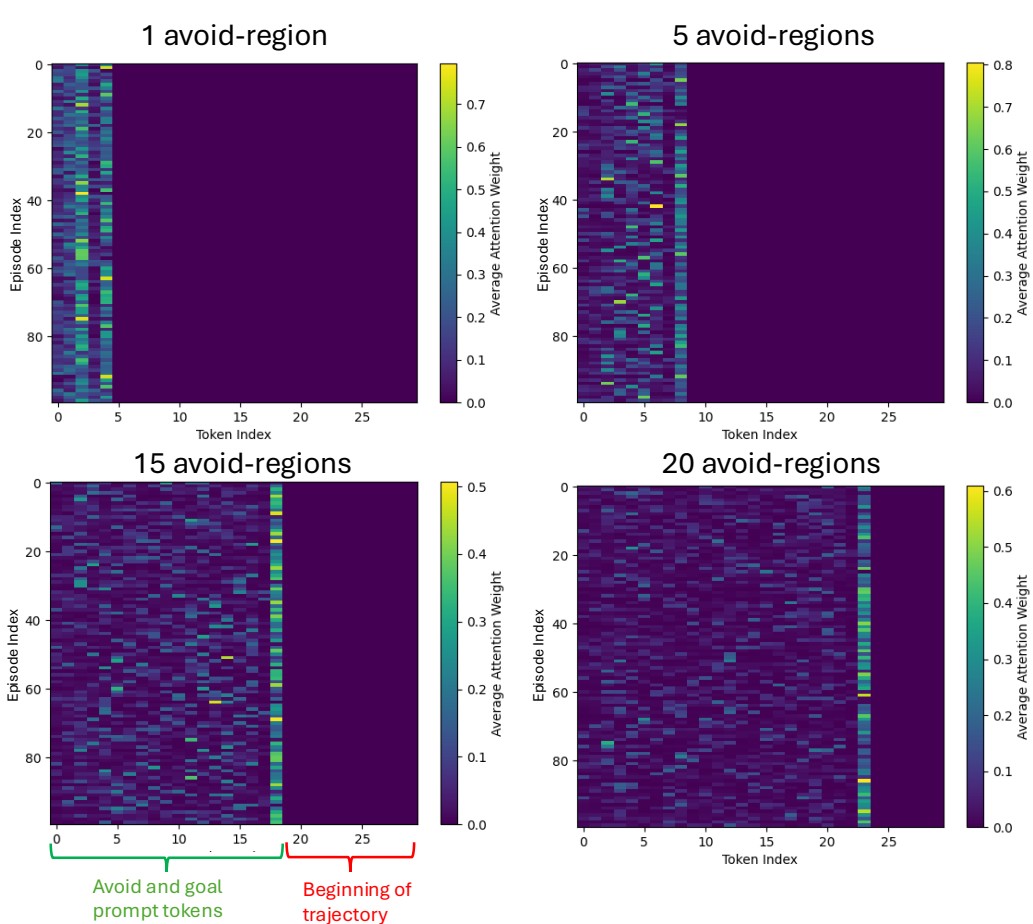

**Figure 11:** Plots of average attention weights to the first 30 tokens of the input sequence when RADT is outputting the last action of an evaluation episode (each episode is a row). The first 30 tokens include both the prompt (which include 1, 5, 15, or 20 avoid-region tokens) and the beginning of the trajectory history. RADT pays little attention to the beginning of the trajectory history as it has learned a mostly Markovian policy. No noticeable attention sinks and debilitating attention uniformity is observed with increasing prompt length, but the attention to individual avoid-region tokens on average does decrease with increasing prompt length. Attention to the goal token remains strong regardless of prompt length.

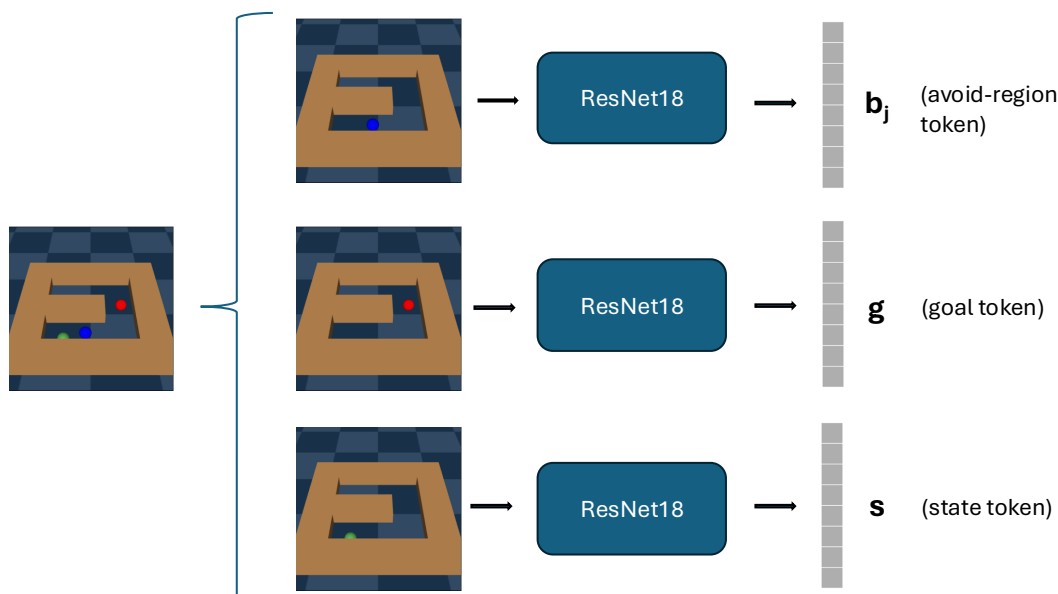

**Figure 12:** To obtain the avoid-region, goal, and state tokens for the `UMazeObstacleImage` environment, we use 224x224 images capturing the positions of the avoid-region(s) (blue), goal state (red), and agent position (green) in the maze. The images are fed through a pre-trained ResNet-18 model to obtain 1000-dimensional flattened vector representations $\mathbf{b}_j$, $\mathbf{g}$, and $\mathbf{s}$.

# I  ADDITIONAL REFERENCES

## ILLUSTRATION CITATIONS (NIAID NIH BIOART)

[The following images are included in our figures.]

NIAID Visual & Medical Arts., (10/7/2024). Fibroblast. NIAID NIH BIOART Source. bioart.niaid.nih.gov/bioart/ 152, a.

NIAID Visual & Medical Arts., (10/7/2024). Fibroblast. NIAID NIH BIOART Source. bioart.niaid.nih.gov/bioart/ 153, b.

NIAID Visual & Medical Arts., (10/7/2024). Fibroblast. NIAID NIH BIOART Source. bioart.niaid.nih.gov/bioart/ 154, c.

NIAID Visual & Medical Arts. (10/7/2024). Generic Im- mune Cell. NIAID NIH BIOART Source. bioart.niaid. nih.gov/bioart/173.

NIAID Visual & Medical Arts. (10/7/2024). Human Male Outline. NIAID NIH BIOART Source. bioart.niaid.nih. gov/bioart/232.

NIAID Visual & Medical Arts. (10/7/2024). Intermediate Progenitor Cell. NIAID NIH BIOART Source. bioart.niaid.nih.gov/bioart/258

## ADDITIONAL REFERENCES

Eitan Altman. Constrained markov decision processes with total cost criteria: Occupation measures and primal LP. *Math. Methods Oper. Res. (Heidelb.)*, 43(1):45–72, February 1996.

Léonard Blier, Corentin Tallec, and Yann Ollivier. Learning successor states and goal-dependent values: A mathematical viewpoint, 2021. URL https://arxiv.org/abs/2101.07123.

R L Graham. An efficient algorith for determining the convex hull of a finite planar set. *Inf. Process. Lett.*, 1(4):132–133, June 1972.

Naoki Katsura and Federico Baldassarre. Cosine annealing with warmup for pytorch. https://github.com/katsura-jp/pytorch-cosine-annealing-with-warmup.git, 2021.

Richard Liaw, Eric Liang, Robert Nishihara, Philipp Moritz, Joseph E Gonzalez, and Ion Stoica. Tune: A research platform for distributed model selection and training. *arXiv preprint arXiv:1807.05118*, 2018.

J.-S Park and S.-J Oh. A new concave hull algorithm and concaveness measure for n-dimensional datasets. *Journal of Information Science and Engineering*, 29:379–392, 03 2013.

Alec Radford, Jeff Wu, Rewon Child, David Luan, Dario Amodei, and Ilya Sutskever. Language models are unsupervised multitask learners. 2019.

Pedro Luiz Silva, Fadhel Ayed, Antonio De Domenico, and Ali Maatouk. Pay attention to what matters. In *MINT: Foundation Model Interventions*, 2024. URL https://openreview.net/forum?id=F6o58A0OrX.

Zhixiong Tang. concave_hull. https://github.com/cubao/concave_hull.git, 2022.

Phan Vinh and Nguyen Dung. *Context-Aware Systems and Applications. 11th EAI International Conference, ICCASA 2022 Vinh Long, Vietnam, October 27–28, 2022 Proceedings.* 04 2023. ISBN 978-3-031-28815-9.

Thomas Wolf, Lysandre Debut, Victor Sanh, Julien Chaumond, Clement Delangue, Anthony Moi, Pierric Cistac, Tim Rault, Rémi Louf, Morgan Funtowicz, Joe Davison, Sam Shleifer, Patrick von Platen, Clara Ma, Yacine Jernite, Julien Plu, Canwen Xu, Teven Le Scao, Sylvain Gugger, Mariama Drame, Quentin Lhoest, and Alexander M. Rush. Transformers: State-of-the-art natural language

processing. In *Proceedings of the 2020 Conference on Empirical Methods in Natural Language Processing: System Demonstrations*, pp. 38–45, Online, October 2020. Association for Computational Linguistics. URL `https://www.aclweb.org/anthology/2020.emnlp-demos.6`.

