# OpenReview forum: "Prompting Decision Transformers for Zero-Shot Reach-Avoid Policies"
_ICLR.cc/2026/Conference — Submitted to ICLR 2026_

### Official Review · Reviewer_Tx2w · 2025-10-28

**Soundness:** 3
**Presentation:** 4
**Contribution:** 3
**Rating:** 6
**Confidence:** 4

**Summary:**

The paper focuses on the reach-avoid problem under the offline goal-conditioned setting, and proposes a decision transformer model to achieve zero-shot generalization to varying numbers and sizes of avoid regions. Specifically, it incorporates the avoid-region information into prompt tokens, whose number can be changed during inference. In addition, the paper introduces an avoid-region hindsight relabeling method, which generates safe and unsafe trajectory pair by modifying the avoid region for each trajectory. Experimental results show that the proposed method performs competitively on in-distribution tasks and achieves better overall performance on out-of-distributiontasks without retraining.

**Strengths:**

Strengths

- The paper is intuitive and well-motivated. Incorporating avoid-region information into prompts to adapt to different numbers and sizes of constraints is reasonable and addresses some limitations of prior work.

- The experimental results are convincing, demonstrating the effectiveness of the proposed approach.

- The paper is clearly written, and the implementation details are easy to understand.

- The experiments on extreme OOD generalization are impressive, showing strong generalization capabilities.

**Weaknesses:**

Weaknesses

- The maze tasks used in experiments are relatively simple. It would be better to include more complex control tasks besides fetchreach, such as those from safe RL benchmarks, to further validate the method.

- I did not find ablation studies on the relabeling component, which is an important part of the contribution and should be discussed in the main text.

- Some prior works on offline safe RL have explored dynamic constraints, such as [1,2], which adjust the constraint threshold dynamically. It would be better to discussed and compared.

- While the motivations and problem setting are novel, similar ideas of using prompt-based models for dynamic task adaptation have been explored in prior works [1,3], which may somewhat limit the perceived technical novelty.

[1] Liu, Zuxin, et al. "Constrained decision transformer for offline safe reinforcement learning." International conference on machine learning. PMLR, 2023.

[2] Lin, Qian, et al. "Safe offline reinforcement learning with real-time budget constraints." International Conference on Machine Learning. PMLR, 2023.

[3] Xu, Mengdi, et al. "Prompting decision transformer for few-shot policy generalization." international conference on machine learning. PMLR, 2022.

**Questions:**

Questions and Comments

- In Table 1, RADT does not dominate all baselines simultaneously on both MNC and SR metrics. While I agree that the smaller difference in SR and the significant improvement in MNC demonstrate the advantage of RADT, I wonder whether there is a quantitative metric that could better summarize the overall performance. For example, in classic constraint RL, the best algorithm is typically defined as the one that achieves the highest reward among all algorithms that satisfy the constraint. However, in the avoid-region setting, since no algorithm achieves MNC=0, it is unclear how to fairly evaluate the tradeoff.

- Figure 3 (c, e) seems to duplicate results from Table 1, and could be streamlined.

- I am not sure whether the terms “reward-based” and “reward-free” are used precisely. The referenced work [1] still relies on reward signals to compute return-to-go. When the goal is known, designing sparse rewards is not difficult, so I am not entirely clear about the essential difference and advantage between these two categories. Does it mean Q-learning–based methods might be more sensitive to reward sparsity than supervised RL–based ones?

- The paper mentions hard constraints in the maze environment. Does this mean that agents are physically prevented from crossing walls? If so, the constraint is already enforced by the environment dynamics—why does it still need to be considered in decision-making?

[1] Janner, Michael, Qiyang Li, and Sergey Levine. "Offline reinforcement learning as one big sequence modeling problem." Advances in neural information processing systems 34 (2021): 1273-1286.

---

> ### Author Response · Authors · 2025-11-20
> **Response to Reviewer Tx2w (1/3)**
>
> ***(Please note this is part 1 in a series of 3 posts.)***
>
> Thank you for your feedback and helpful comments! We are happy to hear you find the paper clearly-written and well-motivated. We also appreciate that you find our empirical results to be convincing and impressive, especially with regards to RADT’s strong OOD generalization capabilities.
>
> We have addressed your comments/suggestions below and have also included the results of new experiments that we hope provide further insight and strengthen our paper. We tag sections that contain new experiment results with a **[new results]** tag in the header. Additionally, a revised version of the paper has been uploaded to OpenReview. If you find your concerns are addressed after reviewing our responses, we would greatly appreciate it if you considered raising your score. If there are any remaining questions/comments, please let us know and we would be happy to address them. Thank you!
>
>
> ## W1 **[new results]**: Difficult environments
>
> We acknowledge that we did not use some of the more difficult environments in the Gymnasium Robotics `Fetch` and `Maze` suites. As mentioned in Appendix C.1 under the section “Choice of Environments”, we chose the relatively simpler environments because state-of-the-art (SOTA) offline RL algorithms are able to achieve high goal-reaching success rates (SR) in these environments while being trained on random-policy data (as demonstrated by results in OGBench (Park et al., 2025)), while these algorithms have quite poor goal-reaching SR in more complex environments (e.g. `PointMaze-Large`). The reason why we choose environments where SOTA algorithms can achieve high goal-reaching SR is because if the goal-reaching success SR is too low, our evaluation of avoid behavior will not be very meaningful. As an extreme example, a policy that makes the agent stay in place will achieve a mean normalized cost (MNC) of 0 (avoids all undesirable states), but its SR will also be 0 (gets nowhere close to reaching the goal either); this would not be considered good reach-avoid behavior despite the low MNC. Thus, in order to properly highlight and isolate the quality of avoid behavior, we choose environments where the task of learning to reach the goal itself is very feasible with a suboptimal training set. Furthermore, the reason we do not use common Offline Safe RL (OSRL) benchmarks, such as those from OSRL package, is because most of those environments/tasks do not involve learning a *truly* goal-conditioned policy (see our discussion in Appendix C.3 for details).
>
> We would also like to **emphasize that despite the originally chosen environments being on the simpler side from a goal-reaching perspective, the additional challenge of avoid-regions adds non-trivial complexity.** For example, we have demonstrated how RADT can generalize to complex shapes in the maze tasks that involved many avoid-regions (Section 5 Results 2 and 4 “Extremities of OOD Generalization”, Appendix G.2), since the introduction of a high number of avoid regions within a small maze inevitably results in many overlapping avoid regions that, in composition, effectively create avoid regions of more complex shapes. We have attached images of such scenarios in Figure 10 and have included this discussion in Appendix G.4.
>
> Nonetheless, we recognize the value of evaluating RADT in a challenging environment from the goal-reaching perspective. Therefore, we have conducted a new experiment reported in Appendix G.5 (“Image-Based Observation Spaces and Avoid-Region Representations”), Table 12, and Fig. 12. To explore RADT’s potential in handling 1) high-dimensional observation spaces and 2) unstructured, flexible avoid-region token representations that can capture abstract avoidance desires without knowledge of a pre-defined vectorization, we conduct a preliminary demonstration of RADT applied to an image-based version of `UMazeObstacle` that we call `UMazeObstacleImage`. Here, we replace the observation space for `UMazeObstacle` with an image-based observation space. To obtain state $\mathbf{s}_t$, we take an 224x224 image of the maze with the agent's current position visualized as a green sphere, then pass the image through Torchvision's pre-trained ResNet-18 model. We then take the representation outputted by the fully-connected layer at the end of the ResNet-18 architecture as our state representation, giving us a 1000-dimensional state vector $\mathbf{s}_t \in \mathbb{R}^{1000}$ (Fig. 12). Similarly, we acquire avoid-region tokens and goal tokens via the same process, except we replace the green sphere representing the agent's position in the original image with blue objects representing the avoid-regions and red spheres representing goals, respectively. Therefore, the avoid-region tokens $\mathbf{b}_j \in \mathbb{R}^{1000}$ and the goal token $\mathbf{g} \in \mathbb{R}^{1000}$ have the same dimensionality as the state vectors.
>
> ***[Response to W1 continued in next comment]***

---

> ### Author Response · Authors · 2025-11-20
> **Response to Reviewer Tx2w (2/3)**
>
> ***[Response to W1 continued]***
>
> We then use the same hindsight-relabeled training trajectories from our `UMazeObstacle-NumAvoid1` dataset for training, but replace all the state, avoid-region, and goal tokens with these new representations. We benchmark against AM-Lag, RbSL, and WGCSL as baselines using the same evaluation process described for the `UMazeObstacle` environments. As shown by the results in Table 12 (simplified version provided below), RADT achieves superior MNC and comparable or superior goal-reaching SR to all baselines. The SR values here are lower than the SR results from the original `UMazeObstacle-NumAvoid1` task, but this is to be expected given the more difficult nature of a high-dimensional state space. The SR values are also within the range of typical SR values that SOTA GCRL approaches achieve on image-based, goal-reaching maze tasks (that do not include an avoidance objective), according to the results from OGBench (which do not involve an avoidance component). While there is room for additional improvement through further optimization of hyperparameters, this preliminary study demonstrates RADT's strong potential in being adapted to both high-dimensional tasks and flexible avoid-region representations that do not require precise knowledge of a strict, vectorized structure to specify avoid-regions.
>
> ||RADT|AM-Lag|RbSL|WGCSL|
> |---:|:---:|:---:|:---:|:---:|
> |**MNC (1e-2)**|1.1|1.6|1.87|2.55|
> |**SR**|0.617|0.427|0.517|0.62|
>
> ## W2 **[new results]**: Avoid-region relabeling ablation
> In addition to the previously-included experiment comparing the different relabeling strategies we used (Appendix A.3.2, Figure 5), we have now conducted an explicit ablation study of avoid-region relabeling as well. We have included the results for this study below and in Section 5 Results 4 “Ablation of avoid-region relabeling”, Appendix G.1, and Table 11 of the revised paper. As shown in Table 11 (simplified version below), the complete ablation of avoid-region relabeling (i.e., using the avoid-regions that were originally generated by the environment during the collection of training data) significantly impairs RADT’s MNC performance across both ID and OOD box sizes, demonstrating the critical nature of avoid-region relabeling in FetchReachObstacle. Note that goal relabeling and the avoidance success indication (SI token) are left intact in these experiments; therefore, the goal-reaching SR is not significantly impacted.
>
> **MNC:**
> |Box width|+ Avoid relabeling|- Avoid relabeling|
> |---:|:---:|:---:|
> |0.16|0.049|0.372|
> |0.18|0.107|0.516|
> |0.2|0.164|0.654|
> |0.22|0.247|0.667|
> |0.24|0.409|0.784|
>
> **SR:**
> |Box width|+ Avoid relabeling|- Avoid relabeling|
> |---:|:---:|:---:|
> |0.16|0.989|0.983|
> |0.18|0.978|0.978|
> |0.2|0.983|0.972|
> |0.22|0.989|0.961|
> |0.24|1.0|0.983|
>
> ## W3: Other OSRL baselines
>
> We acknowledge the relevance of Constrained DT [1] and TREBI [2] to our work. However, Constrained DT does not satisfy being goal-conditioned and avoid-region conditioned like RbSL and AM-Lag (falling in the category of Offline Safe RL but not Offline Goal-Conditioned Safe RL, as described in Section 3), making it difficult for them to be compared/benchmarked effectively against RADT. That is, they cannot generalize zero-shot to any arbitrary goal state in the state/observation space without re-specification of the reward function or re-training. While they may have been evaluated in a couple environments that are seemingly goal-reaching, those tasks are not truly goal-conditioned in the observation space (see our discussion in Appendix C.3). However, they have now been included and cited in the Offline Safe RL section in Section 3 (Related Works) of the revised paper.
>
> ## W4: Other prompt-based models
>
> We acknowledge that Prompt-DT [3] (which is already cited in our paper) utilizes prompting for dynamic task adaptation. However, it is also not truly goal-conditioned and avoid-region-conditioned, failing Properties 3 and 4 (and, less importantly, Property 5). It involves optimizing with respect to a fixed reward function that is not conditioned on a goal, so adapting to a new goal involves using an entirely new reward function. Although Prompt-DT can adapt to new tasks and reward functions in a *few-shot* manner, it cannot do so in a *zero-shot* manner like goal-conditioned methods can.

---

> ### Author Response · Authors · 2025-11-20
> **Response to Reviewer Tx2w (3/3)**
>
> ## Q1: Evaluation metric
> This is a very good point, one that we have discussed throughout this paper. We agree that it is difficult to quantitatively compare reach-avoid quality when the two objectives (measured by SR and MNC) can be in conflict with one another. While we can define a single scoring metric that takes a weighted combination of the SR and MNC achieved, the exact weighting between the two components would be arbitrarily decided and likely not universal, as the optimal tradeoff between SR and MNC may differ depending on the priorities of the specific application domain. Therefore, monitoring both SR and MNC as distinct metrics is an unbiased way to evaluate a reach-avoid policy, as this provides the user with the agency to decide which methodology obtains a tradeoff that is most suitable for their problem and priorities. This being said, a user who is choosing to use a reach-avoid policy instead of a traditional goal-reaching policy would be prioritizing safety/avoidance over goal-reaching, as long as there is no significant drop in goal-reaching ability. As you have mentioned, a small drop in goal-reaching SR and the large improvement in MNC makes RADT a strong reach-avoid policy that fits these priorities.
>
> ## Q2: Overlapping information between table and figure
> We recognize that some plots in Figure 3 convey overlapping information with Table 1. However, we choose to show both representations of results, as the figure visualizations allow the reader to more easily process important trends and relationships across various baselines and avoid-region configurations, while the table provides more comprehensive and precise numerical results.
>
> ## Q3: Reward-free vs reward-based
> It is indeed correct that the referenced work [1], like most DT-based approaches, does utilize returns-to-go; therefore, it is not reward-free, but rather rewards-based. Our definition of reward-free is consistent with how it is conventionally used, which is that a reward-free methodology does not require defining a reward function or acquiring reward signals and can learn desirable behavior solely through data-driven approaches like self-supervised learning + hindsight relabeling. This seems to be empirically a successful paradigm for offline GCRL, with most SOTA methods being reward-free, data-driven approaches (Park et al. 2025).
>
> The reviewer brought up an interesting point regarding the simplicity of just defining a sparse reward for goal-reaching tasks, which would trivialize the motivation related to the difficulties of reward design. Regarding this, reward-free GCRL has one other major advantage: sample-efficiency. The referenced work [4] formally explores why reward-free, self-supervised methods (e.g., contrastive learning methods, DT-based methods, etc.) are more sample-efficient than TD-based methods (e.g., Q-learning variants) with a sparse reward function. The intuition, as explained in [4], is that TD-based methods in sparse reward settings do not get any learning signal until a reward is observed, and when a reward is observed, credit assignment among the various state transitions in the trajectories is difficult to disentangle without large amounts of data. Self-supervised approaches, on the other hand, can learn from every individual state transition (by learning how to reach every successor state). That is, they work by *implicitly* learning state succession dynamics with every state transition data point; therefore, self-supervised approaches can be considered in between model-based RL (explicitly learning environment dynamics) and model-free RL, and are therefore more sample-efficient for the same reasons why model-based RL is often more sample-efficient. This discussion has been added to Appendix E of the revised paper under “Offline Goal-Conditioned RL.”
>
> [4] Blier, L., Tallec, C., and Ollivier, Y. (2021). Learning successor states and goal-dependent values: Amathematical viewpoint. arXiv preprint arXiv:2101.07123.
>
> ## Q4: Hard constraints
> Your interpretation is correct. The agent is physically prevented from crossing the walls. These hard constraints still need to be considered by a policy that optimizes for long-term behavior (e.g., a policy ignoring these hard constraints can attempt to go through a wall and cause the agent to be stuck indefinitely). While learning to circumvent these hard constraints should intuitively be simple given that no training trajectories demonstrate crossing these regions, it is still difficult enough of a task that SOTA reach-avoid papers like RbSL (Cao et al. 2024) use impassable obstacles to evaluate avoid behavior (which we discuss in Section 3, “Offline Goal-conditioned Safe RL”) and still achieve less-than-perfect results. Adding soft constraints (i.e., passable avoid-regions) adds another layer of complexity to the task, which is what we demonstrate in our work.

---

### Official Review · Reviewer_CX5u · 2025-10-28

**Soundness:** 2
**Presentation:** 1
**Contribution:** 2
**Rating:** 2
**Confidence:** 3

**Summary:**

**Summary.**
This work introduces RADT, Reach-Avoid Decision Transformer, a novel safe and goal conditioned RL framework via a tokenized conditioning. The RADT is built upon a decision transformer and encodes both goals and avoid regions as prompt tokens. The model is trained on suboptimal, random-policy offline data using hindsight avoid-region relabeling to enable reward-free learning. The method is benchmarked on robotics and biology domains and claims robust zero-shot generalization to out-of-distribution avoid region configurations.

**Review summary.** This paper presents a sound but incremental approach to safe goal-conditioned offline RL through prompt-based decision transformers. While the motivation and the overall structure is clear, the proposed method relies heavily on established hindsight relabeling techniques and assumes access to precise geometric knowledge of unsafe regions, which limits its practical realism. The evaluation design, i.e., allowing violations without terminatio, and the use of only three random seeds, reduce confidence in the reported performance. Methodologically, the success-indicator token is non-deployable in real-time inference, and several modeling choices are insufficiently analyzed. Besides, the reviewer have a suspicion that some of the entire paragraph level was written using LLM at the moment. Therefore, the reviewer assigns a preliminary score of 2 and would reconsider this rating if the authors provide deeper theoretical justification, stronger baselines, and more rigorous empirical validation.

**Strengths:**

**Writing**
- The paper is well-structured.
- Fig. 2 well illustrates the prompt pipeline.
- The authors clearly develop the motivation behind the necessity of each technology and the problem setting.
    - Prompting as decoupling mechanism. prompt tokens decouple task spec from the state and enable test-time conditioning.
    - Zero-shot matters. The narrative ties zero-shot generalization to realistic deployments where avoidance constraints vary.

**Method**
- The authors explains embeddings and ordering of prompt tokenization clearly.
- $k_t$ prediction and loss encourage explicit awareness of being inside/outside avoid boxes.

**Experiments**
- Table 1 and 2. RADT generalizes to unseen avoid sizes, and unseen counts, often matching or beating retrained baselines. In addition, they shows the performance when extrems up to $20$ avoid regions in Table 7.
- Table 5 and 6. This work includes enough ablations, removing SI, AG, GS degrades MNC.
- Table 9. Alternative spherical encoding shows similar trends
- It is interesting that both the biological domain and the robotics domain are different.

**Weaknesses:**

**LLM**
- I found three times **\*something\*** letters. Although using LLM to correct the grammar and rephrase some sentences, regarding the appendix, the reviewer is concerned that some of the entire paragraphs may have been written by LLM.
    - Line 422: We then run a second set of 200 episodes, this time providing the most visited intermediate state as an **\*avoid token\*** in the prompt.
    - Line 1229 and 1236: **\*larger, OOD\***  and **\*adversely\***

---
**Writing**
- "Avoid" is a verb. Something is not "avoid-region," it is "unsafe/hazardous/dangerous/etc region or condition," which made it a bit clunky to read when this is done so frequently.
- Notation for prompts and embeddings could be formalized more cleanly to aid readers unfamiliar with tokenized sequential modeling.
- Some figures are redundant, for example, Figures 5 and 7 are overlapped, and could be consolidated for clarity.
    - Make sure the text within the figure is at a similar level to the text itself.
    - Make sure the mathematical notation in the figure is the same as in the main text.
    - In Figure 3 and 4, there are cases where there is no error bar or the error bar is not distinct.
    - In Appendix C.2, please make a table for organizing binary state representation, e.g., initial state - 000000000000000.
- The reviewer thinks that **Properties** described in Section 2 are more accurately assumptions and problem formulation rather than properties.
    - The authors would benefit from an explicit discussion of when these assumptions (properties) might fail.

---
**Methodology**
- The method presumes precise and vectorized knowledge of avoid-regions at inference time. The reviewer thinks that this is a strong perception-free assumption, limiting applicability in realistic sensory environments.
- The reviewer thinks that hindsight avoid-region relabeling is incremental, and it seems like engineering improvement, a expansion of goal relabelling into avoid-region.
- The SI is offline/hindsight information. At evaluation, authors always condition on $z=1$ or $z=0$, which cannot be known online and therefore is not a deployable input.
- They set the joint loss coefficient as a fixed $\alpha=1$ with the claim that effects vanish with enough training. However, the impact of tuning this hyperparameter is not deeply evaluated and provided.
- RADT allows prompts of effectively unbounded length, but the impact of prompt length or order on policy inference is not discussed or ablated.

---
**Experiments**
- While RADT maintains strong goal-reaching SR for up to 20 avoid regions, MNC begins to degrade notably past the regime seen in training. The reivewer thinks that this results suggest the zero-shot generalization is limited and could be sensitive to further task scaling.
- Environments allow passing through avoid regions and do not terminate episodes on violation at evaluation. The authors mention that SR can remain high while MNC is non-trivial, decoupling safety from success. The reviewer thinks that this is non-fair evaluation.
- It is hard to strongly trust empirical results due to the low number of seed (3 seeds).
- Baselines are retrained for each configuration, which structurally disadvantages them against RADT’s flexible prompting. More recent constrained- or prompt-DT baselines would strengthen the case [1, 2, 3].
- No wall-clock/params/memory comparison vs. AM-Lag/RbSL/WGCSL, making deployability hard to assess.

---
**Suggested references**

[1] M. Xu, et al. Prompting Decision Transformer for Few-Shot Policy Generalization. ICML 2022.

[2] C. Cao, et al. Offline Goal-Conditioned Reinforcement Learning for Safety-Critical Tasks with Recovery Policy. ICRA 2024.

[3] H. Lin, et al. Safety-aware Causal Representation for Trustworthy Offline Reinforcement Learning in Autonomous Driving. RA-L 2024.

**Questions:**

* Can the authors provide quantitative comparisons of resource footprint between RADT and baselines?
* How does the model handle scaling to situations where avoid regions have complex geometry or there is significant spatial/temporal overlap between many avoid regions?
* What is the effect of prompt length on inference time, training convergence, and attention allocation?
* How sensitive is the model to the weighting parameter $\alpha$ between the action and avoid-awareness losses?
* Could dynamic weighting or alternative normalization improve outcomes, particularly in OOD settings?
* Can the relational inductive bias for structured avoidance be encoded into the prompt or architecture, and does it help?

---

> ### Author Response · Authors · 2025-11-20
> **Response to Reviewer CX5u (1/5)**
>
> ***(Please note this is part 1 in a series of 5 posts.)***
>
> Thank you for your valuable feedback and insights! We are glad that you find our paper to be well-organized with a clearly motivated problem, and we appreciate your acknowledgement of our main empirical results demonstrating strong zero-shot OOD performance, our ablation studies, and our study showcasing approach’s flexibility to alternative encodings.
>
> We have addressed your comments/suggestions below and have also included the results of new experiments that we hope provide further insight and strengthen our paper. We tag sections that contain new experiment results with a **[new results]** tag in the header. Additionally, a revised version of the paper has been uploaded to OpenReview. If you find your concerns are addressed after reviewing our responses, we would greatly appreciate it if you considered raising your score. If there are any remaining questions/comments, please let us know and we would be happy to address them. Thank you!
>
> ## W1: LLM Usage
> Regarding the alleged usage of LLM assistants in the Appendix, we strongly affirm that we did not use LLM assistance in writing any section of our paper, including the Appendix, as we are very particular about the way we express our ideas. We acknowledge that there are a couple instances where we use Markdown-style notation to emphasize/italicize words (surrounding the word with asterisks (*)) instead of LaTEX in the Appendix, which can resemble the Markdown output of an LLM. However, this is due to our habit of using asterisks to emphasize words because of the prevalent usage of Markdown in our everyday lives (Slack, messaging apps, OpenReview, etc.), and that habit occasionally slips into our drafts. We must have missed a couple places in the Appendix where we did not properly convert the intended-to-be italicized words into LaTEX formatting. We apologize if this caused a misunderstanding, and have corrected these couple instances for consistency.
>
> ## W2: “Avoid region” terminology
> Regarding the term “avoid region,” this was actually a point of consideration we discussed when writing the paper. We chose the phrase “avoid region” rather than “unsafe/hazardous/dangerous region” because we wanted a general, broadly applicable term that did restrict the *reason* behind why we would want to avoid a certain region, as some regions may be undesirable for reasons that are unrelated to safety (e.g. a region that introduces inefficiencies). Additionally, the terms “hazardous” and “dangerous” have very precise definitions in the cell biology field, which is one of our primary application domains. However, in the revised paper, we have changed instances of the phrase “avoid region” to the hyphenated “avoid-region” to help with readability, as well as changing “avoid token” to “avoid-region token”. Please let us know if you have any alternative suggestions that are both general and would make the writing clearer.
>
> ## W3: Notation clarity, figure redundancy, and other aspects related to paper organization/formatting
>
> Thank you for the writing suggestions, and we have made the appropriate edits in the revised paper. Specifically, we have:
> 1) Clarified notations and made notation more consistent between the figures and the text
> 2) Provided additional background explanations/definitions for readers who are less familiar with sequential decision making and/or tokenization in Sections 2 and 4
> 3) Re-formatted Section 2 and Section 4 to be visually easier to follow
> 4) Made the axis labels and other text in figures larger for readability
> 5) Removed redundancy between Figures 5 and 7 (this was an editorial mistake, the LaTEX reference for Figure 7 was pointing to the wrong file)
> 6) Organized the example Cardiogenesis trajectories in Appendix C.3 into a table (Table 5)
>
> Regarding the lack of visible error bars in some figures, this means there was no variation. For example, in the `Cardiogenesis` task, in scenario 1, the policy was achieving 100% visitation to the intermediate state without avoid-prompting and 0% visitation with avoid-prompting in every trial that was conducted.
>
> ## W4: “Properties” vs. “assumptions”
>
> We use the term “properties” instead of “assumptions,” as we are proposing that these are properties that an effective reach-avoid model *should* satisfy, rather than assumptions we are using for only our model. The term “desiderata” is also valid and may be preferable to some readers, so we have included this term to the headers/xhdrs in Section 2. We also acknowledge that there may be application domains that do not strictly require all of the properties. Therefore, the set of Desiderata/Desirable Properties we propose in Section 2 is more of an idealistic upper bound, such that an approach that satisfies all of these properties will very likely be appropriate for *any* reach-avoid problem. We have now additionally acknowledged in Appendix F (Limitations) when these properties may not be relevant.

---

> ### Author Response · Authors · 2025-11-20
> **Response to Reviewer CX5u (2/5)**
>
> ## W5 **[new results]**: Inflexibility of avoid-region token representations
>
> We acknowledge that the tasks used in our main experiments assume precise knowledge of the vectorized structure of the state space and avoid-region token representation, which may not be readily available in all environments. However, in most previous OSRL methods, explicit knowledge of the state space structure is also assumed, as the user must define a precise cost function for penalizing entry into avoid-regions using the knowledge about the state space structure, so this does not disadvantage our method.
> Nonetheless, this is a very interesting point, so we have conducted an additional experiment described in Appendix G.5 (“Image-Based Observation Spaces and Avoid-Region Representations”), Table 12, and Fig. 12 in the revised paper. To explore RADT’s potential in handling 1) high-dimensional observation spaces and 2) unstructured avoid-region token representations that can capture abstract avoidance desires without knowledge of a pre-defined vectorization, we conduct a preliminary demonstration of RADT applied to an image-based version of `UMazeObstcale` that we call `UMazeObstacleImage`. Here, we replace the observation space for `UMazeObstacle` with an image-based observation space. To obtain state $\mathbf{s}_t$, we take a 224x224 image of the maze with the agent's current position visualized as a green sphere, then pass the image through Torchvision's pre-trained ResNet-18 model. We then take the representation outputted by the fully-connected layer at the end of the ResNet-18 architecture as our state representation, giving us a 1000-dimensional state vector $\mathbf{s}_t \in \mathbb{R}^{1000}$ (Fig. 12). Similarly, we acquire avoid-region tokens and goal tokens via the same process, except we replace the green sphere representing the agent's position in the original image with blue objects representing the avoid-regions and red spheres representing goals, respectively. Therefore, the avoid-region tokens $\mathbf{b}_j \in \mathbb{R}^{1000}$ and the goal token $\mathbf{g} \in \mathbb{R}^{1000}$ have the same dimensionality as the state vectors.
>
> We then use the same hindsight-relabeled training trajectories from our 1`UMazeObstacle-NumAvoid1` dataset for training, but replace all the state, avoid-region, and goal tokens with these new representations. We benchmark against AM-Lag, RbSL, and WGCSL as baselines using the same evaluation process described for the `UMazeObstacle` environments. As shown by the results in Table 12 (simplified version provided below), RADT achieves superior MNC and comparable or superior goal-reaching SR to all baselines. The SR values here are lower than the SR results from the original `UMazeObstacle-NumAvoid1` task, but this is to be expected given the much more difficult nature of a high-dimensional state space. The SR values are also within the range of typical SR values that SOTA GCRL approaches achieve on image-based, goal-reaching maze tasks (that do not include an avoidance objective), according to the results from OGBench (which do not involve an avoidance component).
>
> While there is room for additional improvement through further optimization of hyperparameters, this preliminary study demonstrates RADT is applicable to high-dimensional tasks and flexible avoid-region representations that do not require precise knowledge of a strict, vectorized structure to specify avoid-regions.
>
> ||RADT|AM-Lag|RbSL|WGCSL|
> |---:|:---:|:---:|:---:|:---:|
> |**MNC (1e-2)**|1.1|1.6|1.87|2.55|
> |**SR**|0.617|0.427|0.517|0.62|
>
> ## W6: Value of conditional SI token at deployment time
>
> At deployment time, we always condition on $z=1$ (as described in Section 4) because we always want the trajectory to be a success. $z=0$ is only used at training time to explicitly teach the model what is considered an undesirable failure example. This is similar to how traditional decision transformers are conditioned on both low and high return during training time to teach the model what behavior leads to good/bad return, but at evaluation time the model is always conditioned on a high return.
>
> ## W7 **[new results]**: Joint loss coefficient HPO
>
> We’ve attached the training curves for early hyperparameter optimization in Fig. 9 of the revised paper. We find that values of $\alpha$ between 0.5 and 2.0 (i.e., one component being weighted up to twice as much as the other) does not have a significant effect on empirical performance when we let the model train for long enough, as measured by validation goal-reaching success rate and unnormalized cost (Fig. 9). Thus, we set it to 1.0 in all our experiments, as that is the value that empirically led to the fastest convergence (but not by a significant amount). We have updated Appendix B.2 in the revised paper with this information.

---

> ### Author Response · Authors · 2025-11-20
> **Response to Reviewer CX5u (3/5)**
>
> ## W8 **[new results]**: Effects of long prompts on attention
>
> This is an interesting question, and we have conducted an additional analysis of the “Extremities of OOD generalization” experiments that show how increasing prompt length at inference time affects attention allocation. The results are included in Appendix G.2 “Effects on attention mechanism” and Fig. 11 of the revised paper. Specifically, we check to make sure that long prompt lengths do not result in critical attention defects such as consistent attention sinks or uniform/zero attention.
>
> For each episode during evaluation, we calculate the attention weights, averaged across all attention heads, that the model gives to the first 30 tokens of the input sequence (including both the prompt and the early state/action tokens in the trajectory history) when generating the *last* action token of the trajectory. We plot these values for 1, 5, 15, and 20 avoid-regions (Fig. 11). We choose to analyze the attention mechanism at the last action token as this is the most challenging token in terms of long-range dependence on the prompt. We notice no major attention sinks or debilitating attention uniformity across the prompt tokens that result from long prompt lengths, as there is still diversity in attention distributions across different trajectories even when there is a large, OOD number of avoid-region tokens in the prompt (e.g. 15-20).
>
> However, we do note a few interesting observations.
>
> * First, RADT pays very little to no attention to the state/action tokens in the early portion of the trajectory history (occurring right after the prompt tokens) when generating the last action in the trajectory. This is likely because the model has learned that it is sufficient to make action predictions with a Markovian-like policy, taking into account only the more recent states while ignoring historical states from earlier parts of the trajectory history.
>
> * Second, while there are no major attention sinks and uniformity observed, we do notice a drop in attention to individual avoid-region tokens on *average* with very OOD prompt lengths (e.g. 15 and 20 avoid-regions). Intuitively, this is to be expected. It is more challenging for a model to keep track of a high number of different avoid-regions, so it must be selective in distributing its attention only to the most relevant ones at any particular action; it cannot afford to constantly pay attention to every single avoid-region like it can when there are only a couple. Additionally, this would be consistent with our discussion in Appendix G.4 “More complex avoid-region shapes.” As avoid-regions increase in the `MazeObstacle` environments, avoid-regions increasingly start to overlap with one another to compose effectively larger avoid-regions, so it would make sense that the significance of individual avoid-regions would decrease given the large amounts of overlap. While we acknowledge the decrease in average attention to individual avoid-region tokens with longer prompts can be a limitation, it is much more likely that the difficulty of the task itself would limit the number of avoid-regions before issues related to attention across long contexts arise. For example, 20 avoid regions in the maze environment already makes the environment way too crowded.
>
> * Third, RADT always puts a lot of attention weight on the *goal token* at the end of the prompt, regardless of how many avoid-region tokens precede it. This is not surprising, as the goal token is likely the most relevant token at the last action of the trajectory, and there is ever only one goal token. However, it does show that RADT is robust to increasing prompt length, as it is still able to identify the critical nature of the goal token despite the many more potentially "distracting" tokens in longer prompts.
>
> The main trends in Fig. 11 are summarized in this table (attn. @ last action token in traj.):
> |**# of avoid-regions**|**1**|**5**|**15**|**20**|
> |---:|:---:|:---:|:---:|:---:|
> |**# of clear attention sinks**|0|0|0|0|
> |**Mean attention to early state/action tokens**|0|0|0|0|
> |**Mean attention to avoid-region tokens**|0.247|0.121|0.047|0.036|
> |**Mean attention to goal token**|0.281|0.238|0.228|0.223|
>
>
> ## W9: Generalization to extreme OOD regimes
>
> We acknowledge your concern. However, we conducted the experiment in Section 5 Results 4 “Extremities in OOD Generalization” precisely to explore and acknowledge the limit to RADT’s OOD generalization ability. Nonetheless, the fact that RADT can maintain competitive MNC to the best *re-trained baseline* up to 10 avoid-regions (over 300% the max number seen training) is still impressive.

---

> ### Author Response · Authors · 2025-11-20
> **Response to Reviewer CX5u (4/5)**
>
> ## W10: Episode termination
>
> We do not terminate the episode upon violating an avoid-region because we would like RADT to be applicable in scenarios where certain regions are undesirable, but not necessarily strictly unsafe or episode-terminating. As an intuitive example, say we want to bike to a destination, and there are sections that involve gravel paths rather than paved roads; we would want to avoid routes that travel through these undesirable gravel sections (the avoid-regions) since it is damaging to the tires and uncomfortable to the rider, but going through these gravel paths is not strictly dangerous enough to terminate the episode. Therefore, allowing full trajectories that pass through avoid-regions is the most comprehensive way to evaluate reach-avoid behavior, because it allows us to analyze exactly how much the policy attempts to minimize violation of an avoid-regions if it does violate any (via the continuous metric of MNC), rather than just a binary violated-or-not metric. In strict safety scenarios where any violation is intolerable, it is straightforward to calculate the binary avoidance success of these runs. Additionally, the discussion of Appendix C.1 “Passable and Impassable Avoid-Regions” is also relevant here.
>
> ## W11: Number of training seeds
>
> 3-5 seeds is typical in RL papers. Our baseline methods (RbSL, AM-Lag) and the decision transformer works that precede ours (MGPO, Prompt-DT, DT) were all trained on 3-5 seeds to obtain their paper results. Note that each seed represents a separate *training* run; then for each trained policy, we conduct 50-100 test episodes at evaluation time for each avoid-region configuration.
>
> ## W12: Re-trained baselines and proposed alternatives
>
> As discussed in the paper, re-training the baselines for each configuration actually *disadvantages* RAFT them against the baselines, as the test-time configurations are in-distribution for the baseline, but OOD for RADT. This is because baselines cannot flexibly adapt to OOD avoid-region configurations while RADT can, which is one of the major benefits of RADT, and although the comparison is unfair, it is unfair toward RADT and further emphasizes RADT’s strengths. Of the three suggested alternative baselines, [2] already provides two of our major baselines we compare with (RbSL and AM-Lag), and the other works [1,3] are not truly goal-conditioned RL methods and are thus not directly comparable under our setting. Work [1] was already cited under the category “Offline Safe RL” in Section 3 (Related Works). We have now included the third citation under the Offline Safe RL category in our revised paper.
>
> ## W13: Wall clock time and memory consumption
>
> On average, RADT takes around 10 hours to converge on a strong policy using 1 GPU (depending on the task) in terms of pure training time. The total amount of time depends on how often we perform validation/evaluation. The max memory utilization during training hit 40 GB, typical of DT-based approaches. We acknowledge this is a higher computation overhead compared to baselines, which can converge on a strong policy using 1 GPU within 2 hours of training time, with the maximum memory utilization during training hitting 20-30 GB. However, given that all training was done with a single GPU and a strong policy can be achieved within a day, we do not think RADT is prohibitively expensive to train. Additionally, RADT has the additional benefit of being able to generalize zero-shot to novel avoid-region configurations at test time while other methods do not, significantly saving test-time computation. For this reason, this tradeoff is not unreasonable, as RADT *front-loads* the computational expense to the initial training stage and removes the necessity to re-train on individual avoid specifications downstream. We have included these details in the revised version of Appendix B.2 “Compute Resources.”

---

> ### Author Response · Authors · 2025-11-20
> **Response to Reviewer CX5u (5/5)**
>
> ## Q1, 3, 4:
>
> Q1, 3, and 4 have been addressed in our responses to W13, W8, and W7, respectively.
>
> ## Q2: Complex avoid-regions
>
> This is a good question. In fact, we have implicitly demonstrated how RADT generalizes to complex shapes in the maze results with multiple avoid-regions (Section 5 Results 2 and 4 “Extremities of OOD Generalization”, Appendix G.2), since the introduction of many avoid-regions within a small maze inevitably results in many overlapping avoid-regions that, in composition, effectively create avoid-regions of more complex shapes. We have attached images of such scenarios Fig. 10 and included this discussion in Appendix G.4 “More complex avoid-regions” of the revised paper. Additionally, the new image-based results discussed in W5 suggest that RADT is capable of generalizing to abstract, image-based avoid-region representations, which may allow for more flexible representation of complex avoid-regions.
>
> ## Q4: Dynamic weighting
>
> The attention mechanism does provide dynamic weighting to different avoid-regions depending on the context. However, we acknowledge that having a functionality where we can *manually* specify the relative importance of avoid-regions at test time is very interesting and can be worth exploring in follow-up studies.
>
> ## Q5: Relational inductive biases
>
> This is definitely an interesting idea, where instead of having an all-to-all architecture like the transformer (equivalent to a fully-connected GAT), we can instead have a sparser GAT where each action token only pays attention to the most recent state and to the prompt tokens (but not to previous actions, i.e. Markovian), etc. But, in the spirit of building a reward-free, primarily data-driven method (via hindsight-relabeling), which is the most modern paradigm for GCRL, we choose the all-to-all transformer architecture to allow for maximal flexibility. It is also the architecture that has the most empirical basis in all previous DT works. Interestingly, we do note that in our attention analysis (discussed in W8), the model is able to *learn sparsity*, with the model learning to effectively be Markovian (last action token pays no attention to early states/actions (Appendix G.2 “Effects on attention mechanism”, Fig. 10). Therefore, explicitly exploring sparser GAT architecture is an interesting idea for follow-up studies.

---

> > ### Comment · Reviewer_CX5u · 2025-11-21
> >
> > Thank you for your detailed and thoughtful responses, as well as for the substantial revisions and new experimental results. The reviewer appreciates the level of care you have taken in addressing each point. The added analyses are helpful and improve the clarity of the work. That said, a few concerns remain or have been raised anew after reading the rebuttal.
> >
> > ---
> >
> > `On W4`. While the reviewer appreciates the clarification and the inclusion of **desiderata**, the reviewer still believes that several of the proposed **properties** functionally act as **assumptions** for the method to succeed. This is not necessarily a problem, but I encourage the authors to be more explicit about which elements are truly desirable properties of any reach-avoid method, and which are assumptions specific to RADT. A clearer distinction would improve conceptual transparency.
> >
> > ---
> > `On W5`. First of all, thank you for the additional experiments.
> >
> > The image-based experiment is a promising direction, and the reviewer appreciates the effort to address this concern. However, the reviewer would like to request more concrete experimental details to better understand the setup:
> > 1. What exactly constitutes the input image?
> > 2. How are avoid-regions provided in the image-based setting?
> > 3. How is the exact agent / goal positioning represented?
> > 4. At training time, how do you ensure consistency between vectorized representations in the original dataset and the new image-based embedding version?
> >
> > ---
> > `On W7`. The explored range of 0.5–2.0 seems quite limited. Given that α governs the tradeoff between action prediction and avoid-awareness, the reviewer would expect that a much broader range (e.g., 0.1–10) may reveal more meaningful behaviors.
> >
> > Additionally, **Figure 9 is incomplete**. The learning curves seem to exhibit different lengths, making them hard to compare fairly. Some curves do not appear fully converged.
> >
> > ---
> > `On W8`. The attention-based analysis is interesting, but the reviewer is not fully convinced that the interpretation is complete without further statistical information. Specifically:
> > - Could the authors provide standard deviation or variance of attention weights across episodes?
> > - Are the reported means stable across seeds?
> >
> > This would help confirm whether the observed trends are statistically meaningful or partially noise-driven.
> >
> > ---
> > `On W11`. While the authors argue that 3–5 seeds is typical in RL papers, the reviewer unfortunately cannot fully agree. In many contemporary offline RL and safe RL studies **8–10 seeds** (at least 6 seeds for state-based environments) is considered a more reliable standard for assessing statistical robustness.
> >
> > At minimum, could you report:
> >
> > - **p-values** for the main comparisons?
> > - Bootstrap confidence intervals across evaluation episodes?
> >
> > This would provide stronger statistical grounding and alleviate concerns about variance-driven performance differences.

---

> > > ### Comment · Reviewer_yjF7 · 2025-11-21
> > > **Random Seeds Issue**
> > >
> > > Dear Reviewer CX5u,
> > >
> > > As a fellow reviewer, I agree that the paper still has several weaknesses. However, in the context of offline RL and offline safe RL, using three random seeds is generally considered sufficient for empirical evaluation. Given the limited time during the rebuttal period, I believe it is more important for the authors to focus on addressing the main methodological issues rather than further expanding the statistical robustness analysis.

---

> > > > ### Comment · Reviewer_CX5u · 2025-11-21
> > > >
> > > > Dear Reviewer yjF7,
> > > >
> > > > Thank you for your helpful comments. I fully understand the practical limitations you mentioned, which is why I only requested the minimum necessary p-values and related statistics based on the results currently available.

---

> > > > > ### Author Response · Authors · 2025-11-21
> > > > >
> > > > > Thank you for the timely response and the additional questions/comments.
> > > > >
> > > > > **W5:** As described by Figure 12 and its caption in the revised paper:
> > > > > * Input images are birds-eye renderings of the maze
> > > > > * Avoid-regions are depicted as blue objects within the maze image
> > > > > * Agent position is depicted as a green sphere in the maze image
> > > > > * Goal position is depicted as a red sphere in the maze image
> > > > >
> > > > > We will include these details in the text description of the task in Appendix G as well. Additionally, please note that while this was the specific image-based representation we used, it is not inherent to RADT's ability to use image-based representations. Now that it has been shown that RADT can adapt to image-based representations, any arbitrary representation could work in theory, as long as avoid-regions, goals, and states are always consistently represented visually.
> > > > >
> > > > > Regarding the question “At training time, how do you ensure consistency between vectorized representations in the original dataset and the new image-based embedding version?”: We are not using the vectorized representations of state/goal/avoid-region in the original dataset at all in this new experiment (other than for the environment to use to calculate reward/cost under the hood), so we do not think ensuring consistency between the new image-based observation space and the original observation space is meaningful. The new image-based environment should be treated as a separate environment, independent of the original `MazeObstacle` environments.
> > > > >
> > > > > **W11**: We thank Reviewer yjF7 for the contribution and agree that re-training 8-10 times for each model per experiment setting is impractical given our resources and the limited time in the rebuttal period. Additionally, we have re-confirmed that our baselines (RbSL, AM-Lag, WGCSL) and the base model we build upon (MGPO) were all evaluated on 3-5 seeds in their respective papers. However, we will conduct a non-parametric test using the results we do have to obtain p-values.
> > > > >
> > > > > With regards **W7, W8, and W11,** we will respond with the results once we obtain them.

---

> > > > > > ### Author Response · Authors · 2025-11-22
> > > > > > **W7 Follow-up**
> > > > > >
> > > > > > ### W7 Follow-up: alpha hyperparameter
> > > > > >
> > > > > > Thank you for the suggestion regarding HPO study with $\alpha$! We have now tested a larger range of values for $\alpha$ according to the suggested range (0.1 to 10) and have let the models train for longer. We do indeed uncover additional insights from these experiments. The results are shown in the revised version of Fig. 9 and the discussion in Appendix B.2 is now updated.
> > > > > >
> > > > > > Very large values of $\alpha$ (e.g., 5 and 10) de-prioritize the action component of the loss too much and significantly impairs learning a policy with strong goal-reaching (SR) and avoidance (MNC) behavior. While very small values of $\alpha$ (e.g., 0.1) result in the fastest convergence to a policy with strong SR and MNC performance, the intermediate value of $\alpha = 1$ allows the model to eventually converge on a policy with the *lowest MNC* (overtaking the MNC of  $\alpha=0.1$ runs starting at ~50k training steps), with equally good SR compared to the $\alpha=0.1$ runs. This is consistent with the fact that the role of the avoid-awareness loss component is to boost learning strong avoidance behavior. It is possible that the runs with even higher $\alpha$ values (e.g., $\alpha = 5, 10$) would eventually converge on a policy with even better MNC, but given that they have not achieved a policy anywhere close to acceptable after 24 hours of training, the tradeoff of significantly more computation for potential marginal improvement is difficult to justify. Therefore, we choose the intermediate value of $\alpha = 1$.

---

> ### Author Response · Authors · 2025-11-22
> **W8 Follow-up**
>
> ### W8 Follow-up: attention statistics
>
> Thank you for your suggestion to include more granular results for the attention analysis. Below are tables containing the mean + standard deviation of attention weights across 100 episodes *for each avoid-region (AR) and goal (G) token position*. Additionally, we include a row for each of 3 training seeds in each table.
>
> These results show:
>
> 1) The attention distributions and trends for prompt tokens are consistent across training seeds.
>
> 2) There are no attention sinks or other point abnormalities, as there are no avoid-region token *positions* that have a drastically different mean or standard deviation from the rest.
>
> 3) The distributional spread of attention weights *across episodes* appears healthy. There are no near-zero standard deviation values that would suggest a lack of dynamic adaptation to the particular scenarios of different episodes. The coefficient of variation is rather high (ranging from 0.5 to 1.25 times the mean), but this is not necessarily concerning and may in fact be indicative of strong adaptability, as there can be large differences in avoid-region setups between episodes that will require large differences in how the policy allocates attention amongst the individual avoid-regions. What would be concerning is if there were any particular token positions that had much higher variance relative to the rest, as that would suggest a methodological artifact rather than meaningful variation, but we have observed in point (2) there appears to be no such case.
>
> **Token-wise attention means (across 100 episodes):**
>
> *1 AR:*
>
> |**seed**|**AR1**|**G**|
> |---|---|---|
> |1|0.301|0.315|
> |2|0.339|0.310|
> |3|0.355|0.304|
>
> *5 ARs:*
>
> |**seed**|**AR1**|**AR2**|**AR3**|**AR4**|**AR5**|**AR Avg**|**G**|
> |---|---|---|---|---|---|---|---|
> |1|0.136|0.129|0.095|0.129|0.109|0.120 |0.255|
> |2|0.125|0.118|0.124|0.118|0.125|0.122 |0.225|
> |3|0.129|0.125|0.127|0.123|0.107|0.122 |0.241|
>
> *15 ARs:*
>
> |**seed**|**AR1**|**AR2**|**AR3**|**AR4**|**AR5**|**AR6**|**AR7**|**AR8**|**AR9**|**AR10**|**AR11**|**AR12**|**AR13**|**AR14**|**AR15**|**AR Avg**|**G**|
> |---|---|---|---|---|---|---|---|---|---|---|---|---|---|---|---|---|---|
> |1|0.047|0.048|0.046|0.043|0.051|0.040|0.055|0.049|0.052|0.040|0.055|0.039|0.051|0.049|0.044|0.048 |0.222|
> |2|0.059|0.058|0.051|0.047|0.041|0.043|0.041|0.039|0.044|0.051|0.045|0.044|0.044|0.048|0.049|0.046 |0.212|
> |3|0.056|0.057|0.043|0.045|0.043|0.042|0.038|0.050|0.044|0.046|0.042|0.048|0.045|0.042|0.056|0.047 |0.234|
>
> *20 ARs:*
>
> |**seed**|**AR1**|**AR2**|**AR3**|**AR4**|**AR5**|**AR6**|**AR7**|**AR8**|**AR9**|**AR10**|**AR11**|**AR12**|**AR13**|**AR14**|**AR15**|**AR16**|**AR17**|**AR18**|**AR19**|**AR 20**|**AR Avg**|**G**|
> |---|---|---|---|---|---|---|---|---|---|---|---|---|---|---|---|---|---|---|---|---|---|---|
> |1|0.052|0.046|0.039|0.035|0.049|0.038|0.035|0.027|0.024|0.038|0.035|0.031|0.042|0.029|0.033|0.033|0.036|0.033|0.030|0.030|0.032 |0.210|
> |2|0.055|0.045|0.040|0.033|0.033|0.029|0.031|0.029|0.032|0.034|0.033|0.032|0.037|0.032|0.031|0.038|0.027|0.038|0.032|0.039|0.035 |0.236|
> |3|0.042|0.041|0.041|0.037|0.034|0.034|0.031|0.029|0.040|0.031|0.035|0.037|0.041|0.038|0.037|0.035|0.033|0.037|0.035|0.031|0.034 |0.231|
>
>
>
>
> **Token-wise attention std. deviations (across 100 episodes):**
>
> *1 AR:*
>
> |**seed**|**AR1**|**G**|
> |---|---|---|
> |1|0.139|0.174|
> |2|0.157|0.178|
> |3|0.173|0.167|
>
> *5 ARs:*
>
> |**seed**|**AR1**|**AR2**|**AR3**|**AR4**|**AR5**|**AR Avg**|**G**|
> |---|---|---|---|---|---|---|---|
> |1   |0.138|0.123|0.100|0.148|0.126|0.127 |0.144|
> |2   |0.130|0.121|0.137|0.117|0.124|0.126 |0.145|
> |3   |0.144|0.119|0.145|0.127|0.112|0.129 |0.150|
>
> *15 ARs:*
>
> |**seed**|**AR1**|**AR2**|**AR3**|**AR4**|**AR5**|**AR6**|**AR7**|**AR8**|**AR9**|**AR10**|**AR11**|**AR12**|**AR13**|**AR14**|**AR15**|**AR Avg**|**G**|
> |---|---|---|---|---|---|---|---|---|---|---|---|---|---|---|---|---|---|
> |1|0.061|0.065|0.059|0.054|0.070|0.055|0.073|0.061|0.056|0.060|0.069|0.049|0.052|0.059|0.056|0.057 |0.124|
> |2|0.069|0.080|0.065|0.060|0.060|0.055|0.055|0.044|0.048|0.054|0.054|0.058|0.053|0.064|0.066|0.059 |0.125|
> |3|0.060|0.067|0.049|0.058|0.050|0.058|0.048|0.055|0.053|0.053|0.055|0.060|0.065|0.052|0.059|0.058 |0.144|
>
> *20 ARs:*
>
> |**seed**|**AR1**|**AR2**|**AR3**|**AR4**|**AR5**|**AR6**|**AR7**|**AR8**|**AR9**|**AR10**|**AR11**|**AR12**|**AR13**|**AR14**|**AR15**|**AR16**|**AR17**|**AR18**|**AR19**|**AR 20**|**AR Avg**|**G**|
> |---|---|---|---|---|---|---|---|---|---|---|---|---|---|---|---|---|---|---|---|---|---|---|
> |1|0.058|0.055|0.045|0.045|0.053|0.056|0.046|0.033|0.028|0.048|0.045|0.038|0.052|0.043|0.045|0.041|0.048|0.043|0.040|0.040|0.042|0.118|
> |2|0.065|0.056|0.047|0.047|0.047|0.035|0.039|0.031|0.041|0.046|0.039|0.036|0.054|0.045|0.035|0.042|0.031|0.052|0.048|0.052|0.045|0.142|
> |3|0.054|0.043|0.051|0.049|0.039|0.046|0.038|0.036|0.048|0.044|0.043|0.052|0.049|0.044|0.053|0.042|0.043|0.049|0.044|0.035|0.043|0.121|

---

> > ### Author Response · Authors · 2025-11-22
> > **W11 Follow-up (1/2)**
> >
> > ### W11 Follow-up: CIs, p-values
> >
> > Thank you for your suggestion to obtain results that provide insight into statistical significance. Calculated p-values (p-vals) and bootstrapped confidence intervals (CIs) for the main results of the paper (i.e., the results from Tables 1 and 2) are included below.
> >
> > We calculate bootstrap CIs by sampling with replacement 1000 times to calculate 95% CIs. We calculate the p-values using the non-parametric two-sample Kolmogorov–Smirnov (K-S) test between the RADT results and the results of *each baseline individually* to test if the RADT results come from a significantly different distribution from the baseline results. P-values are two-tailed (i.e., they indicate significance in the observed direction of difference), and we use the approximation for evaluating Kolmogorov’s distribution and correction for small sample size as proposed by [1] and [2], respectively.
> >
> > Using an p-value cutoff of 0.05, we observe that the significance values indicated by these p-values and 95% CIs are consistent with our claim that zero-shot RADT performs comparably (no statistical significance) or superior (statistically significant and better) to all re-trained baselines, with the already-noted exception of `FetchReachObstacle-BoxWidth0.16` (where AM-Lag maintains the statistically significant superior MNC at the cost of statistically significant inferior SR).
> >
> > Regarding p-values, due to the very small sample size (3 replicates), there are only a handful of unique possible values that can result from the K-S test (the lowest being 0.04). While we can use boostrapping to obtain higher granularity in p-value, such p-values resulting from bootstrapping from such a small sample would not be very robust and the increase in granularity will not be very meaningful. It would be much more meaningful to just check if the 95% bootstrapped CIs overlap as a test of significance instead. This lack of granularity in possible p-values due to few replicates is likely why calculating p-values is not standard in the field, as none of the baselines (RbSL, AM-Lag, WGCSL), related preceding methods (DT, MGPO), or other cited works in offline RL and goal-conditioned RL (including all the benchmarking done in the well-reputed OGBench paper) include p-values in their work, and the vast majority report +/- 1 std. dev. for their error bounds. However, we acknowledge some works in the field that are primarily focused on benchmarking (e.g. OGBench) do use bootstrapped 95% CIs instead of +/- 1 std. deviation for error bounds.
> >
> > [1] Marsaglia G, Tsang WW, Wang J (2003). "Evaluating Kolmogorov's Distribution". Journal of Statistical Software. 8 (18): 1–4.
> >
> > [2] Vrbik, Jan (2018). "Small-Sample Corrections to Kolmogorov–Smirnov Test Statistics".

---

> > > ### Author Response · Authors · 2025-11-22
> > > **W11 Follow-up (2/2)**
> > >
> > > **FetchReachObstacle:**
> > >
> > > *MNC CI:*
> > >
> > > |**Box Width**|**RADT**|**AM-Lag**|**RbSL**|**WGCSL**|
> > > |---|---|---|---|---|
> > > |0.16|(0.034,0.066) |(0.008,0.013)|(0.289,0.417)|(0.447,0.474)|
> > > |0.18|(0.094,0.128) |(0.099,0.118)|(0.466,0.496)|(0.458,0.542)|
> > > |0.2|(0.135,0.187) |(0.233,0.279)|(0.558,0.583)|(0.528,0.659)|
> > > |0.22|(0.227,0.261) |(0.348,0.426)|(0.449,0.464)|(0.677,0.712)|
> > > |0.24|(0.398,0.4163)|(0.449,0.464)|(0.683,0.72) |(0.696,0.77) |
> > >
> > >
> > > *SR CI:*
> > >
> > > |**Box Width**|**RADT**|**AM-Lag**|**RbSL**|**WGCSL**|
> > > |---|---|---|---|---|
> > > |0.16|(0.983,1)    |(0.916,0.95) |(0.983,1)|(1,1)    |
> > > |0.18|(0.967,0.983)|(0.883,0.967)|(1,1)    |(0.983,1)|
> > > |0.2|(0.967,1)    |(0.917,0.967)|(1,1)    |(1,1)    |
> > > |0.22(0.967,1)    |(0.917,0.983)|(0.983,1)|(0.983,1)|
> > > |0.24 |(1,1)        |(0.867,0.95) |(1,1)    |(0.983,1)|
> > >
> > > *MNC p-vals:*
> > >
> > > |**Box Width / RADT vs…**|**AM-Lag**|**RbSL**|**WGCSL**|
> > > |---|---|---|---|
> > > |0.16 |0.04  |0.04|0.04 |
> > > |0.18 |0.989 |0.04|0.04 |
> > > |0.2|0.04  |0.04|0.04 |
> > > |0.22|0.04  |0.04|0.04 |
> > > |0.24|0.04  |0.04|0.04 |
> > >
> > >
> > > *SR p-vals:*
> > >
> > > |**Box Width / RADT vs…**|**AM-Lag**|**RbSL**|**WGCSL**|
> > > |---|---|---|---|
> > > |0.16|0.04  |0.989|0.363|
> > > |0.18|0.363 |0.04 |0.363|
> > > |0.2|0.363 |0.363|0.363|
> > > |0.22|0.363 |0.989|0.989|
> > > |0.24|0.04  |1.000|0.363|
> > >
> > >
> > > **UMazeObstacle:**
> > >
> > > *MNC CI:*
> > >
> > > |**Num Avoid**|**RADT**|**AM-Lag**|**RbSL**|**WGCSL**|
> > > |---|---|---|---|---|
> > > |1|(0.0142,0.0158)|(0.0215,0.0246)|(0.0199,0.0297)|(0.0238,0.0386)|
> > > |3|(0.0385,0.0548)|(0.0385,0.0547)|(0.0502,0.0648)|(0.0665,0.0738)|
> > > |4|(0.0568,0.0776)|(0.0633,0.0897)|(0.0738,0.0805)|(0.0795,0.1135)|
> > > |5|(0.0814,0.0907)|(0.0806,0.1178)|(0.0901,0.0997)|(0.0985,0.1083)|
> > > |6|(0.0958,0.1152)|(0.0943,0.1251)|(0.0991,0.1307)|(0.1077,0.1345)|
> > > |7|(0.1179,0.1377)|(0.1218,0.1352)|(0.2015,0.2729)|(0.1347,0.1613)|
> > >
> > >
> > > *SR CI:*
> > >
> > > |**Num Avoid**|**RADT**|**AM-Lag**|**RbSL**|**WGCSL**|
> > > |---|---|---|---|---|
> > > |1|(0.91,0.945) |(0.79,0.8388)|(0.87,0.915)|(0.7788,0.8463)|
> > > |3|(0.85,0.9)   |(0.635,0.655)|(0.13,0.235)|(0.81,0.875)   |
> > > |4|(0.845,0.885)|(0.735,0.745)|(0.72,0.8)  |(0.805,0.89)   |
> > > |5|(0.84,0.87)  |(0.75,0.77)  |(0.04,0.06) |(0.785,0.845)  |
> > > |6|(0.805,0.88) |(0.7,0.805)  |(0.715,0.83)|(0.88,0.91)    |
> > > |7|(0.795,0.87) |(0.745,0.8)  |(0,0.005)   |(0.805,0.84)   |
> > >
> > >
> > > *MNC p-vals:*
> > >
> > > |**Num Avoid / RADT vs…**|**AM-Lag**|**RbSL**|**WGCSL**|
> > > |---|---|---|---|
> > > |1|0.04  |0.04 |0.04 |
> > > |3|0.989 |0.363|0.04 |
> > > |4|0.989 |0.989|0.04 |
> > > |5|0.363 |0.363|0.04 |
> > > |6|0.363 |0.989|0.363|
> > > |7                   |0.989 |0.04 |0.363|
> > >
> > >
> > > *SR p-vals:*
> > >
> > > |**Num Avoid / RADT vs…**|**AM-Lag**|**RbSL**|**WGCSL**|
> > > |---|---|---|---|
> > > |1|0.04  |0.363|0.04 |
> > > |3|0.04  |0.04 |0.363|
> > > |4|0.04  |0.04 |0.989|
> > > |5|0.04  |0.04 |0.363|
> > > |6|0.363 |0.363|0.363|
> > > |7|0.363 |0.04 |0.363|
> > >
> > >
> > > **AMazeObstacle:**
> > >
> > > *MNC CI:*
> > >
> > > |**Num Avoid**|**RADT**|**AM-Lag**|
> > > |---|---|---|
> > > |3|(0.027,0.034)|(0.042,0.047)  |
> > > |5|(0.047,0.052)|(0.0435,0.0627)|
> > > |7|(0.057,0.077)|(0.053,0.078)  |
> > >
> > >
> > > *SR CI:*
> > >
> > > |**Num Avoid**|**RADT**|**AM-Lag**|
> > > |---|---|---|
> > > |3|(0.92,0.93) |(0.785,0.825)|
> > > |5|(0.89,0.935)|(0.745,0.85) |
> > > |7|(0.91,0.935)|(0.835,0.875)|
> > >
> > >
> > > *MNC p-vals:*
> > >
> > > |**Num Avoid/RADT vs…**|**AM-Lag**|
> > > |---|---|
> > > |3|0.04  |
> > > |5|0.363 |
> > > |7|0.989 |
> > >
> > >
> > > *SR p-vals:*
> > >
> > > |**Num Avoid/RADT vs…**|**AM-Lag**|
> > > |---|---|
> > > |3|0.04  |
> > > |5|0.04  |
> > > |7|0.04  |
> > >
> > >
> > > **HMazeObstacle:**
> > >
> > > *MNC CI:*
> > >
> > > |**Num Avoid**|**RADT**|**AM-Lag**|
> > > |---|---|---|
> > > |3|(0.027,0.03) |(0.026,0.03) |
> > > |5|(0.051,0.057)|(0.054,0.056)|
> > > |7|(0.063,0.068)|(0.05,0.067) |
> > >
> > >
> > > *SR CI:*
> > >
> > > |**Num Avoid**|**RADT**|**AM-Lag**|
> > > |---|---|---|
> > > |3|(0.84,0.86) |(0.66,0.715)|
> > > |5 |(0.84,0.895)|(0.775,0.8) |
> > > |7|(0.84,0.87) |(0.805,0.82)|
> > >
> > >
> > > *MNC p-vals:*
> > >
> > > |**Num Avoid/RADT vs…**|**AM-Lag**|
> > > |---|---|
> > > |3|0.9887|
> > > |5|0.9887|
> > > |7|0.9887|
> > >
> > >
> > > *SR p-vals:*
> > >
> > > |**Num Avoid/RADT vs…**|**AM-Lag**|
> > > |---|---|
> > > |3|0.04|
> > > |5|0.04|
> > > |7|0.04|

---

> ### Comment · Reviewer_CX5u · 2025-11-24
>
> Thank you once again for your thorough and detailed rebuttal. After reading the full paper, the revised manuscript, and all rebuttal responses, the reviewer has decided to update the score from 2 to 4 and the confidence from 3 to 4.
>
> ---
>
> First of all, thank you for your thorough rebuttal. The reviewer resolves many misunderstanding. But, there are still unresolved issue on the reviewer's mind.
>
> For example, **Environments allow passing through avoid regions and do not terminate episodes on violation at evaluation. The authors mention that SR can remain high while MNC is non-trivial, decoupling safety from success. The reviewer thinks that this is non-fair evaluation.**
>
> ---
> > **LLM and presentation issue**
>
> While the reviewer appreciate the additional analyses and revisions, several concerns remain what appears to be extensive LLM style text patterns throughout the manuscript. The reviewer summarize the issues that, in own view, were not addressed.
>
> 1. Unusal mathematical notation
>     - In line 973, the manuscript uses $\int_\sqcup$ to denote the current state. The state $s$ can be written simply.
>     - Across the manuscript, at least three distinct notational conventions appear, that is, $2 * d_s$ and $2 * 10^6$, $n_{\text{avoid}} \cdot d_s$, and $2 \times d_h$. Could the authors explain how these notational differences arose?
>
> 1. Excessive quotation marks and italics
>     - The manuscript contains an unusually large number of double quotes, "something", in places where they are unnecessary. In LaTeX, keyboard-typed ASCII quotes (“) and LLM’s quotes (“) are encoded differently at the PDF level.
>     - italicization appears in too many sentences, often on function words or arbitrary nouns, for example, *training* time,  *total* amount of time, pass *through* any b, they *effectively* fails to satisfy.
>         - **What is the purpose of this italicization?** It is unclear how the authors are deciding what to italicize.
> 2. Figures
>     - Some figures appear to be direct screenshots from wandb without proper reformatting.
>
> ---
> > **Doubl-blind**
>
> - In line 1025, the authors explicitly mention using **Harvard University’s Kempner Institute compute cluster**. To the best of my knowledge, this can be grounds for desk rejection.
>
> ---
> > **Final remark**
>
> Overall, the reviewer acknowledges the importance of the evaluation and the practical motivation of this work, and the reviewer finds the problem setting genuinely interesting. However, in terms of novelty, the proposed method appears to be a composed extension of existing components applied to this reach-avoid setting.
>
> The reviewer would not mind if this manusciprt if accepted, but believes we need to carefully reflect on the issues raised above.

---

> > ### Author Response · Authors · 2025-11-25
> >
> > Thank you for taking the time to engage in these insightful discussions over the past week, and we appreciate you raising your score.
> >
> > **Episode termination:**
> >
> > Regarding episodes terminating upon violation, we see from our response to W10 that this is a comprehensive strategy to evaluate reach-avoid behavior, and that the performance of a reach-avoid policy where episodes do terminate upon violating an avoid-region can be trivially derived from our setup (i.e., any trajectory with non-zero MNC can be labeled a failure, if this is happens to be the criteria for your specific problem). Additionally, the reach-avoid and safe RL works prior to ours (including the baseline works of RbSL and AM-Lag) were evaluated under the same strategy we used, with goal-reaching SR and cost metrics disentangled from one another.
> >
> > **Writing style and LLMs:**
> >
> > **We would like to emphasize again that we did not use LLM assistance in writing this paper**. We apologize for any inconsistencies in mathematical notation throughout our paper; we have multiple authors editing the paper and we occasionally fail to catch notational inconsistencies. We will do a thorough pass through the paper and fix any inconsistencies in the camera-ready version. The state notation in line 973 of the Appendix is due to the accidental usage of \mathcal{} (which we use to denote state space) instead of \mathbf{}. We recognize there needs to be a thorough proofreading through the Appendix to catch and fix typos like this, as well.
> >
> > We acknowledge there may be disagreements in writing style regarding when and how often to emphasize words using italics. While we believe this is a subjective preference of writing style, we use italics if there are subtle but important differences that a certain keyword makes. For example, we wanted to emphasize the word “training” when discussing pure *training* time, which excludes intermittent evaluation steps, to contrast it with the *total* training, which includes both training steps and evaluation steps. When we italicize words, it is our intent to help with the reading flow, but if it is significantly impending reading flow we are happy to make modifications. Regardless, we can affirm this is a conscious and intentional choice of ours, and not a result of LLM usage.
> >
> > **Double-blind:**
> >
> > The mention of using Kempner Institute’s compute cluster does not reveal our identity. **Kempner’s AI cluster resides in the Massachusetts Green High Performance Computing Center (MGHPCC) and is used by a variety of universities and private institutions. Its usage is not limited to any particular institution/organization despite its namesake affiliation with Harvard, which manages the cluster.** We mention Kemper’s compute resources to properly credit them, in the same way we would credit the usage of Amazon AWS’s or Google’s cloud compute resources; this does not imply any further affiliation with the host organization of those compute resources. However, we are happy to remove this line if it is an issue, and apologize if it has caused any misunderstandings.

---

### Official Review · Reviewer_yjF7 · 2025-10-31

**Soundness:** 3
**Presentation:** 3
**Contribution:** 3
**Rating:** 6
**Confidence:** 3

**Summary:**

This paper introduces RADT (Reach-Avoid Decision Transformer), an offline RL framework for goal-conditioned reach-avoid tasks that addresses flexible specification of avoid regions at evaluation time. The method uses a prompting-based approach where goals and avoid regions are encoded as discrete tokens in the input sequence to a causal transformer.

**Strengths:**

- Encodes both goals and avoid regions as prompt tokens, decoupling reach-avoid specifications from state representation and enabling zero-shot generalization to arbitrary avoid region counts, locations, and sizes
- Creative strategy that generates trajectory pairs with opposite avoid success labels, allowing the model to learn from both successful and unsuccessful demonstrations without requiring expert data
- Eliminates brittle reward/cost function design by learning directly from hindsight-relabeled trajectories, addressing practical challenges of balancing conflicting objectives

**Weaknesses:**

- Only 2 robotics environments with relatively simple geometric constraints; no high-dimensional state spaces (e.g., pixel observations) or complex avoid region shapes beyond boxes/spheres
- Training time (72 GPU hours mentioned in appendix), memory overhead, and model size (GPT-2 architecture) not compared against baselines; acknowledged in limitations but critical for practical deployment
- RbSL/AM-Lag baselines use impassable obstacles (Figure 3b) rather than passable avoid regions, making direct comparison questionable
- Table 8 shows introducing expert trajectories improves avoid performance on OOD box sizes but degrades goal-reaching SR, suggesting reliance on random-policy data may limit ceiling for certain configurations

**Questions:**

1. What is the wall-clock training time, memory usage, and inference latency compared to RbSL/AM-Lag? Is the zero-shot capability worth the computational overhead?

2. How would RADT handle complex, non-convex avoid regions (e.g., L-shaped obstacles, multiple disconnected regions)? Could you use learned embeddings instead of hand-crafted box/sphere representations?

3. Since RbSL/AM-Lag use impassable obstacles (no training data violates avoid regions), can you evaluate all methods on the same passable-obstacle setup to ensure fair comparison?

4. How does RADT perform with pixel observations or point cloud inputs? Would you need to learn avoid region embeddings from visual features?

5. What is the maximum number of avoid regions RADT can handle before prompt length becomes prohibitive? Does attention mechanism degrade with very long prompts?

---

> ### Author Response · Authors · 2025-11-20
> **Response to Reviewer yjF7 (1/4)**
>
> ***(Please note this is part 1 in a series of 4 posts.)***
>
> Thank you for the helpful feedback and insightful comments! We are grateful that you acknowledge the practical value of creating a reward-free approach to reach-avoid learning that decouples avoid-region specifications from the state space and has zero-shot capabilities. We also appreciate that you find our hindsight-relabeling strategy to be a creative solution to achieving this.
>
> We have addressed your comments/suggestions below and have also included the results of new experiments that we hope provide further insight and strengthen our paper. We tag sections that contain new experiment results with a **[new results]** tag in the header. Additionally, a revised version of the paper has been uploaded to OpenReview. If you find your concerns are addressed after reviewing our responses, we would greatly appreciate it if you considered raising your score. If there are any remaining questions/comments, please let us know and we would be happy to address them. Thank you!

---

> ### Author Response · Authors · 2025-11-20
> **Response to Reviewer yjF7 (2/4)**
>
> ## W1 (also Q2, Q4) **[new results]**: Complex avoid-regions, learned avoid-region representations, pixel-based observation spaces
>
> We acknowledge that it is important that a reach-avoid method is generalizable to complex avoid-region shapes and representations. In terms of representation, we used three distinct avoid region representations (box corner representation, centroid + radius representation, and soft-point representation for the discrete state space for cellular reprogramming) to demonstrate that our approach of prompted reach-avoid training does not depend on a particular vector representation of avoid region.
>
> In terms of handling complex avoid-region shapes, we have implicitly demonstrated how RADT can generalize to complex shapes in the maze tasks that involved many avoid-regions (Section 5 Results 2 and 4 “Extremities of OOD Generalization”, Appendix G.2), since the introduction of a high number of avoid regions within a small maze inevitably results in many overlapping avoid regions that, in composition create avoid regions of more complex shapes. We have attached images of such scenarios in Figure 10 and have included this discussion in Appendix G.4 “More complex avoid-region shapes” in the revised paper.
>
> Nonetheless, we recognize the value of being able to adapt to even more generalizable avoid-region representations that do not assume precise knowledge of the vectorized structure of the state space and avoid-region token representation, which may not be readily available in all environments. Therefore, we have conducted a new experiment described in Appendix G.5 (“Image-Based Observation Spaces and Avoid-Region Representations”), Table 12, and Fig. 12 in the revised paper. To explore RADT’s performance in handling 1) high-dimensional observation spaces and 2) unstructured, flexible avoid-region token representations that can capture abstract avoidance desires without knowledge of a pre-defined vectorization, we conduct a proof-of-concept experiment applying RADT to an image-based version of `UMazeObstcale` that we call `UMazeObstacleImage`. Here, we replace the observation space for `UMazeObstacle` with an image-based observation space. To obtain state $\mathbf{s}_t$, we take an 224x224 image of the maze with the agent's current position visualized as a green sphere, then pass the image through Torchvision's pre-trained ResNet-18 model. We then take the representation outputted by the fully-connected layer at the end of the ResNet-18 architecture as our state representation, giving us a 1000-dimensional state vector $\mathbf{s}_t \in \mathbb{R}^{1000}$ (Fig. 12). Similarly, we acquire avoid-region tokens and goal tokens via the same process, except we replace the green sphere representing the agent's position in the original image with blue objects representing the avoid-regions and red spheres representing goals, respectively. Therefore, the avoid-region tokens $\mathbf{b}_j \in \mathbb{R}^{1000}$ and the goal token $\mathbf{g} \in \mathbb{R}^{1000}$ have the same dimensionality as the state vectors.
>
> We then use the same hindsight-relabeled training trajectories from our 1`UMazeObstacle-NumAvoid1` dataset for training, but replace all the state, avoid-region, and goal tokens with these new representations. We benchmark against AM-Lag, RbSL, and WGCSL as baselines using the same evaluation process described for the `UMazeObstacle` environments. As shown by the results in Table 12 (simplified version provided below), RADT achieves superior MNC and comparable or superior goal-reaching SR to all baselines. The SR values here are lower than the SR results from the original `UMazeObstacle-NumAvoid1` task, but this is to be expected given the much more difficult nature of a high-dimensional state space. The SR values are also within the range of typical SR values that SOTA GCRL approaches achieve on image-based, goal-reaching maze tasks (that do not include an avoidance objective), according to the results from OGBench (which do not involve an avoidance component). While there is room for additional improvement through further optimization of hyperparameters, this preliminary study demonstrates RADT's strong potential in being adapted to both high-dimensional tasks and flexible avoid-region representations that do not require precise knowledge of a strict, vectorized structure to specify avoid-regions.
>
> ||RADT|AM-Lag|RbSL|WGCSL|
> |---:|:---:|:---:|:---:|:---:|
> |**MNC (1e-2)**|1.1|1.6|1.87|2.55|
> |**SR**|0.617|0.427|0.517|0.62|

---

> ### Author Response · Authors · 2025-11-20
> **Response to Reviewer yjF7 (3/4)**
>
> ## W2 (also Q1): Training time and memory overhead
>
> On average, RADT takes around 10 hours to converge on a strong policy using 1 GPU (depending on the task) in terms of pure training time. The total amount of time depends on how often we perform validation/evaluation. The max memory utilization during training hit 40 GB, typical of DT-based approaches. We acknowledge this is a higher computation overhead compared to baselines, which can converge on a strong policy using 1 GPU within 2 hours of training time, with the maximum memory utilization during training hitting 20-30 GB. However, given that all training was done with a single GPU and a strong policy can be achieved within a day, we do not think RADT is prohibitively expensive to train. Additionally, RADT has the additional benefit of being able to generalize zero-shot to novel avoid-region configurations at test time while other methods do not, significantly saving test-time computation. For this reason, this tradeoff is not unreasonable, as RADT *front-loads* the computational expense to the initial training stage and removes the necessity to re-train on individual avoid specifications downstream. We have included these details in the revised version of Appendix B.2 “Compute Resources.”
>
> ## W3 (also Q3): Impassable objects and baseline comparison
>
> Your concern regarding comparing models trained with passable vs impassable objects is valid. However, in our study, we actually train *all* models, including baselines like RbSL and AM-Lag, on training trajectories that involve passable objects (as described in Section 5 Results 1 “Environment” and Section 5 Results 2 “Environment”). So while the RbSL and AM-Lag results in the original RbSL paper (Cao et al. 2024) were based on training with impassable objects, all the RbSL and AM-Lag results in our work were based on training with passable objects. The fact that the original RbSL model used training trajectories produced in an environment with impassable objects was actually one of the major criticisms we had regarding that work, since there were no failure demonstrations in the training dataset where the trajectory fails to avoid the obstacle (by passing through it), making the learning of avoid behavior more trivial (i.e., the training data is not completely suboptimal and fails to satisfy Property 2.2 in Section 2). Thus, in our study, we wanted to see if these baselines still perform well under the regime we care about, which is when there *are* failure/suboptimal demonstrations in the dataset.
>
> ## W4: Random-policy data ceiling
> We acknowledge that only using random-policy data does limit the ceiling of performance, as we demonstrate with the results in Table 9. However, the performance under this ceiling is precisely one of the major points we want to explore in our study. We want to explicitly evaluate the performance potential of algorithms in the absolute worst-case scenario (where all the training data is as suboptimal as can be) as a “lower bound” of performance. This way we can most clearly demonstrate the potential of these algorithms for learning super-demonstration behavior (Section 2, Property 2).
>
> ## Q1-Q4:
>
> Q1-Q4 have been addressed in our responses to W1-W3 above.

---

> ### Author Response · Authors · 2025-11-20
> **Response to Reviewer yjF7 (4/4)**
>
> ## Q5 **[new results]**: Effects of long prompts
>
> We have conducted an additional analysis of the “Extremities of OOD generalization” experiments that show how increasing prompt length at inference time affects attention allocation. The results are included in Appendix G.2 “Effects on attention mechanism” and Fig. 11 of the revised paper. Specifically, we check to make sure that long prompt lengths do not result in critical attention defects such as consistent attention sinks or uniform/zero attention.
>
> For each episode during evaluation, we calculate the attention weights, averaged across all attention heads, that the model gives to the first 30 tokens of the input sequence (including both the prompt and the early state/action tokens in the trajectory history) when generating the *last* action token of the trajectory. We plot these values for 1, 5, 15, and 20 avoid-regions (Fig. 11). We choose to analyze the attention mechanism at the last action token as this is the most challenging token in terms of long-range dependence on the prompt. We notice no major attention sinks or debilitating attention uniformity across the prompt tokens that result from long prompt lengths, as there is still diversity in attention distributions across different trajectories even when there is a large, OOD number of avoid-region tokens in the prompt (e.g. 15-20).
>
> However, we do note a few interesting observations.
>
> * First, RADT pays very little to no attention to the state/action tokens in the early portion of the trajectory history (occurring right after the prompt tokens) when generating the last action in the trajectory. This is likely because the model has learned that it is sufficient to make action predictions with a Markovian-like policy, taking into account the more recent states while ignoring historical states from earlier parts of the trajectory history.
>
> * Second, while there are no major attention sinks and uniformity observed, we do notice a drop in attention to individual avoid-region tokens on *average* with very OOD prompt lengths (e.g. 15 and 20 avoid-regions). Intuitively, this is to be expected. It is more challenging for a model to keep track of a high number of different avoid-regions, so it must be selective in distributing its attention only to the most relevant ones at any particular action; it cannot afford to constantly pay attention to every single avoid-region like it can when there are only a couple. Additionally, this would be consistent with our discussion in Appendix G.4 “More complex avoid-region shapes.” As avoid-regions increase in the `MazeObstacle` environments, avoid-regions increasingly start to overlap with one another to compose effectively larger avoid-regions, so it would make sense that the significance of individual avoid-regions would decrease given the large amounts of overlap. While we acknowledge the decrease in average attention to individual avoid-region tokens with longer prompts can be a limitation, it is more likely that the difficulty of the task itself would limit the number of avoid-regions before issues related to attention across long contexts arise. For example, 20 avoid regions in the maze environment already makes the environment way too crowded.
>
> * Third, RADT always puts a lot of attention weight on the *goal token* at the end of the prompt, regardless of how many avoid-region tokens precede it. This is not surprising, as the goal token is likely the most relevant token at the last action of the trajectory, and there is ever only one goal token. However, it does show that RADT is robust to increasing prompt length, as it is still able to identify the critical nature of the goal token despite the many more potentially "distracting" tokens in longer prompts.
>
> The main trends in Fig. 11 are summarized in this table (attn. @ last action token in traj.):
> |**# of avoid-regions**|**1**|**5**|**15**|**20**|
> |---:|:---:|:---:|:---:|:---:|
> |**# of clear attention sinks**|0|0|0|0|
> |**Mean attention to early state/action tokens**|0|0|0|0|
> |**Mean attention to avoid-region tokens**|0.247|0.121|0.047|0.036|
> |**Mean attention to goal token**|0.281|0.238|0.228|0.223|

---

> ### Comment · Reviewer_yjF7 · 2025-11-25
> **Double-Blind Concerns**
>
> According to the Kempner Institute handbook (https://handbook.eng.kempnerinstitute.harvard.edu/s1_high_performance_computing/kempner_cluster/overview_of_kempner_cluster.html#who-is-eligible-to-use-the-kempner-institute-ai-cluster), Section **1.2.2 _“Who is eligible to use the Kempner Institute AI Cluster?”_** states:
>
> > In general terms, the cluster is reserved for individuals who are either associated with a lab that is affiliated with the Kempner Institute or those who are engaged in a project that has received approval from the Kempner Institute.
>
> Because of this restriction, explicitly mentioning the use of the Kempner Institute AI Cluster in the submission effectively discloses the authors’ institutional affiliation. This appears to contradict ICLR’s double-blind review policy, which requires that identifying institutional information not be revealed in the paper.

---

> > ### Author Response · Authors · 2025-11-25
> >
> > Thank you for following up. We would like to clarify that this does not reveal our institutional affiliation, as Kempner does not publicly release the list of all external organizations/labs/institutes/companies that have been approved to use its cluster. We are only crediting them properly as we would credit any other resource (e.g. Python libraries, NVIDIA GPUs, etc.) that we used to conduct our work. For example, the OLMo-7B work (https://arxiv.org/pdf/2501.00656) from Allen Institute for AI (which is unaffiliated with the Kempner Institute, and none of the authors have Kempner Institute listed as an affiliation) acknowledges using Kempner Institute's compute resources on their Huggingface page (https://huggingface.co/allenai/OLMo-7B) and release article (https://allenai.org/blog/hello-olmo-a-truly-open-llm-43f7e7359222), in the same section where they credit AMD, CSC (Lumi Supercomputer), DataBricks, and other compute resources hosted by non-affiliated organizations.  Nonetheless, we have removed the line for now, as it seems to be causing some misunderstandings.
> >
> > We hope you may engage with us in technical discussion, as we have put in a lot of effort into addressing all the technical concerns you have brought up.

---

### Author Response · Authors · 2025-12-04
**Author Final Remarks (Part 1/2)**

We thank the reviewers for all the feedback they have provided in their initial reviews and for engaging in insightful discussions with us over the past couple weeks. We have summarized the main takeaways from the discussion below.

The main additional experiments requested by the reviewers were: 1) an experiment demonstrating RADT’s performance in a challenging, high-dimensional environment (e.g., pixel-based environment), 2) an experiment demonstrating RADT’s ability to handle flexible, learned avoid-region token representations that do not depend on precise knowledge of a particular vectorized structure, 3) an experiment demonstrating the effects of increasing prompts lengths on the behavior of the self-attention mechanism, and 4) an additional ablation study explicitly demonstrating the impact on RADT’s performance with the removal of the avoid-region relabeling component.

* **Requests (1) and (2):** We have addressed the first two requests by conducting a new preliminary demonstration of RADT applied to an image-based version of `UMazeObstacle` that we call `UMazeObstacleImage` (as described in Appendix G.5 “Image-Based Observation Spaces and Flexible Avoid-Region Representations”, Table 12, and Fig. 12 in the revised paper). Here, we replace the observation space for `UMazeObstacle` with an image-based observation space. States, goals, and avoid-regions are initially represented as 224x224 image-based renderings of the maze with colored entities indicating the agent’s positions, goals, and avoid-regions, respectively. The initial observations are then passed through a pre-trained ResNet-18 model to obtain 1000-dimensional vectorized representations that are fed into RADT. We find that RADT achieves **superior MNC and comparable or superior goal-reaching SR to all baselines**, with goal-reaching SR values within the range of typical SR values that SOTA GCRL approaches achieve on image-based, goal-reaching maze tasks (that do not include an avoidance objective), according to the results from OGBench. This demonstrates that RADT is **applicable to high-dimensional tasks** and **flexible avoid-region representations** that do not require precise knowledge of a strict, vectorized structure to specify avoid-regions.

* **Request (3):** We have addressed the third request with a new study analyzing the effects of increasing prompt length on the distribution of attention weights in the prompt (as described in Appendix G.2 “Effects on attention mechanism” and Fig. 11 of the revised paper). Specifically, we check to make sure that long prompt lengths do not result in critical attention defects (e.g., attention sinks or uniform/zero attention) by calculating the average attention scores to each token in the prompt when generating the last action token of the trajectory. We calculate these values for evaluation episodes involving 1, 5, 15, and 20 avoid-regions. We notice **no major attention sinks or debilitating attention uniformity** across the prompt tokens that result from long prompt lengths. Additionally, the **attention to the goal token remains consistently strong regardless of prompt length**, demonstrating that RADT is robust to increasing prompt length, as it is still able to identify the critical nature of the goal token despite the many more potentially "distracting" tokens in longer prompts. We do notice a drop in attention to individual avoid-region tokens on *average* with very OOD prompt lengths (e.g. 15 and 20 avoid-regions), which may be a limitation with very long prompts. However, this can also be attributed to the fact that large numbers of avoid-regions in the Maze environments lead to a lot of overlap between individual avoid-regions, so it would make sense that the significance of individual avoid-regions would decrease given the large amounts of redundancy. Upon request by Reviewer CX5u during the follow-up discussion to our rebuttal (under our comment titled “W8 Follow-up”), we have also included precise standard deviation and mean values for the attention scores on a *per-token level* across *3 different training seeds*. This analysis demonstrates: 1) the attention distributions and trends for prompt tokens are **consistent across training seeds**, 2) there are **no point abnormalities** in the attention mechanism, as no avoid-region token *positions* have a drastically different mean or standard deviation from the rest, and 3) there is **healthy distributional spread of attention weights *across episodes***, with no concerningly low standard deviation values that would suggest a lack of dynamic adaptation to the particular scenarios of different episodes.

---

> ### Author Response · Authors · 2025-12-04
> **Author Final Remarks (Part 2/2)**
>
> * **Request (4):** To address the fourth request, we have conducted an explicit ablation study of avoid-region relabeling (Section 5 Results 4 “Ablation of avoid-region relabeling”, Appendix G.1, Table 11 of the revised paper), in addition to the existing set of ablation studies in the initial paper version. We observe that the complete ablation of avoid-region relabeling significantly impairs RADT’s MNC performance across both ID and OOD box sizes, demonstrating the critical nature of avoid-region relabeling.
>
> Additionally, there were a couple non-technical concerns brought up during the discussion that we have addressed in our responses to individual reviewers. We summarize our positions below:
>
> * **LLM Usage:**  Despite Reviewer CX5u’s accusation, **we strongly affirm that we did not use LLM assistants in writing our paper**, as we are very particular about the way we phrase and present our ideas. It is easy to see from the discussions below that most of the examples the reviewer points to as “suspicious” are just editorial/formatting mistakes (mostly in the Appendix) due to lack of proofreading on our part, or are related to our particular writing style (i.e., the tendency to emphasize keywords using italics to point out nuanced conceptual differences). We do acknowledge that the paper and the Appendix could use more proofreading, and will make a very rigorous pass to make sure to fix minor mistakes/inconsistencies for the camera-ready version.
>
> * **Double-blind Concern:** We would like to clarify that crediting the usage of Kempner Institute’s compute resources does not reveal our institutional affiliation and **does not violate the double-blind policy**. As we have explained in our responses, external organizations/institutions that are *not* affiliated with the managing institution (Harvard Kempner Institute) can and do use Kempner’s compute resources given approval. The list of which external entities are approved to use Kempner’s compute resources is *not publicly released*; therefore, the usage of Kempner’s resources is not identifying information and does not reveal any group affiliation. For example, the OLMo-7B work (https://arxiv.org/pdf/2501.00656) from the Allen Institute for AI (which is unaffiliated with the Kempner Institute, and none of the authors have Kempner Institute listed as an affiliation) acknowledges using Kempner Institute's compute resources on their Huggingface page (https://huggingface.co/allenai/OLMo-7B), in the same section where they credit AMD, CSC (Lumi Supercomputer), DataBricks, and other compute resources hosted by non-affiliated organizations. We treat and credit Kempner’s compute cluster like we would any external resource that has helped us complete this project, the same way we would credit AWS’s cloud compute resources, Google’s cloud compute resources, etc. We claim no affiliation with the institutions/organizations managing such compute resources. To prevent further misunderstanding, however, we have removed this part from the paper.
>
> There are a handful of other, more nuanced points of discussion not included in this post, but are included in the full discussions with the individual reviewers below.
>
> Again, we thank the reviewers for taking the time and effort to carefully read our paper and provide helpful feedback! We believe we have addressed all of the reviewers’ concerns in our responses, and we appreciate that our paper is now stronger as a result.

---

### Meta-Review · Area_Chair_gY8x · 2026-01-07

**Summary:**

RADT reframes reinforcement learning for reach-avoid tasks (of which there are many in robotics) as a prompt‑conditioning problem, where goals and avoid‑regions are encoded as tokens for a Decision Transformer. By using hindsight relabeling on purely suboptimal offline data, it learns what successful and failed trajectories look like without any rewards or expert demonstrations. This lets a single trained model generalize zero‑shot to entirely new avoidance constraints at test time, outperforming baselines that must be retrained for each scenario.

Reviewers appreciate the simple idea to decouple reach-avoid specifications from the state representation by treating them as discrete prompt tokens. They also appreciate the strong results in OOD scenarios.

Concerns are explained in the next field.

**Reviewer Concerns:**

They also raise several concerns:

Addressed

- Training time and memory overhead:
    - Authors provide details, acknowledge the increased (but non-prohibitive) requirements, but point to improved capabilities.
- Concerns about baselines being trained on different data and being treated unfairly.
    - Rebuttal clarifies that they are in fact trained on the same data.
- Small number of seeds for RL runs (3 seeds):
    - Authors point to prior works using offline decision transformers (which are more stable to train than most conventional RL techniques) that also only use 3-5 seeds. Authors also provide confidence intervals, which are reassuring.
- How sensitive is the method to the joint loss coefficient hyperparameter?
    - Authors report reasonably stable preliminary results around their original choice of 1.0.

Partially addressed / unaddressed

- Simple environments: only 2 robotics environments with simple avoid region geometries and low-dim state spaces.
    - Rebuttal adds new preliminary experiments with an image-based maze task, and with complex avoid regions. However, since this was a rebuttal experiment, limited details are available to fully judge this significant addition.
    - Rebuttal also argues that they had to pick environments where pure reach behaviors were easy to learn, so that they could shine a spotlight on avoid behaviors.
- While the paper speaks of safety-criticality, which would normally suggest hard constraints on avoid regions, the evaluation does not seem to be consistent with this. Strict avoidance is not enforced, and episodes are not terminated on violation. This comes up in issues raised by CX5u and Tx2w re evaluation setups and metrics.
    - The author response to this attempts to justify soft avoidance, but the rest of the paper up until the experiments does not seem to be consistent with this framing. There is some work to be done on making a consistent problem formulation. For example, if we care about soft avoidance, and could assign weights to the two objectives, does this not permit a simple application of standard hindsight-relabeling offline RL approaches where the reward label would now be “goal reaching - \lambda times number of violations”?
    - Author response to Tx2w also defends the choice of reporting two metrics, but the paper in this case would benefit from a more standard treatment of multi-objective problems, e.g. by tracing the performance-safety pareto-frontiers for multiple methods.
- Is it truly generalizable to arbitrary avoid region shapes? How many avoid regions can RADT handle?
    - Authors inspect attentions as number of avoid regions increase and report no irregularities. They do not directly report task performance degradation curves at these larger numbers of avoid regions.

There was discussion about violation of double blind. However, upon consideration and discussion with SAC, we rule that this does not quite rise to the level of violating double blind, even if it may leak some non-zero (but minimal) information about authors’ location / institutional affiliations. Authors are advised to avoid sharing such information (or indeed information about their areas of research outside this submission — they mention “cell biology” is “one of our primary application domains” in their rebuttal responses) in future double-blind submissions (acknowledgements are typically added in a camera-ready after peer review is complete and double blind considerations no longer matter).

**Reviewer Scores:**

yjfF7: 6→ 6 (ignoring the double blind point)

CX5u: 2→4 (ignoring the double blind point)

Tx2w: 6→6

Overall, this is borderline on reviewer ratings, but my assessment is that there remain somewhat fundamental issues with problem formulation (hard vs soft constraints) as well as with thoroughness of evaluations (more interesting environments, continuing the work that began in the rebuttal period).

---

### Decision · Program_Chairs · 2026-01-26

Reject